JCB Journal of Cell Biology

# Roles of Srs2/PARI-family DNA helicases in NoCut checkpoint signaling and abscission regulation

Monica Dam[1,2,3,4]*, David F. Moreno[1,2,3,4]*, Nicola Brownlow[5,6]*, Audrey Furst[1,2,3,4], Coralie Spiegelhalter[1,2,3,4], and Manuel Mendoza[1,2,3,4]

**The coordination of chromosome segregation with cytokinesis is crucial for maintaining genomic stability. Chromatin bridges, arising from DNA replication stress or catenated chromosomes, can interfere with this process, leading to genomic instability if not properly resolved. Here, we uncover that the budding yeast DNA helicase Srs2 is essential for delaying abscission in the presence of chromatin bridges, thereby preventing chromosome breakage during cytokinesis. We also find that its human paralog PARI delays abscission-associated events, including midbody severing and actin-patch disassembly, in human cells with chromatin bridges. Although PARI depletion does not lead to increased bridge breakage or binucleation, our data indicate that PARI has nonessential functions within the Aurora B–mediated abscission checkpoint pathway. These findings establish a key role of Srs2 in NoCut checkpoint signaling in yeast, and suggest a functionally related role of PARI in coordinating abscission timing with chromatin bridge resolution in human cells.**

## Introduction

The proper coordination between chromosome segregation and cell division (cytokinesis) is critical for maintaining genomic stability and supporting cell proliferation. Chromatin bridges that persist during cytokinesis can arise from various sources, including defects in DNA replication, chromatin condensation, DNA decatenation, and dicentric chromosomes formed by telomere fusion, among other anomalies (Ganem and Pellman, 2012; Mankouri et al., 2013). If these chromatin bridges are not adequately resolved, they can either be damaged during cytokinesis or cause cytokinesis failure, both of which have detrimental effects on cell proliferation.

Cytokinesis failure can result in binucleation and tetraploidy, conditions that are linked to the promotion of tumorigenesis (Fujiwara et al., 2005). Notably, a large fraction of human carcinomas are thought to originate from tetraploid cells, in part due to failed cytokinesis (Zack et al., 2013). Conversely, damage to unsegregated DNA during cytokinesis can lead to aneuploidy, gross chromosome rearrangements, and impaired cell growth (Janssen et al., 2011; Umbreit et al., 2020). At the organismal level, these defects can contribute to severe pathologies such as T-cell lymphoma and microcephaly (Woodward et al., 2016; Martin et al., 2016). Therefore, chromatin bridges serve as both indicators of genomic instability and precursors to cellular transformation. Understanding how cells respond to chromatin bridges of different origins remains a critical question.

Given the potentially damaging consequences of chromatin bridges, cells have evolved mechanisms to mitigate their harmful effects. One such mechanism is the "NoCut" checkpoint, which delays the final step of cytokinesis, plasma membrane abscission, in budding yeast cells with chromatin bridges (Norden et al., 2006; Mendoza et al., 2009; Amaral et al., 2016). This checkpoint is conserved in animal cells, where it is known as the "abscission checkpoint" (Steigemann et al., 2009; Carlton et al., 2012; Agromayor and Martin-Serrano, 2013; Andrade and Echard, 2022). The NoCut-dependent delay in abscission is thought to protect chromatin bridges from damage and reduce the likelihood of abortive cytokinesis, which can occur if chromosomes obstruct the division site. Consequently, defects in the NoCut pathway in otherwise normal cells can lead to cytokinesis failure, tetraploidy, and genome rearrangements, all of which are hallmarks of cancer and may contribute to tumorigenesis.

In both yeast and animal cells, the function of the NoCut checkpoint depends on the association of the kinase Aurora B with microtubules at the division site (the spindle midzone or intercellular bridge [ICB]), suggesting that chromatin bridges are detected at this location. Aurora B regulates the timing of abscission by controlling membrane remodeling events at the division site. In budding yeast, this regulation involves effectors such as Boi2, which influences the evolutionarily conserved exocyst complex that facilitates the fusion of secretory vesicles with the plasma membrane (Masgrau et al., 2017). In human

[1]Institut de Génétique et de Biologie Moléculaire et Cellulaire, Illkirch, France; [2]Centre National de la Recherche Scientifique, UMR7104, Illkirch, France; [3]Institut National de la Santé et de la Recherche Médicale, U964, Illkirch, France; [4]Université de Strasbourg, Strasbourg, France; [5]Centre for Genomic Regulation (CRG), The Barcelona Institute of Science and Technology, Barcelona, Spain; [6]Universitat Pompeu Fabra (UPF), Barcelona, Spain.

*M. Dam, D.F. Moreno, and N. Brownlow contributed equally to this paper. Correspondence to Manuel Mendoza: mendozam@igbmc.fr.

cells, Aurora B modulates abscission by regulating the endosomal sorting complex required for transport (ESCRT) III, a midbody-associated complex crucial for abscission timing in animal cells (Carlton et al., 2012; Addi et al., 2018). In addition, the human abscission checkpoint has been proposed to recognize DNA bridges through recruitment of topoisomerase II alpha (Topo IIα) to DNA intertwines, which would then signal through a MRN-ATM-CHK2-INCENP pathway (Petsalaki and Zachos, 2021; Petsalaki et al., 2023). While the regulation of abscission timing downstream of Aurora B is becoming clearer, the mechanisms by which chromatin bridges are sensed upstream of Aurora B remain largely unknown.

Previous findings from our laboratory suggested that the yeast NoCut checkpoint does not merely respond to chromatin bridges, but specifically to bridges associated with DNA replication stress, possibly by detecting specific molecules associated with these bridges (Amaral et al., 2016; Amaral et al., 2017). A similar phenomenon was also observed in human cells (Petsalaki et al., 2023). DNA helicases, known for their roles in DNA repair and replication stress resolution, have been implicated in resolving chromatin bridges in human cells, where they prevent bridge breakage and cytokinetic failure (Chan et al., 2018). Given these findings, we hypothesized that yeast helicases such as Srs2 could influence cytokinesis by resolving chromatin bridges during replication stress. Srs2 is a multifunctional helicase involved in DNA repair and homologous recombination regulation, particularly in dismantling Rad51 filaments (Marini and Krejci, 2010), and shares functional similarities with human helicases like PARI (Moldovan et al., 2012; Burkovics et al., 2016; Mochizuki et al., 2017). In this study, we explored the role of Srs2 in cytokinesis and were initially expecting it to resolve chromatin bridges that would otherwise delay abscission. Surprisingly, we found that Srs2 has an additional function: it is required for the inhibition of abscission in response to chromatin bridges. This finding suggests that Srs2 plays an active role in the NoCut checkpoint, acting as a mediator of chromatin-based signaling during cytokinesis. We also found that the human homolog PARI delays abscission-associated events (including midbody severing and actin-patch disassembly) in cells with chromatin bridges, although PARI depletion does not lead to increased chromatin breakage or binucleation. Further, our results indicate that PARI acts within the Aurora B–mediated abscission checkpoint pathway. Together, these findings support a conserved role of Srs2/PARI-family helicases in coordinating chromatin bridge recognition with cytokinesis timing, thereby contributing to genome stability during cell division.

## Results

### Srs2 promotes the resolution of anaphase chromatin bridges following replication stress

The budding yeast DNA helicase Srs2 is known for its role in inhibiting homologous recombination during the S phase and removing replication protein A (RPA) from chromatin (Pfander et al., 2005; Marini and Krejci, 2010; Dhingra et al., 2021), but its involvement in chromosome segregation and cytokinesis has not been explored. To investigate whether Srs2 contributes to the coordination of chromosome segregation and abscission, we

monitored chromosome segregation and RPA localization (visualized using histone Htb2-mCherry and Rfa2-GFP) by time-lapse imaging in wild-type (WT) and Srs2-deficient budding yeast cells. Given the importance of Srs2 in suppressing replication intermediates during replication stress, we also examined chromosome segregation in srs2Δ cells exposed to hydroxyurea (HU), which perturbs DNA replication.

Treatment with a 3-h pulse of HU generated late-segregating chromatin bridges in WT cells, as previously reported (Amaral et al., 2016), and this phenotype was exacerbated in cells lacking SRS2 (Fig. 1, A and B; and Fig. S1). Chromosome segregation was considered complete only when bridges were no longer detectable, while fragmented or discontinuous bridges were still classified as unresolved since stretched DNA could result in weak or absent nucleosome signals. In WT cells, 20% of log-phase cells entered anaphase (detected by nuclear elongation) with RPA foci, a percentage that increased to 40% under replication stress induced by HU and was further increased to 60% in the absence of Srs2 (Fig. 1 C). In WT cells, RPA foci were primarily confined to the main nuclear masses, whereas in srs2Δ cells, over 20% of anaphase cells showed RPA foci located within chromatin bridges, regardless of prior HU exposure (Fig. 1, A and D). In cells with anaphase RPA foci, regardless of their subcellular location, chromatin bridges persisted longer than in cells without RPA foci, consistent with previous observations in unchallenged WT cells (Ivanova et al., 2020) (Fig. 1 E). The longest lived bridges were observed in HU-treated srs2Δ cells with anaphase RPA foci, indicating a severe delay in bridge resolution in the absence of Srs2. These findings suggest that the increased RPA on mitotic chromatin in srs2Δ cells, particularly following replication stress, may result from an accumulation of unresolved RPA-coated DNA intermediates formed during the S phase. This could hinder the rapid resolution of chromatin bridges during anaphase.

### Srs2 is required for delaying abscission in the presence of HU-dependent chromatin bridges

To assess the impact of Srs2 loss in abscission timing, we used time-lapse confocal microscopy to monitor the ingression and resolution of the plasma membrane at the abscission site. For this, we used the fluorescent reporter GFP-CAAX, in which GFP is fused to the membrane-targeting CAAX motif of Ras2. The spindle pole protein Spc42 fused to GFP was used to monitor spindle elongation, which marks the start of anaphase. As previously described (Amaral et al., 2016), the morphology of membranes labeled with GFP-CAAX allows visualization of abscission progression: following anaphase onset, the actomyosin ring begins to constrict the bud neck, leading to membrane ingression and eventually separation. The separation of the mother and daughter membranes marks abscission (Fig. 2 A). The relative timing of late mitotic events, including anaphase onset, nuclear division, membrane ingression, and abscission, is summarized in Fig. S2 A.

In unchallenged WT cells, chromatin bridges resolve before membrane ingression, whereas in HU-treated cells, bridges resolve around the time of cytokinesis, concurrently with or shortly after membrane ingression (Fig. S2 B). In WT cells, the

Figure 1. **Srs2 promotes chromatin bridge resolution. (A)** Chromosome segregation (Htb2-mCherry) and RPA foci formation (Rfa2-GFP) in cells of the indicated strains with and without previous exposure to HU. Selected frames are shown, and images are acquired every 2 min (all frames are shown in Fig. S1).

Only cells with RPA foci are shown. Yellow arrows indicate chromatin bridges, green arrows point toward RPA foci, white arrowheads mark RPA bridges, and asterisks note bridge resolution. Scale bar: 5 µm. **(B)** Time of bridge resolution (Htb2-mCherry) from the time of anaphase onset, defined by the rapid elongation of the nucleus. n = number of cells pooled from two independent experiments with similar results. **(C)** Fraction of cells in B undergoing chromosome segregation with RPA foci. **(D)** Fraction of cells in B, which segregated with RPA foci and had RPA-coated chromatin bridge. **(E)** Same data as in B but distinguishing between cells containing vs. not containing RPA anaphase foci. The P values correspond to Dunn's multiple comparison test (B), Fisher's exact test (C and D), and Mann–Whitney test (E).

median time from membrane ingression to abscission is 16 min, but this is significantly delayed to 24 min upon HU treatment (Fig. 2, B and C). This aligns with our previous findings indicating that DNA replication stress delays bridge resolution and abscission through the NoCut checkpoint (Amaral et al., 2016). However, in the presence of HU-induced chromatin bridges, srs2Δ cells exhibited accelerated membrane ingression and failed to delay abscission (Fig. 2, B and C; and Fig. S2 B). These findings indicate that Srs2 is crucial for inhibiting cytokinetic abscission under conditions of DNA replication stress.

The DNA double-strand break repair protein Mre11 forms nuclear foci indicative of DNA damage. In NoCut-deficient cells with chromatin bridges, Mre11-GFP foci accumulate after cytokinesis, likely due to chromatin bridge breakage (Amaral et al., 2016). To determine whether the lack of coordination between chromosome segregation and abscission in Srs2-deficient cells results in DNA damage after cytokinesis, we visualized the formation of DNA damage foci using Mre11-GFP. In normally cycling cells, ∼14% of WT cells exhibit Mre11 foci after nuclear division, with this fraction increasing to 25% in the absence of SRS2. To test whether increased damage in srs2Δ cells is due to replication stress–induced lesions in the main nucleus or due to cytokinesis-induced damage of chromatin bridges, abscission was delayed by deletion of the cytokinesis gene CYK3, which delays septum formation and subsequent abscission (Amaral et al., 2016; Onishi et al., 2013). Notably, the fraction of srs2Δ cells with DNA damage was reduced to near-WT levels in srs2Δ cyk3Δ cells (Fig. 2 D, "untreated"). This suggests that premature cytokinesis causes damage to chromatin bridges in Srs2-deficient cells. Following replication stress induced by HU, the fraction of WT cells exhibiting Mre11 foci was increased from 24% in WT to 40% in srs2Δ cells; also in this case, delayed abscission in CYK3-deficient cells was able to reduce the fraction of cells with Mre11 foci (Fig. 2 D, "HU pulse"). These results demonstrate that Srs2 is required to delay cytokinesis and prevent chromatin bridge damage in response to replication stress.

### Srs2 delays abscission in the presence of catenated chromatin bridges

Next, we investigated whether Srs2 is involved in the abscission delay caused by catenated chromatin bridges by inactivating topoisomerase II (Topo II) using the temperature-sensitive allele top2-4. Cells were grown at 25°C and synchronized in G1 with α-factor for 2 h. G1-arrested cells were then released into pre-heated medium at 37°C and placed in a preheated imaging chamber at the same temperature. At the restrictive temperature (37°C), these mutants exhibit chromatin bridges in all anaphases and delay abscission in an Aurora B–dependent manner (Amaral et al., 2016).

Notably, at 37°C, a high proportion of WT anaphase cells showed RPA foci (60%), with 40% of those displaying foci on chromatin bridges, likely reflecting heat-induced replication stress. At this temperature, neither TOP2 inactivation nor SRS2 deletion significantly increased the overall frequency of RPA foci in anaphase. SRS2 deletion increased the fraction of cells with RPA foci on chromatin bridges in otherwise WT cells (from 40% to 60%) but not in a top2-4 background (Fig. S2 C). These findings suggest that Srs2 does not play a major role in RPA removal from catenated chromatin at 37°C.

Imaging of the actomyosin ring component Myo1 tagged with GFP (Myo1-GFP) confirmed that chromatin bridge resolution in top2-4 cells (visualized through Htb2-mCherry) occurred after actomyosin ring contraction, and was unaffected by the presence or absence of Srs2 (Fig. 3, A and B). WT and srs2Δ cells expressing the GFP-CAAX plasma membrane marker exhibited similar abscission timing (Fig. 3, C and D). As reported previously (Amaral et al., 2016), inactivation of top2-4 resulted in significant impairment in membrane resolution, with 92% of cells failing to complete abscission within 60 min. Remarkably, this impairment was partially but significantly rescued in cells lacking Srs2, reducing the failure rate from 92% to 41% (Fig. 3, C and D). Together, these results suggest that Srs2 plays a critical role in inhibiting abscission in cells challenged with both HU-induced and catenated chromatin bridges.

To further investigate whether Srs2 functions through the canonical Aurora B–dependent NoCut pathway, we analyzed abscission timing in top2-4 cells lacking SRS2, carrying a functional inactivation of Aurora B (via the temperature-sensitive ipl1-321 allele, which inactivates the yeast Aurora B homolog Ipl1), or both. Live-cell imaging of the GFP-CAAX plasma membrane marker revealed that simultaneous inactivation of Srs2 and Ipl1 rescued the abscission delay to the same extent as ipl1-321 alone (Fig. 3, E and F). These findings suggest that Srs2 does not further delay abscission in the absence of Aurora B activity, supporting the conclusion that Srs2 and Ipl1 act in the same pathway.

### Association of Srs2 with PCNA is required for full abscission inhibition

Srs2 is recruited to the replication fork during the S phase by SUMOylated PCNA, through its C-terminal SUMO-interacting motif (SIM) and PCNA-interacting protein box (PIP box) (Armstrong et al., 2012; Kolesar et al., 2012) (Fig. 4 A). To investigate whether this interaction is essential for the role of Srs2 in NoCut function, we first deleted the SIM of Srs2 and assessed abscission timing. Interestingly, deletion of the SIM alone was not sufficient to rescue abscission failure in top2-4 mutants (Fig. 4 B). Although SIM-defective Srs2 has a lower affinity to

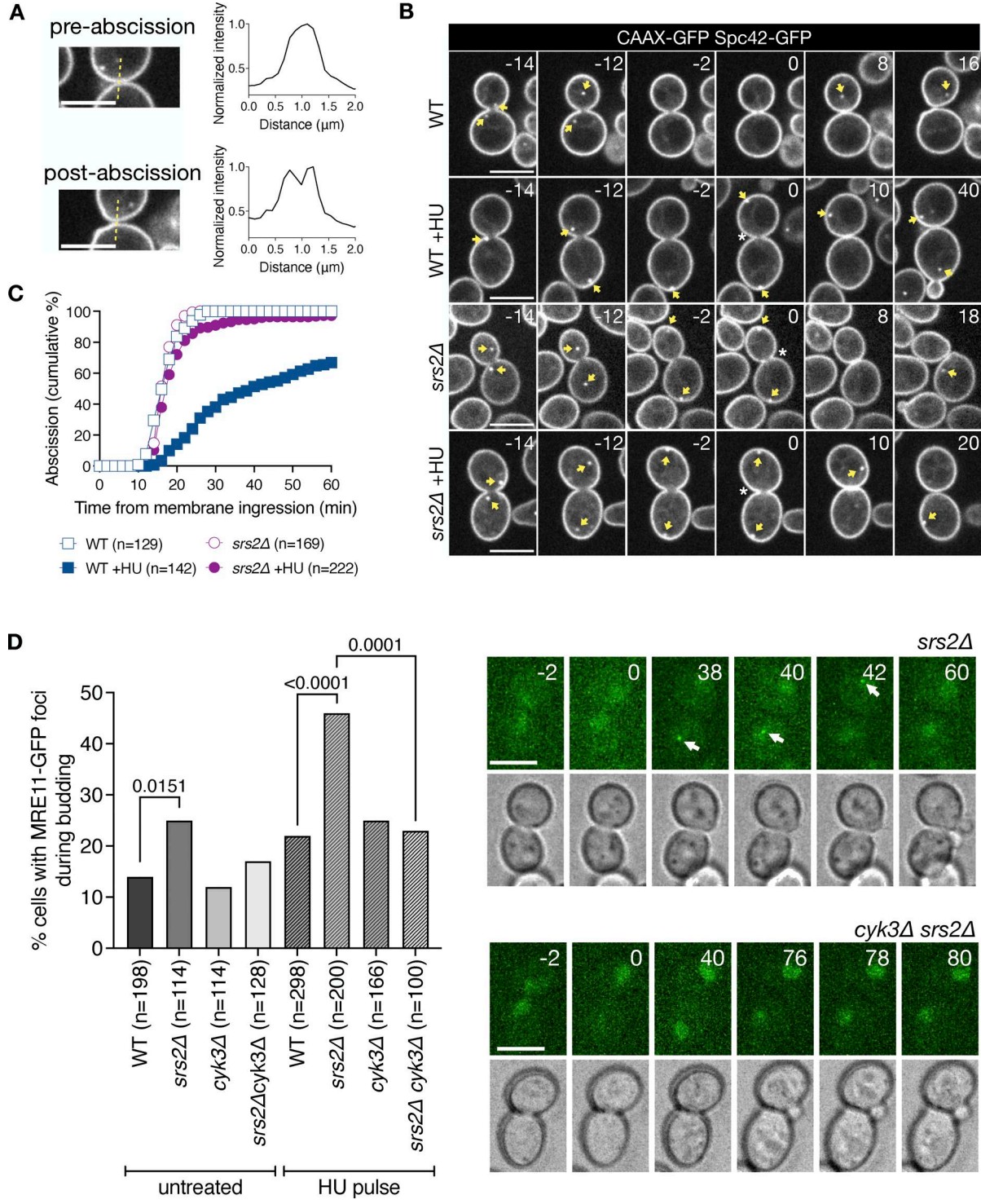

Figure 2. **Srs2 prevents abscission-dependent DNA damage following replication stress. (A)** Abscission is scored by measuring the distribution of GFP-CAAX intensity values across the mother–daughter cell axis at the bud neck. Upper panel: Cells that have yet to abscise show a single peak of intensity at the bud neck. Lower panel: Membrane separation is marked by two distinct peaks in GFP intensity. **(B)** Membrane ingression and abscission in WT and *srs2Δ* cells with or without an HU pulse. Cells express the plasma membrane marker CAAX-GFP and the spindle pole marker Spc42-GFP (yellow arrows). Time is indicated in minutes; 0 min indicates the start of membrane ingression. Asterisk specifies membrane ingression. White arrowhead marks complete constriction of the plasma membrane (abscission). Single Z-slices are shown, but 12 Z-slices spaced 0.3 µm, spanning the whole cell, were used for image analysis. **(C)** Fraction of cells that complete abscission from the time of membrane ingression. *n* = number of cells pooled from *N* = 3 independent experiments with similar results. WT vs. WT +HU, and WT +HU vs. *srs2Δ* +HU, P < 0.0001, Mann–Whitney test. **(D)** Frequency of Mre11-GFP focus formation after cytokinesis in the indicated strains and conditions. The P values correspond to Fisher's exact test. Representative images of cells expressing Mre11-GFP, after cytokinesis following a HU pulse. The arrow points to a nuclear Mre11 focus. Scale bars in A, B, and D: 5 µm.

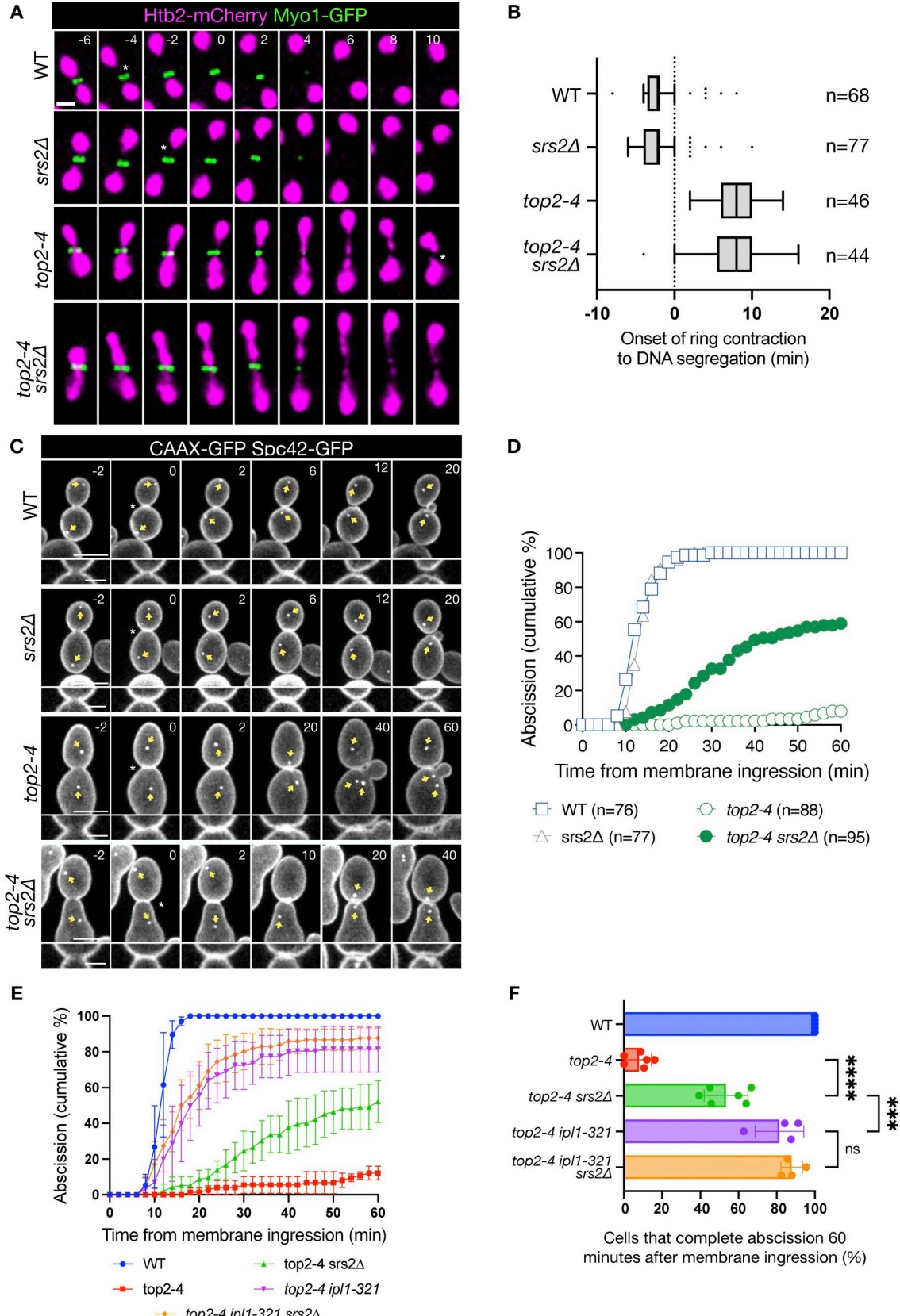

Figure 3. **Srs2 promotes abscission inhibition in Topo II–deficient cells. (A)** Time-lapse images of representative cells of the indicated genotypes expressing Htb2-mCherry (magenta) and Myo1-GFP (green) to visualize chromatin and the actomyosin ring, respectively. Numbers indicate time (min) relative to

the start of ring contraction (t = 0). Scale bar: 2 μm. **(B)** Box plot showing the time from onset of contractile ring constriction to chromatin bridge segregation in the indicated strains. *n* = number of cells pooled from two independent experiments. Box indicates median and interquartile range, whiskers are calculated with the Tukey method, and outliers are shown as dots. **(C)** Membrane ingression and abscission in cells of the indicated genotypes. Time in minutes, 0 min indicates the start of membrane ingression. Yellow arrow marks the position of spindle poles. Asterisk specifies membrane ingression. White arrowhead marks complete separation of the plasma membrane. Time is indicated in minutes; 0 min indicates the start of membrane ingression. Asterisk specifies membrane ingression. White arrowhead marks complete constriction of the plasma membrane (abscission). Single Z-slices are shown, but 12 Z-slices spaced 0.3 μm, spanning the whole cell, were used for image analysis. Lower zoom-in panel shows the dynamics of membrane ingression at the central Z-slice. Scale bar: 5 μm; inset, 2 μm. **(D)** Fraction of cells that complete abscission from the time of membrane ingression. *n* = cells pooled from *N* = 3 independent experiments with similar results. WT vs. *top2-4,* and *top2-4* vs. *top2-4 srs2Δ,* P < 0.0001, Mann–Whitney test. **(E)** Time course of abscission completion in cells of the indicated genotypes, scored as in D. Data represent the mean ± SD from four independent biological replicates, with 20–30 cells per genotype per replicate. **(F)** Fraction of cells that completed abscission within 60 min of membrane ingression. Bars represent the mean ± SD from four to six biological replicates (dots), with 20–30 cells per replicate. Statistical analysis was performed using Tukey's multiple comparisons test; ****P < 0.0001. Ns, nonsignificant, P > 0.05.

SUMOylated PCNA, it can still interact with non-SUMOylated PCNA (Armstrong et al., 2012). We then deleted both the PIP box and the SIM, thereby disrupting both interactions with SUMO and PCNA. The absence of both the SIM and PIP box partially but significantly rescued the fraction of cells able to complete abscission, increasing it to 28% (Fig. 4 B). Deletion of the PIP box alone, however, was not sufficient to bypass abscission inhibition in *top2-4* cells, indicating that the SIM and PIP box cooperate to mediate Srs2 function in this context (Fig. 4 B). Finally, live-cell imaging of Myo1-GFP and Htb2-mCherry confirmed that chromatin bridge segregation occurred after actomyosin ring contraction in all mutants analyzed (Fig. 4 C). We conclude that interacting with PCNA is critical for Srs2 to fully promote a NoCut-mediated abscission delay in the presence of catenated bridges.

To directly test whether Srs2's helicase activity is required for this function, we examined the behavior of the *srs2-R337S* (also referred to as *srs2-R3*) mutant, which is defective in helicase activity but retains PCNA-binding ability (Le Breton et al., 2008). Remarkably, this mutant fully supported abscission inhibition in *top2-4* cells (Fig. 4 D). These results indicate that Srs2's helicase activity is dispensable for abscission delay, and support the model that its association with PCNA, rather than catalytic activity, is essential for NoCut checkpoint function.

To determine whether PCNA is required during cytokinesis to modulate abscission timing, we used a cold-sensitive mutant of PCNA (*pol30-S115P*). Due to weakened interactions within the Pol30 homotrimer, this mutant shows reduced chromatin accumulation and does not grow at 14°C (Ayyagari et al., 1995; Johnson et al., 2016). WT and *pol30-S115P* cells were grown at the permissive *pol30-S115P* temperature of 30°C and exposed to HU for 2 h, before shifting to the restrictive temperature of 14°C. Following HU exposure, *pol30* mutants were significantly less delayed in abscission than WT cells (Fig. 4 E). Pol30 inactivation in otherwise unchallenged cells (without HU) did not affect abscission timing. Together, these results further suggest that PCNA integrity, and likely its association with Srs2 on chromatin, is important for the NoCut checkpoint.

### Deletion of the PCNA unloader Elg1 is sufficient for abscission inhibition in response to dicentric bridges

We next tested whether association of PCNA with chromatin is not only necessary but also sufficient to inhibit abscission. We took advantage of a conditional dicentric chromosome,

generated by end-to-end fusion of chromosomes IV and XII, in which centromere 4 is kept inactive thanks to activation of the strong *GAL1,10* promoter. On activation of *CEN4* in glucose-containing media, chromatin bridges (visualized with Htb2-mCherry) form in 50% of anaphases (Fig. 5 A). These bridges do not trigger a NoCut response, possibly due to their failure to recruit a NoCut-activating factor (Amaral et al., 2016). We asked whether retaining PCNA and Srs2 on chromatin could render dicentric chromatin bridges detectable by the NoCut checkpoint. After replication termination, PCNA is unloaded from chromatin by the replication factor C-like complex Elg1. In cells lacking Elg1, PCNA remains associated with chromatin (Kubota et al., 2013).

We monitored abscission in conditionally dicentric mutants with WT *ELG1* and confirmed that abscission proceeds with near-WT efficiency, regardless of the presence of dicentric bridge (Fig. 5 B, *ELG1*). Similarly, *elg1Δ* cells without chromatin bridges completed abscission with dynamics comparable to WT cells (Fig. 5 B, *elg1Δ,* "no bridge"), indicating that PCNA retention on chromatin alone does not affect abscission timing. However, in the presence of a dicentric bridge, *elg1Δ* mutants exhibited delayed abscission, with ~50% of cells unable to resolve the membrane within 60 min (Fig. 5 B, *elg1Δ,* "with bridge"). This finding suggests that PCNA retention, or the persistence of PCNA-associated factors, specifically on dicentric chromatin bridges is necessary to inhibit abscission, rather than PCNA retention on chromatin in general.

### The Srs2 human homolog PARI delays midbody severing in response to chromatin bridges

Like Srs2, the human protein PARI acts as an anti-recombinase and prevents the accumulation of recombination intermediates that could lead to genomic instability during DNA replication (Burkovics et al., 2016; Moldovan et al., 2012; Mochizuki et al., 2017). To investigate the role of PARI in the human NoCut/abscission checkpoint, we developed an assay to measure chromatin bridge formation and abscission-associated events in HeLa cells. Cells were synchronized in the cell cycle using the double thymidine method. Cells were treated with thymidine for 16 h to inhibit deoxynucleotide synthesis, causing arrest in the S phase. This was followed by an 8-h release into normal culture media and a subsequent 16-h thymidine treatment to achieve a more synchronized block at the G1/S boundary. Following release from the double thymidine block, cells progressed to G2/

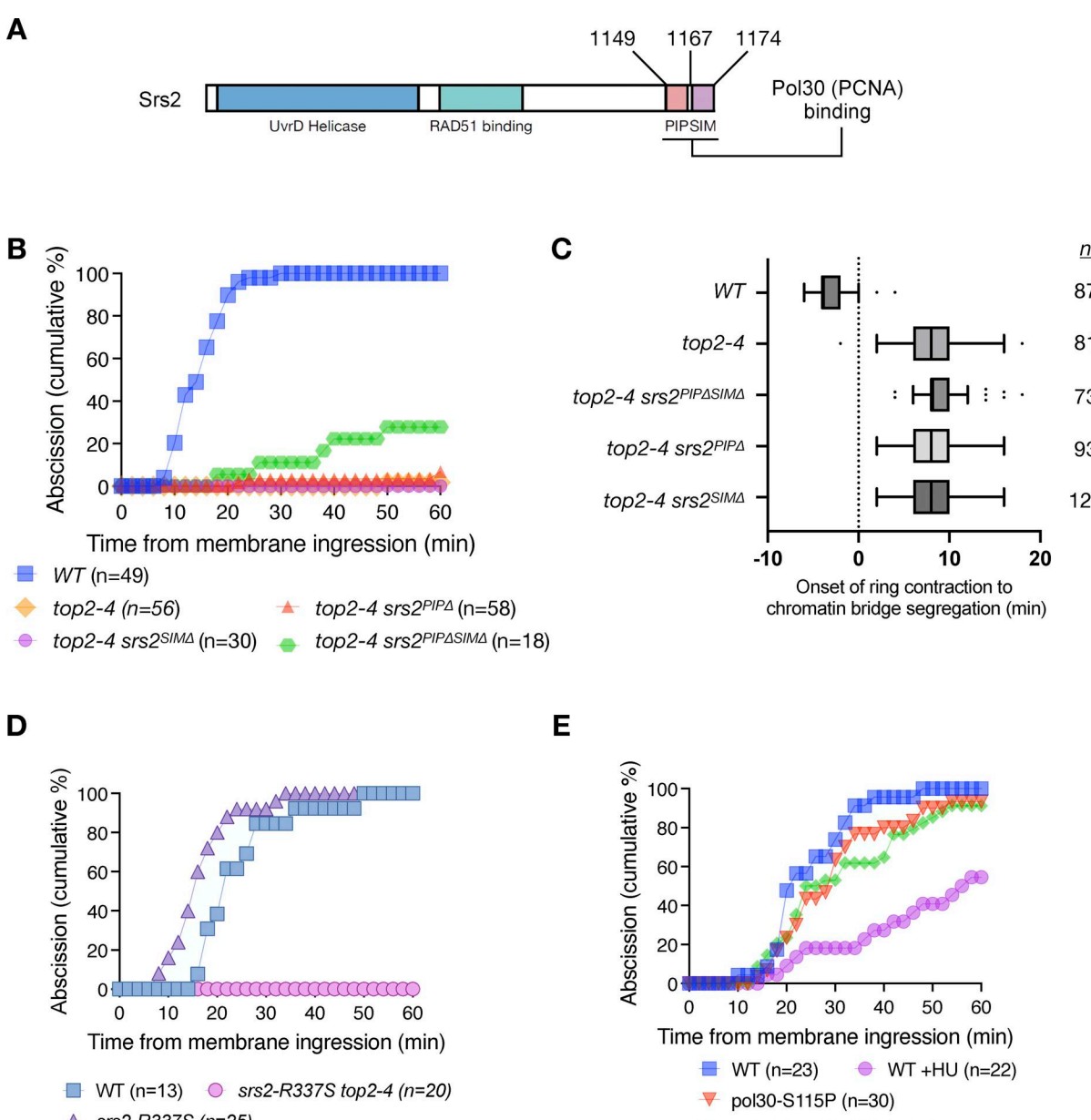

Figure 4. **Chromatin association between Srs2 and PCNA promotes inhibition of abscission in response to chromatin bridges. (A)** Schematic representation of the Srs2 protein highlighting functional domains. The PIP and SIM regions are shown with their amino acid positions indicated. **(B)** Abscission dynamics of srs2 PIP and SIM mutants in the presence of catenated bridges. WT vs. *top2-4*, P < 0.0001; *top2-4* vs. *top2-4 srs2^SIMΔ* and *top2-4 srs2^PIPΔ*, P > 0.05; *top2-4* vs. *top2-4 srs2^PIPΔSIMΔ*, P < 0.0001, Mann–Whitney test. **(C)** Box plot showing the time from onset of contractile ring constriction to chromatin bridge segregation in the indicated strains. *n* = number of cells pooled from two independent experiments. Box indicates the median and interquartile range, whiskers are calculated with the Tukey method, and outliers are shown as dots. **(D)** Abscission dynamics of *srs2-R337S* (helicase-defective) in Top2-defective cells. WT vs. *top2-4 srs2-R337S*, <0.0001, Mann–Whitney test. **(E)** Inactivation of cold-sensitive PCNA (*pol30-S115*) in cells exposed to DNA replication stress. WT+HU vs. *pol30-S11P*+HU, P = 0.0004, Mann–Whitney test.

mitosis after ~7.5 h and entered cytokinesis 4.5 h later (Fig. S3 A). To induce chromatin bridges with minimal S-phase perturbation, a low dose (250 nM) of the catalytic Topo II inhibitor ICRF-193, which causes both catenated chromatin bridges and ultrafine bridges (Germann et al., 2013; Bhowmick et al., 2019), was added during the late G2 phase (Fig. S3 A).

PARI was depleted in HeLa cells using siRNA. Since neither commercial nor in-house polyclonal antibodies detected the PARI protein at the expected 65 kDa size, RT-qPCR confirmed an 80% reduction in PARI mRNA levels compared to cells transfected with control siRNA (Fig. S3 B).

Next, we evaluated cytokinesis progression by measuring midbody stability. The duration from midbody assembly (marked by the constriction of central spindle microtubules into a bundle) to midbody severing is known to increase in the presence of chromatin bridges and has been used as a proxy for abscission timing (Steigemann et al., 2009; Carlton et al., 2012; Advedissian et al., 2024). We term this duration "midbody lifetime." To assess

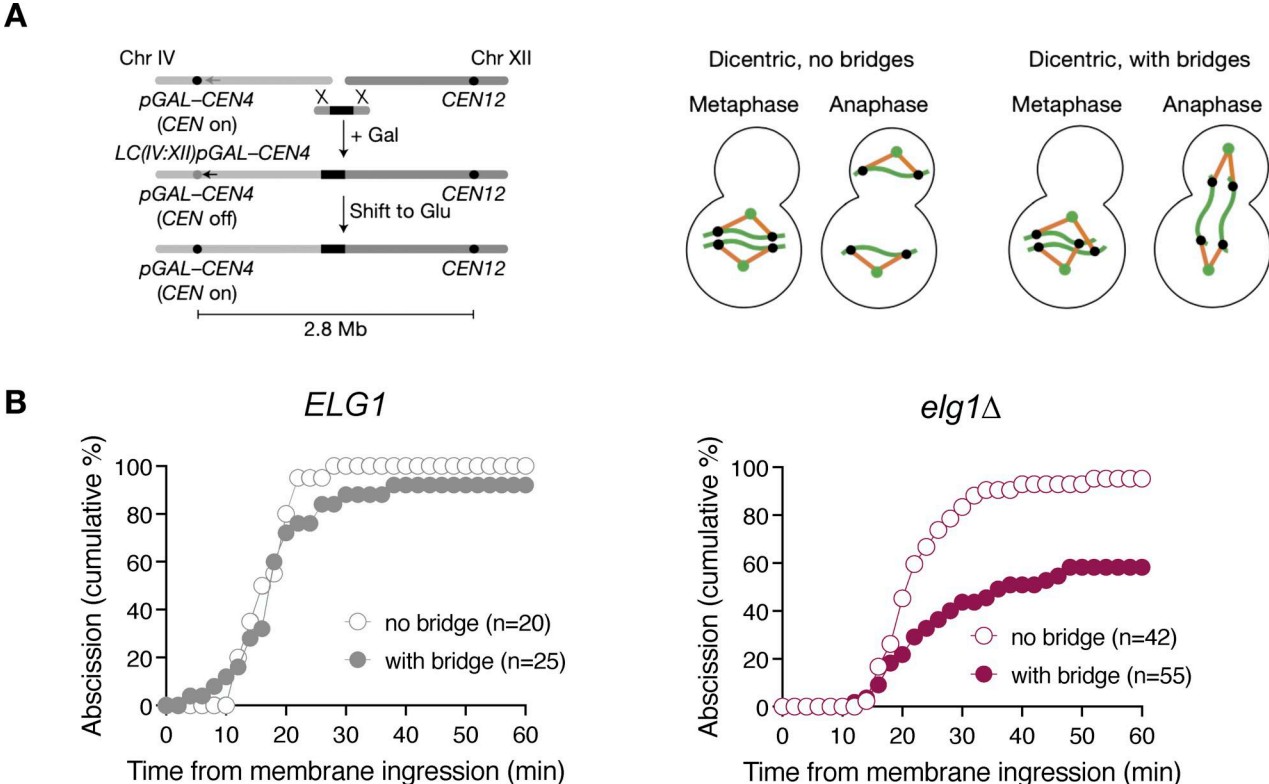

**Figure 5. Deletion of the PCNA unloader Elg1 is sufficient for abscission inhibition in response to dicentric bridges. (A)** Schematic illustrating the generation of the conditional dicentric strain (left) and the formation of chromosome bridges upon biorientation of kinetochores (right). Metaphase and anaphase cells with or without dicentric bridges are shown on the right. Green dots: spindle poles; black dots: kinetochores; orange lines: kinetochore microtubules; green lines: dicentric chromosomes. **(B)** Abscission dynamics in dicentric strains of the indicated genotypes. Dicentric bridges were imaged with Htb2-mCherry, and abscission was scored with GFP-CAAX. *ELG1* cells with bridge vs. *elg1Δ* with bridge, P < 0.0001, Mann–Whitney test. *n* = number of cells pooled from two (B, *ELG1*) or three (B, *elg1Δ*) independent time-lapse experiments.

midbody lifetime, we conducted time-lapse spinning disk microscopy on synchronized HeLa cells stably expressing GFP-α-tubulin. Chromatin was visualized using H2B-mCherry.

Our results demonstrated that in untreated HeLa cells, the average midbody lifetime was 130 min. However, in the presence of ICRF-193, which caused the formation of anaphase chromatin bridges, as determined by H2B-mCherry, midbody severing was significantly delayed to ~200 min (Fig. 6, A–C; Videos 1, 2, and 3). Remarkably, in the absence of PARI, this delayed midbody severing was partially but significantly reduced, to around 150 min (Fig. 6, A and C; and Video 4). The severing of midbody microtubules in PARI-depleted cells occurred in the presence of chromatin bridges (Fig. 6, A and B) and was confirmed using correlative light and electron microscopy (CLEM) (Fig. 6 D and Fig. S4). Chromatin was observed traversing the ICB, passing through the microtubule-dense midbody and Flemming body in cells treated with ICRF-193. The microtubules of the midbody remained straight and ordered along the ICB, even in the presence of a chromatin bridge. Furthermore, depletion of PARI did not appear to disrupt the structure of the ICB during cytokinesis, with morphologies similar to those observed in cells with intact PARI (Fig. 6 D).

To determine whether the role of PARI in regulating midbody lifetime is specific and not due to off-target effects of the siRNA

used, we designed a siPARI-1–resistant version of PARI. The putative seed region of siPARI-1 was identified; three silent mutations were introduced that rendered the mRNA resistant to degradation by siPARI-1 (Fig. S5, A and B). Additionally, an eGFP tag was added to the N terminus of PARI. We generated HeLa Kyoto cell lines stably expressing either eGFP or siPARI-1–resistant eGFP-PARI (eGFP-PARI$^R$) (see Materials and methods). These cells were depleted of endogenous PARI using siPARI-1 and treated with ICRF-193 as described previously. The midbody and microtubules were visualized using SiR-tubulin dye, and midbody lifetime was measured from assembly to severing or, if this could not be detected due to the weaker signal compared with GFP-tubulin, from midbody assembly to disassembly. In cells expressing eGFP, siPARI-1 treatment significantly reduced midbody lifetime, consistent with previous observations in control HeLa cells. However, in cells expressing eGFP-PARI$^R$, siPARI-1 did not reduce midbody lifetime (Fig. S5, C and D). Together, these results suggest that the human homolog of Srs2, PARI, specifically delays midbody severing in the presence of chromatin bridges induced by Topo II inactivation.

**PARI depletion stabilizes chromatin bridges during interphase but does not lead to cytokinesis failure**

To determine the consequences of PARI depletion on bridge stability and cell division, we performed time-lapse imaging of

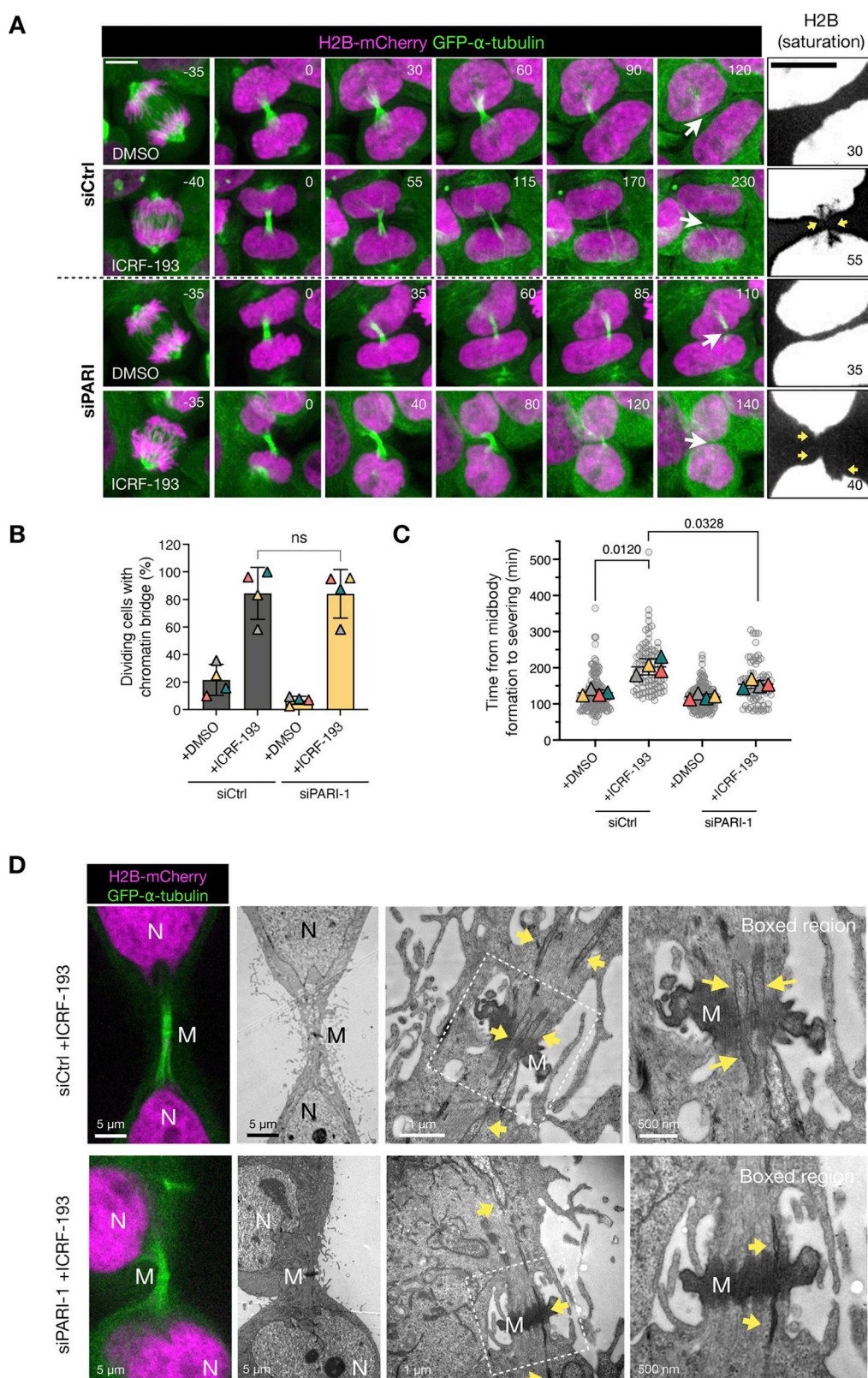

Figure 6. **PARI is required to delay midbody severing in cells with catenated bridges. (A)** Time-lapse microscopy of HeLa cells expressing H2B-mCherry (magenta) and GFP-α-tubulin (green), treated with either DMSO or ICRF-193 to induce chromatin bridges, and transfected with siCtrl or siPARI-1. Selected frames show progression from metaphase to midbody severing. Arrows indicate chromatin bridges. Insets (rightmost column) show saturated H2B signal to highlight persistent bridges. Asterisks mark chromatin bridges in the ICRF-193 condition. Time is indicated in minutes relative to midbody assembly (t = 0), defined as the time point at which the Flemming body becomes well defined and the midbody arms display bundled microtubules. Movies corresponding to the cells shown are provided as Videos 1, 2, 3, and 4. Scale bar: 5 µm. **(B)** Quantification of the percentage of dividing cells exhibiting chromatin bridges under each

condition. Data are the mean ± SD from four independent experiments. ns = nonsignificant. **(C)** SuperPlot showing time from midbody formation to severing for individual cells. Each circle represents one cell, pooled from four independent experiments; triangles represent the mean of each experiment; horizontal lines indicate means. The P-values from Mann–Whitney tests are indicated. **(D)** CLEM of cells expressing H2B-mCherry and GFP-α-tubulin after ICRF-193 treatment. Confocal images (left panels) correspond to EM fields shown on the right. Yellow arrows highlight chromatin fibers traversing the midbody (M) between nuclei (N).

HeLa cells treated as in Fig. 6, stained with SiR-tubulin and expressing eGFP fused to barrier-to-autointegration factor (eGFP-BAF), a sensitive reporter for chromatin bridges whose signal is not compromised by stretching of the bridge (Umbreit et al., 2020) (Fig. 7 A, Videos 5, 6, and 7). In agreement with previous reports, spontaneous chromatin bridges were observed in ~20% of control (siCtrl) cells, indicating a baseline level of chromatin entanglement in HeLa cells. This frequency was unaffected by PARI depletion, suggesting that PARI is not involved in the formation of spontaneous bridges. Treatment with ICRF-193 induced bridge formation in nearly all dividing cells, confirming widespread catenation of sister chromatids, and this was also unaffected by PARI knockdown (Fig. 7, A and B).

We next measured the midbody lifetime. In siCtrl cells, the presence of spontaneous bridges led to a significant increase in midbody lifetime. This delay was absent in PARI-depleted cells, indicating that PARI is required to stabilize midbodies in response to spontaneous bridges (Fig. 7 C). Similarly, ICRF-193–induced bridges resulted in prolonged midbody persistence in control cells, and this delay was abolished upon PARI knockdown (Fig. 7 D). These results are consistent with our previous observations using H2B-mCherry and GFP-tubulin, and further confirm that PARI promotes stabilization of the midbody in response to chromatin bridges.

To follow the fate of chromatin bridges beyond midbody disassembly, we tracked the time to bridge resolution during the subsequent interphase. As shown in Fig. 7 E, most spontaneous bridges eventually resolved, though this resolution occurred with delayed kinetics in PARI-depleted cells, suggesting that PARI facilitates interphase bridge resolution. In contrast, bridges induced by ICRF-193 rarely resolved, and PARI knockdown further impaired their resolution. These observations suggest that PARI and Topo II function in parallel or distinct pathways of bridge resolution, and that loss of PARI exacerbates the persistence of catenated DNA.

Finally, we evaluated the frequency of binucleation as a readout of cytokinesis failure (Fig. 7 F). In DMSO-treated cells, binucleation occurred in ~10% of cells with spontaneous bridges, with a modest, nonsignificant increase upon PARI depletion. ICRF-193 treatment caused a marked increase in binucleated cells (~50%), independent of the PARI status. These results suggest that although PARI is required for bridge-dependent midbody stabilization, its depletion does not promote furrow regression and cytokinesis failure. This outcome contrasts with the effects of inhibiting core abscission checkpoint components such as Aurora B or CHMP4C, whose inactivation results in furrow regression or bridge breakage, respectively. This raises the possibility that PARI plays a supportive but nonessential role in the abscission checkpoint. Alternatively, incomplete depletion of PARI may allow residual checkpoint activity.

## PARI does not stabilize the midbody upon nuclear pore complex defects

It has been shown that defects in nuclear pore assembly, such as those caused by depleting the basket component NUP153, can delay midbody severing without the presence of chromatin bridges (Mackay et al., 2010). To determine whether PARI is involved in this response, we partially codepleted NUP153 along with PARI in HeLa cells using siRNA (see Materials and methods). The NUP153 protein level was assessed by western blotting and was found to be partially depleted after transfection; codepletion with PARI did not alter these levels (Fig. 8 A). Cells were synchronized in G1/S with thymidine for 24 h and then released for 16 h. NUP153-depleted cells distinctively arrested in cytokinesis, exhibiting cytoplasmic "abscission checkpoint bodies", which are indicative of an active abscission checkpoint (Strohacker et al., 2021) in cells treated with both control and PARI-targeting siRNAs (Fig. 8, B and C). These results suggest that PARI is not required for midbody stabilization in response to nuclear pore assembly defects, but rather plays a specific role in delaying midbody severing in the presence of chromatin bridges.

## PARI promotes actin stabilization at the ICB

F-actin has been shown to promote abscission checkpoint–mediated delay by accumulating at the ICB as actin patches, which stabilize the canal in the presence of chromatin bridges (Steigemann et al., 2009; Dandoulaki et al., 2018; Bai et al., 2020). To determine whether PARI regulates this process, we examined F-actin dynamics in HeLa cells stably expressing H2B-mCherry and GFP-actin.

As shown in Fig. 9 A and Video 8, in control cells (siCtrl + DMSO) GFP-actin accumulated at the cleavage furrow and was fully dispersed ~60 min after anaphase onset. ICRF-193 treatment (siCtrl + ICRF-193) triggered bright, highly dynamic actin clusters that frequently co-localized with H2B-mCherry bridges and persisted for several hours (Fig. 9 A and Video 9). PARI depletion by itself (siPARI + DMSO) did not alter the normal actin clearance pattern, but it visibly reduced the longevity of ICRF-induced clusters (siPARI + ICRF-193) (Fig. 9 A; and Videos 10 and 11). To capture actin dynamics, we calculated an *actin-cluster index* (GFP-actin signal in clusters within the division plane; see Materials and methods for details). Fig. 9 B shows representative traces for five cells per condition (out of ~30 analyzed), showing a low-level actin-cluster peak in controls (corresponding to furrow ingression), prolonged fluctuating plateaus after ICRF-193 (reflecting actin clusters associated with chromatin bridges), and a reduction of that plateau in a fraction of cells when PARI is depleted. Cumulative frequency plots of actin-cluster clearance (Fig. 9 C) show that 50% of control or siPARI cells had cleared actin from the division plane by 60 min, whereas only 25% of ICRF-treated controls had done so even

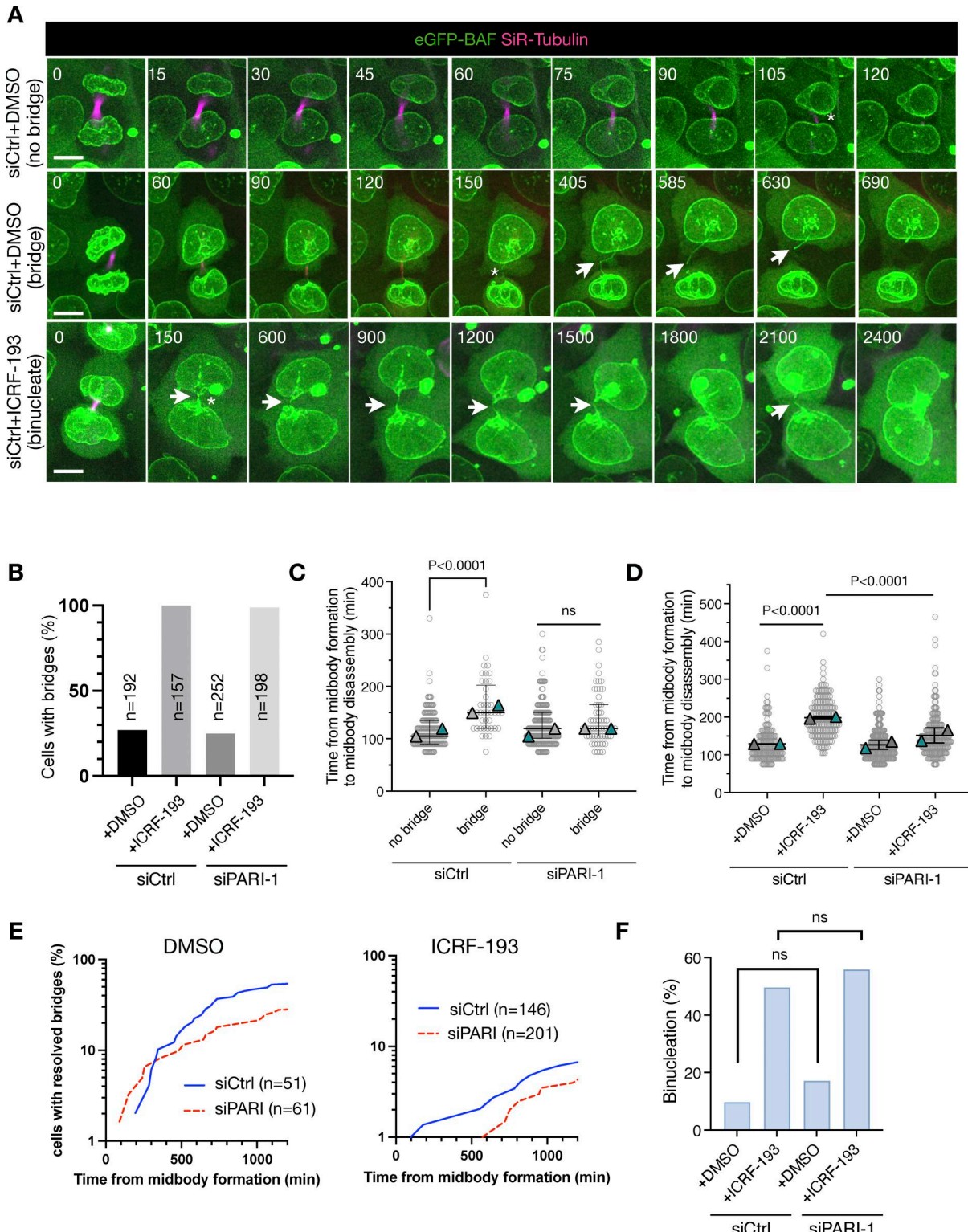

Figure 7. **PARI depletion stabilizes chromatin bridges during interphase but does not lead to cytokinesis failure. (A)** Time-lapse imaging of HeLa cells expressing eGFP-BAF (green) and stained with SiR-tubulin (magenta) to visualize chromatin bridges and midbody microtubules. Three representative divisions are shown: siCtrl+DMSO without bridge (top), siCtrl+DMSO with a spontaneous chromatin bridge (middle), and siCtrl+ICRF-193 with a persistent chromatin bridge (bottom). White arrows indicate chromatin bridges; asterisks indicate midbody disassembly. Time in minutes after midbody formation. Scale bar: 5 µm. Movies corresponding to the cells shown are provided as Videos 5, 6, and 7. **(B)** Quantification of cells with visible chromatin bridges in the indicated conditions. *n* = number of cells pooled from two independent experiments with similar results. **(C and D)** Time from midbody formation to disassembly in cells under the indicated conditions. Each circle represents one cell, pooled from two independent experiments; triangles indicate the mean value of each experiment; bars indicate median ± IQR. ns: not significant; P-values from the Mann–Whitney test. **(E)** Plots showing the time from midbody formation to chromatin bridge resolution for cells with bridges. *n* = number of cells pooled from two independent experiments with similar results. P < 0.0005, Mann–Whitney test for both

DMSO and ICRF-193. **(F)** Frequency of binucleation in cells with chromatin bridges under the indicated conditions. ns, nonsignificant (P > 0.05, Fisher's exact test).

after 800 min. PARI knockdown partially rescued this delay: 70% of siPARI + ICRF-193 cells removed clusters within the same 800-min window. These results suggest that PARI is required to sustain the long-lived F-actin accumulations generated by the abscission checkpoint in response to Topo II inhibition.

### PARI regulates midbody stability through an Aurora B–dependent pathway

To determine whether catenated bridges induce a delay in midbody severing dependent on the Aurora B–mediated abscission checkpoint, and to investigate whether PARI is a component of this checkpoint, we inhibited Aurora B at the time of cytokinesis. HeLa cells stably expressing H2B-mCherry and GFP-α-tubulin were synchronized using the double thymidine block and treated with ICRF-193 following the standard protocol. Based on previous observations, midbody severing typically occurred ~120 min after the start of acquisition. At this point, the Aurora B inhibitor hesperadin was added (Fig. 10, A and B). As expected, control cells treated only with ICRF-193 exhibited a mean midbody lifetime of 200 min, and Aurora B inhibition accelerated midbody disassembly, reducing the mean midbody lifetime to 90 min. In the absence of PARI, the midbody lifetime was similarly reduced, consistent with our previous observations in the presence of catenated bridges. Notably, co-

depletion of PARI and inhibition of Aurora B did not further reduce the already shortened midbody lifetime, suggesting that Aurora B and PARI function within the same pathway to regulate midbody disassembly.

To explore whether this effect involved altered recruitment of key abscission checkpoint regulators to the midbody, we examined the localization of phosphorylated Aurora B (pT232), Topo IIα, and the MRN complex component MRE11. HeLa cells were treated as previously described, fixed 12 h after thymidine release, and stained with antibodies against pT232-Aurora B (marking its active form), Topo IIα, and MRE11. Protein intensity was quantified around the Flemming body. We observed a modest, nonsignificant increase in pT232-Aurora B levels in ICRF-193–treated cells, with no substantial difference upon PARI knockdown (Fig. 10 C). Topo IIα was detected on chromatin in all cells and localized to the midbody region in ICRF-treated cells, likely reflecting its association with unresolved chromatin bridges (Fig. 10 D). Importantly, the ICRF-193 concentration used did not impair Topo IIα recruitment. PARI depletion did not alter Topo IIα localization in nuclei or bridges (Fig. 10 D), consistent with our earlier finding that PARI knockdown does not affect bridge formation. MRE11 was weakly present at midbodies and slightly reduced upon ICRF treatment, consistent with its role in responding to Topo II–dependent DNA breaks (Fig. 10 E).

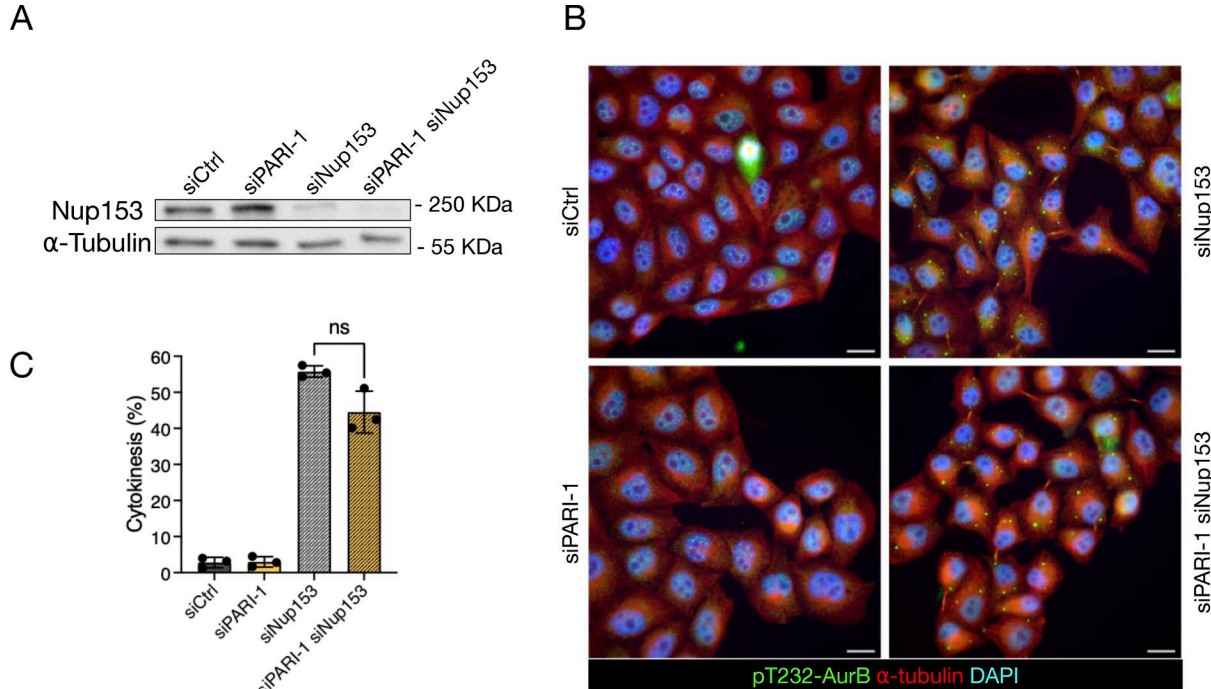

Figure 8. **PARI does not stabilize the midbody upon NPC defects. (A)** Western blot showing partial depletion of NUP153. Membrane probed with a NUP153 antibody and alpha-tubulin as a loading control. **(B)** Cells transfected with siCtrl, siNUP153, siPARI-1, and a combination of siPARI-1 + siNUP153. Cells were synchronized with thymidine for 24 h, released for 16 h, and fixed. Scale bars: 10 μm. **(C)** Quantification of the fraction of cells showing midbodies (% cytokinesis). Mean and SD are shown. Student's paired *t* test, P > 0.05 = ns (*n* > 789, *N* = 3). NPC, nuclear pore complex. Source data are available for this figure: SourceData F8.

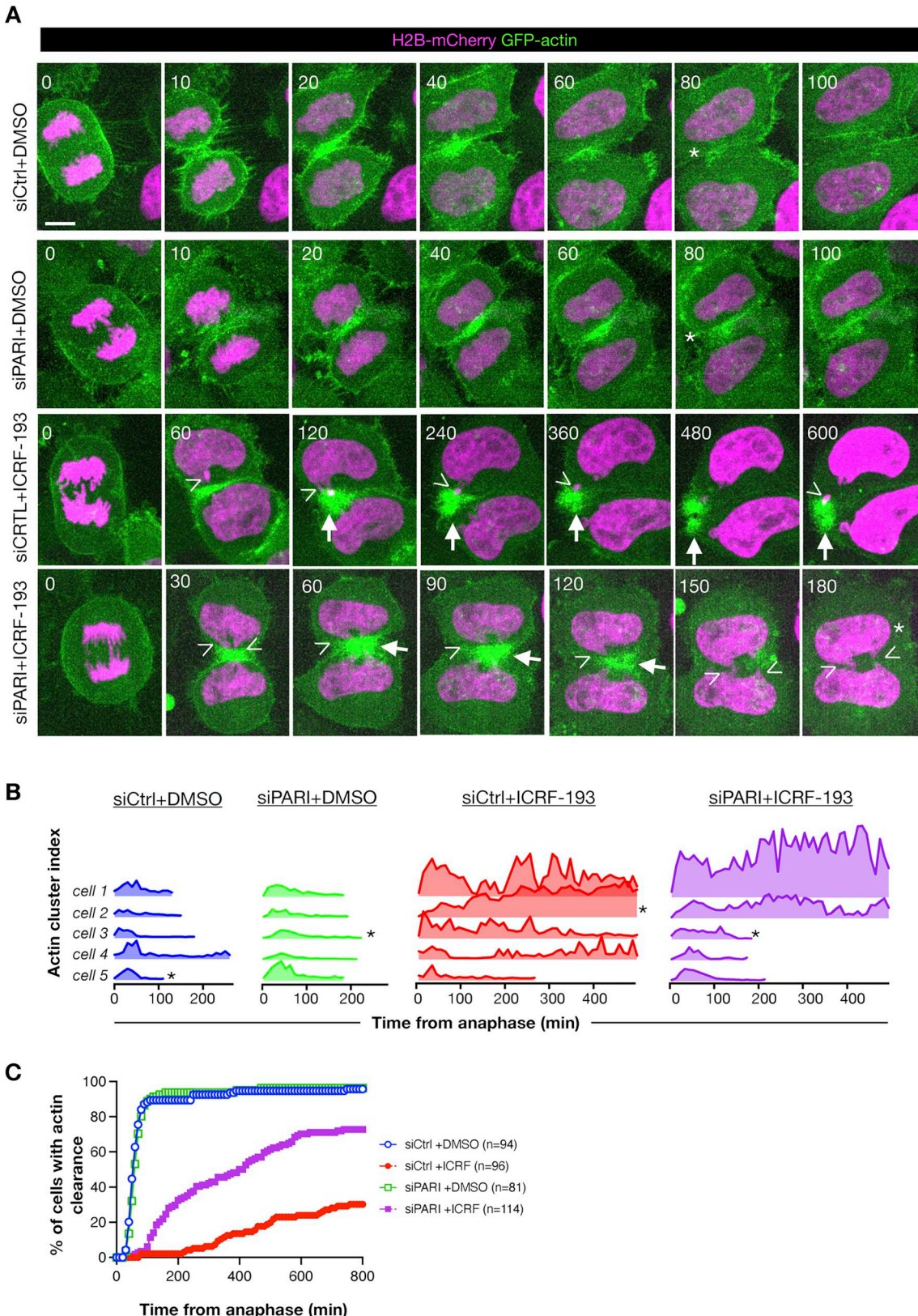

Figure 9. **PARI promotes actin-patch clearance at the intracellular bridge in cells with chromatin bridges. (A)** Time-lapse images of HeLa cells expressing H2B-mCherry (magenta) and GFP-actin (green), treated with either DMSO or ICRF-193, and transfected with siCtrl or siPARI-1. Images show time relative to anaphase onset (t = 0). Scale bar: 5 µm. Arrows point to actin patches, arrowheads point to chromatin bridges, and asterisks mark actin-patch clearance. Movies corresponding to the cells shown are provided as Videos 8, 9, 10, and 11. **(B)** Actin-cluster index plotted for five representative cells per condition. Asterisks mark traces of cells shown in A. **(C)** Population-level analysis showing the cumulative percentage of cells that have cleared actin clusters from the ICB over time after anaphase onset. n = cells pooled from N = 3 independent experiments with similar results. siCtrl +DMSO vs. siCtrl +ICRF-193, P < 0.0001; siCtrl +ICRF-193 vs. siPARI-1 +ICRF-193, P < 0.0001, Mann–Whitney test.

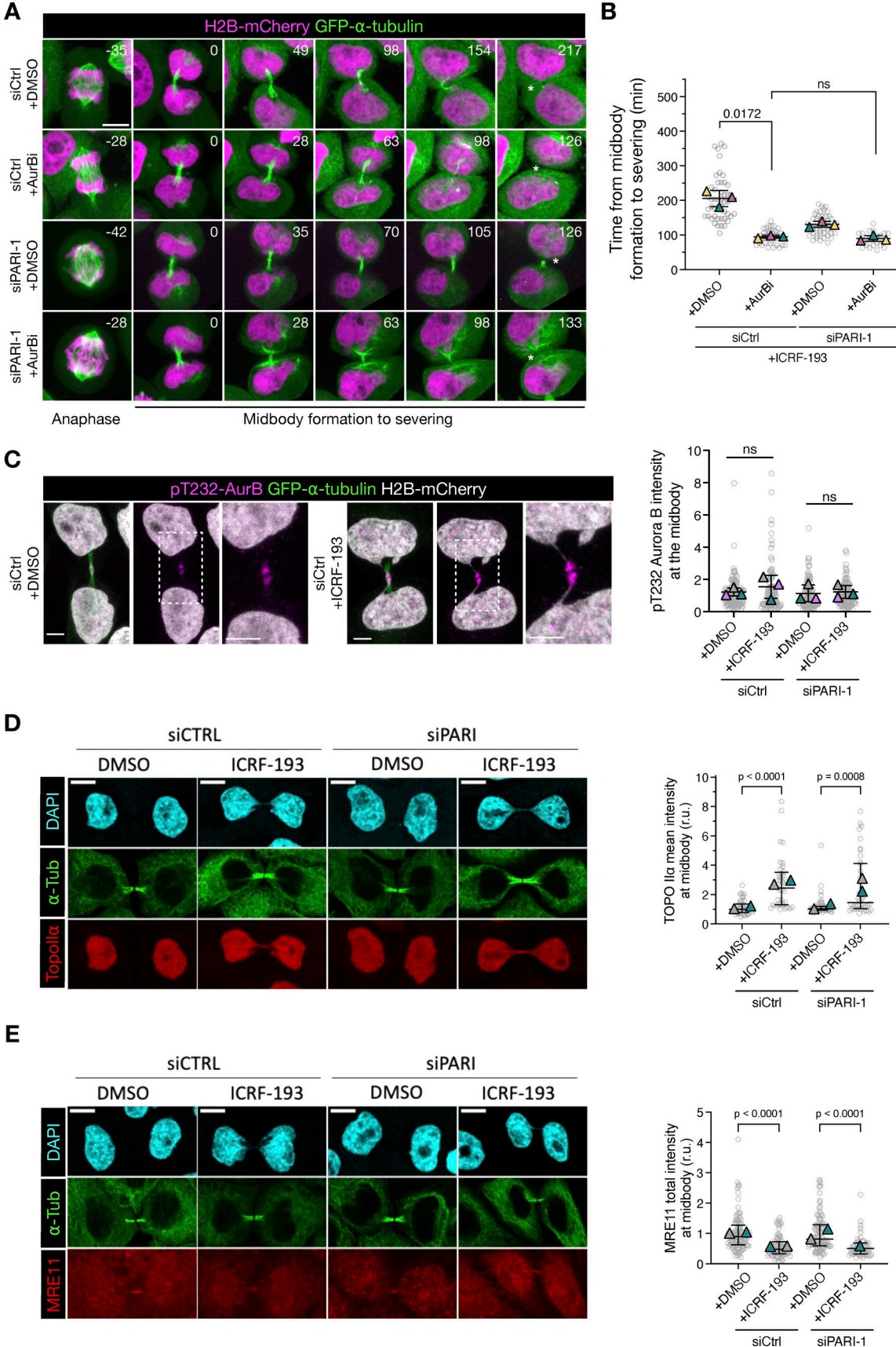

Figure 10. **PARI and Aurora B function in the same pathway to inhibit abscission. (A)** Cells treated with ICRF-193 progressing from anaphase to midbody formation (GFP-alpha-tubulin) and its disassembly. The Aurora B inhibitor hesperadin was added 120 min after the start of acquisition when most cells on

average had formed the midbody. Asterisk specifies midbody disassembly. Scale bar: 5 um. **(B)** Quantification of the midbody lifetime (time from midbody formation to severing). **(C–E)** Immunofluorescence images showing Aurora B phosphorylated on T232 (pT232-AurB, magenta), GFP-α-tubulin (green), and H2B-mCherry (gray) (in C); DAPI (cyan), α-tubulin (green), and Topo IIα (red) (in D); and DAPI, α-tubulin (green), and MRE11 (red) in E. Quantifications (right) show the levels of the indicated proteins at the midbody. In plots in B–E, each circle represents one cell, pooled from two or three independent experiments; triangles indicate the mean value of each experiment; bars indicate the median ± IQR. The Mann–Whitney test was used for statistical analysis; P-values are indicated. P > 0.05 = ns. Scale bars: 5 µm in A and C; 10 µm in D and E.

MRE11 levels were unchanged following PARI depletion. Together, these results suggest that PARI acts upstream or in concert with Aurora B to regulate midbody stability, but does not influence the localization of abscission checkpoint components such as active Aurora B, Topo IIα, or MRE11, at least at the time point examined.

## Discussion

The NoCut checkpoint plays a critical role in ensuring the proper coordination between chromosome segregation and cytokinesis, thus safeguarding genomic integrity during cell division. Disruption of this checkpoint can lead to the damage of chromatin bridges and genomic instability. Understanding the molecular mechanisms behind NoCut is crucial for elucidating how cells protect themselves against such potentially catastrophic events. We hypothesized that DNA helicases might play a role in NoCut signaling because they are well positioned to mark, either directly or indirectly, chromatin bridges that arise from replication or topological defects, both of which are known triggers of NoCut signaling. Srs2, for example, is recruited to replication forks by SUMOylated PCNA to prevent HR during the S phase and G2, and is associated with ultrafine bridges. However, its potential functions during mitosis and cytokinesis had not been explored prior to this study.

Our findings reveal that Srs2 plays a crucial role in chromosome segregation, particularly following DNA replication stress. Srs2 appears to prevent the accumulation of unreplicated DNA during anaphase, reducing the formation of aberrant DNA structures that could otherwise lead to segregation defects. In the absence of Srs2, cells exhibited increased RPA-coated chromatin bridges associated with delayed bridge resolution. This suggests that Srs2's activity is important for resolving these aberrant DNA structures caused by replication stress.

Importantly, Srs2 may play a dual role in protecting cells from DNA damage: first, by promoting the resolution of these structures, and second, by potentially triggering the NoCut checkpoint if these structures remain unresolved by the end of cytokinesis. The fact that srs2Δ cells exhibited delayed bridge resolution but did not display the NoCut-dependent abscission delay typically seen in response to chromatin bridges implies that Srs2 might be involved in signaling unresolved structures for NoCut checkpoint activation.

We further demonstrate that Srs2 is required to delay abscission in the presence of both HU-dependent and catenated chromatin bridges. Our observation that depleting the abscission factor Cyk3 reduces DNA damage in srs2Δ cells suggests that Srs2's role in delaying abscission and protecting chromatin bridges from damage is distinct from its anti-recombinogenic

function during fork resolution after HU treatment. This indicates that Srs2's function in cytokinesis (at least in the formation of breaks) is independent of its role in DNA damage repair. Moreover, our experiments with mutants affecting the Srs2-PCNA interaction suggest that Srs2 retention in chromatin bridges may be both necessary and sufficient to enforce an abscission delay. We also show that Srs2 and Aurora B/Ipl1 act in the same genetic pathway, and that Srs2's helicase activity is not required for abscission inhibition. Thus, Srs2 may act as a chromatin-based signal for the presence of chromatin bridges, potentially marking these structures for recognition by the NoCut checkpoint machinery.

In human cells, we found that PARI, the Srs2 homolog, plays similar roles in both chromosome segregation and the abscission checkpoint. Like Srs2, PARI is an inhibitor of HR and is recruited to stalled or collapsed replication forks under replication stress. We also found that PARI promotes resolution of chromatin bridges in interphase, and stabilizes midbodies and actin structures at the division site, both of which are hallmarks of the NoCut/abscission checkpoint, in response to chromatin bridges. This role seems to be specific to the response to chromatin bridges, as PARI does not participate in the midbody stabilization seen with nuclear pore complex assembly defects, highlighting its specificity within the mammalian NoCut checkpoint.

The Aurora B–dependent abscission checkpoint acts via effectors like phosphorylated Aurora B, Topo IIα, and the MRN complex. These proteins localize correctly in cells with chromatin bridges even when PARI is depleted, suggesting that PARI does not affect their initial recruitment. The nonadditive effects of Aurora B inhibition and PARI depletion support a model in which PARI operates either upstream or together with Aurora B to sustain midbody integrity in the presence of chromatin bridges.

A notable distinction between yeast and human systems is that unlike Srs2, PARI does not appear to be essential for abscission inhibition. In HeLa cells, chromatin bridges can persist well after midbody disassembly, and despite PARI depletion. Although it is possible that incomplete knockdown may have left residual PARI activity, these findings suggest that abscission inhibition in human cells can be sustained independently of PARI or midbodies, implying the existence of additional mechanisms. We note that the high level of binucleation induced by 250 nM ICRF-193 may have masked more subtle protective effects of PARI; titration with lower inhibitor concentrations could help clarify this possibility in future studies.

In summary, our data reveal an evolutionarily conserved function of the DNA helicase Srs2/PARI in controlling the timing of abscission-associated events in response to chromatin bridges. Despite these insights, the molecular mechanisms by

which Srs2 and PARI function in NoCut remain unclear. We do not yet understand how Srs2 promotes the recognition of DNA bridges. One possibility is that its presence on incompletely replicated or catenated DNA at the spindle midzone—where Aurora B resides—might enhance Aurora B activity or signaling to downstream NoCut components. While Srs2 has been associated with chromatin bridges, there is no clear evidence that PARI or PCNA localize to these bridges in human cells. Our attempts to detect PARI-GFP or PCNA by immunofluorescence were unsuccessful, although this could be due to their low abundance at these sites.

Interestingly, Topo II has been suggested to mark catenated bridges for recognition by the human abscission checkpoint (Petsalaki et al., 2023). Yet in our study, low-dose ICRF-193, which partially inhibits Topo II, still permits midbody and actin stabilization, suggesting that at least some degree of checkpoint activation persists under partial Topo II inhibition. The low concentration of ICRF-193 used in our study, 40-fold lower than in Petsalaki et al. may only partially inactivate Topo II, allowing the formation of chromatin bridges while retaining sufficient Topo II activity to maintain abscission checkpoint function. It is not yet clear whether Srs2/PARI and Topo II mark bridges independently or as part of a larger complex. The possibility of such a complex is supported by yeast two-hybrid studies demonstrating an interaction between Srs2 and Topo II (Chiolo et al., 2005), suggesting that these proteins could collaborate in marking chromatin bridges for NoCut signaling.

Finally, the association of PARI with cancer raises the intriguing possibility that its function in the NoCut checkpoint may contribute to maintaining genomic stability and preventing tumorigenesis. Interestingly, a polymorphism in the human ESCRT-III component CHMP4C, which impairs the abscission checkpoint, has been linked to increased cancer susceptibility (Sadler et al., 2018). Understanding the specific role of PARI in the NoCut checkpoint and how it might intersect with other pathways involved in genomic stability could provide valuable insights into the mechanisms underlying cancer development.

# Materials and methods

## Yeast strains and culture
*Saccharomyces cerevisiae* strains are described in Table S1. Gene deletions and insertions were generated by standard PCR methods (Janke et al., 2004) or through crossing. Cells were grown in YPDA media (yeast extract, peptone, 2% glucose, and adenine) at the permissive temperature of 30°C, and temperature-sensitive strains were grown at 25°C and shifted to 37°C to inactivate the protein. Dicentric strains were grown in YPDA supplemented with 2% galactose instead of glucose.

For all synchronizations, cells were grown overnight in YPDA at the permissive temperature, diluted to $OD_{600} = 0.1$, and grown for 3 h to reach the log phase. To induce expression and localization of CAAX-GFP to the plasma membrane, 90 nM β-estradiol was added to the media for 3 h. To generate replication stress, 200 mM HU was added to the media for 3 h. To arrest cells in G1, cells were synchronized with 20 μm/ml α-factor for 2 h. To release from the G1 block or HU treatment, cells were washed twice in freshly prepared and preheated minimal synthetic (Yeast nitrogen base, 2% glucose, essential amino acids) media and immediately plated on concanavalin A–coated Lab-Tek chambers for microscopy. For analysis of Rfa2-GFP during cell division, visual inspection showed that all cells displayed GFP-labeled nuclear foci during interphase, as described previously (Ivanova et al., 2020). Cells that retained foci with intensity higher than the nucleoplasmic background after the onset of nuclear elongation were classified as "chromosome segregation with RPA foci." $SRS2^{\Delta SIM}$ and $SRS2^{\Delta PIP\Delta SIM}$ were constructed by introducing a stop codon after amino acid positions 1167 or 1149 (respectively) with the selection marker; $SRS2^{\Delta PIP}$ was constructed by introducing after the amino acid position 1149 the sequence coding for the amino acids 1167–1174 and a stop codon with the selection marker.

## Human cell lines and culture conditions
All cell lines were cultured at 37°C in a 5% $CO_2$ humidified incubator. Standard media, DMEM GlutaMAX (4.5 g/Liter glucose), supplemented with 10% fetal calf serum, 1% penicillin, and streptomycin were used to culture all cell lines. The HeLa Kyoto cell line was obtained from the in-house cell culture facility. HeLa stable cell lines were kindly shared by Dr. Daniel Gerlich (Institute of Molecular Biotechnology, Vienna, Austria) and were grown in standard media with additional selective antibiotics.

## Plasmid and siRNA transfection
A FLAG-PARI Gateway Destination plasmid (Burkovics et al., 2016) was used to clone PARI into pcDNA3.1(+) plasmid by the molecular biology facility at IGBMC. All plasmid cloning and primer design were performed by the molecular biology facility. An eGFP fluorophore was introduced at the N terminus of PARI. For the siPARI-1–resistant version of eGFP-PARI, three silent mutations were introduced at the seed region of the siPARI-1 target site.

To generate stable cell lines, HeLa Kyoto cells were transfected with linearized pcDNA3.1(+)-derived plasmids using X-tremeGENE 9 DNA Transfection Reagent (ROCHE) according to the manufacturer's protocol. Transfected cells were selected for 2–3 wk in standard media supplemented with G418 (0.8 mg/ml). Transgene-positive cells were isolated by FACS (FACS ARIA, BD Biosciences) or by manually isolating single colonies. Expression was validated by PCR, IF, and WB.

All siRNAs used are of the siGENOME product line from Dharmacon and were purchased from Horizon (Table S2). To knock down PARI, resuspended cells were (reverse)-transfected using 25 nM siRNA and Lipofectamine RNAiMAX (Invitrogen) according to the manufacturer's protocol, the siRNA–lipid complexes were removed after 7 h, and a second (forward) transfection was performed 24 h after the first one. For co-depletion of PARI and NUP153, 10 nM siNUP153 was used together with 25 nM siCtrl or siPARI-1 24 h after reverse transfection with siCtrl and siPARI-1, and the amount of Lipofectamine RNAiMAX was adjusted accordingly.

## Quantitative PCR and analysis
RNA was extracted using the RNeasy kit (Qiagen), and 2.5 μg RNA was incubated with ezDNase for 10 min at 37°C before performing

reverse transcription using SuperScript IV VILO Master Mix with ezDNase (11766050; Invitrogen) according to the manufacturer's protocol. Alternatively, DNase I (4716728001; ROCHE DIAGNOSTICS) was used, and reverse transcription was performed with random hexamer (Invitrogen) and oligo(dT) primers (N8080127 and 18418020; Invitrogen), SuperScript IV, and RNaseOUT Recombinant RNase Inhibitor (10777019; Invitrogen) according to the manufacturer's protocol. SYBR Green–based qPCR and analysis were carried out on the LightCycler 480.

All primer pairs were validated with PCR and serial dilution to determine the primer efficiency; only those that showed >90% amplification efficiency were used for analysis (Table S3). The Ct value was obtained from LightCycler 480 software and used to measure the relative expression (RE) of each sample, which was calculated using the following equation:

$$RE = 2^{-(\Delta Ct)},$$

where $\Delta Ct = Ct(\text{target gene}) - Ct(\text{reference gene})$.

The RE of the test sample is then normalized to the RE of the control.

### Immunofluorescence microscopy and image analysis
20,000 cells were seeded on 13-mm sterile glass coverslips in 24-well culture plates and fixed with 4% PFA for 15 min at RT. PFA was washed out three times with 1× PBS, and cells were permeabilized in 0.2% Triton X-100 in 1× PBS, for 10 min at RT, followed by a 1-h block with 3% BSA in 1× PBS at RT with gentle agitation. After blocking, cells are incubated with primary antibodies (Table S4) for 1 h at RT or overnight at 4°C, followed by three washes with 1× PBS. Cells are then subjected to secondary antibodies (Table S4) and DAPI for 1 h at RT, followed by three washes with 1× PBS. Excess 1× PBS was carefully tapped off from the coverslips and mounted on a glass slide with Vectashield mounting medium (Vecta Laboratories) or ProLong Gold (Invitrogen). The HCX PL APO 63×/1.40 OIL PH3 C3 objective was used on an upright motorized Leica DM 4000 B equipped with CoolSNAP HQ2 camera. For quantification of proteins at the midbody, an elliptical region of interest (ROI) was manually defined around the midbody arms in the best-focused Z-slice, selected based on the tubulin signal. A maximum intensity projection of the five Z-slices centered on the selected slice was generated for all channels. The tubulin signal in the midbody arms was used to segment the region and automatically fit a tight ellipse to it. The area and mean fluorescence intensity of this fitted ellipse were then measured in the channel corresponding to the protein of interest in the immunofluorescence experiment. Total intensity was calculated as the product of the ellipse area and its mean fluorescence intensity.

### Time-lapse fluorescence spinning disk confocal microscopy of HeLa cells
Cells were seeded 3 days in advance on a 4-chamber 35-mm glass bottom dish (Cellvis) and placed in a preheated chamber at 37°C and 5% $CO_2$ for time-lapse microscopy. For cell lines that do not stably express fluorescently tagged tubulin, cells were incubated with 15 nM SiR-tubulin (SC0022; Spirochrome) and 1 μM Verapamil hydrochloride (TEBU-BIO T1010-1) for 9 h in media before

acquisition. All time-lapse acquisitions were set up using Meta-Morph software. Images were acquired using 63× water objective (HC PL APO 63×/1,20 W CORR CS2) with Leica Water Immersion Micro Dispenser connected to a Bartek extended micropump on an inverted Leica DMI8 microscope equipped with Yokogawa CSU W1 spinning disk and Evolve 512 camera or an inverted Nikon Eclipse equipped with Yokogawa CSU X1 spinning disk and photometric prime 95B camera. Z-stack (15 μm range, 0.3 μm step size) images were acquired every 5–15 min for 14–24 h. For time-lapse microscopy of Aurora B inhibition with hesperadin, media with a final concentration of 100 nM hesperadin were added on top of the cells in between intervals.

### Analysis of actin clearance
We defined an "actin-cluster index" to quantify how strongly bright actin signal accumulates in a tight cluster near the midzone, and used it to track furrow-associated actin dynamics over time. A rectangular ROI encompassing the dividing cell was manually defined within the whole field of view. Actin and histone fluorescence channels were thresholded to segment regions with positive signal, and background signal in the actin channel was eliminated by setting nonrelevant pixels to zero. Nuclei were segmented from the thresholded histone channel, with objects touching the image borders excluded from analysis. Only frames containing exactly two nuclei were retained for further analysis. For each time frame, the centroid coordinates and major axis lengths of the two nuclei were determined. A rectangular ROI was then constructed between the two centroids. This ROI had a height equal to half the internuclear distance and a width equal to twice the average major axis length of the two nuclei, thus covering the region where the cleavage furrow and postanaphase actin patches are typically located. Within this ROI, the average actin intensity was calculated in two ways: across all nonzero (actin-positive) pixels, and among the top 5% brightest pixels. A *brightness enrichment* metric was defined as: $\frac{Int_{top5\%} - Int_{all}}{Int_{all}}$, where $Int_{top5\%}$ is the mean intensity of the top 5% brightest actin-positive pixels, and $Int_{all}$ is the mean intensity of all actin-positive pixels in the ROI. To assess spatial organization, the center of mass (COM) of the top 5% bright pixels was calculated. Two distance metrics were then computed: the average distance from all actin-positive pixels to the COM ($dCPM_{all}$) and the average distance from only the top 5% bright pixels to the COM ($dCOM_{top5\%}$). A *clustering* index was defined as: $\frac{dCOM_{all}}{dCOM_{top5\%}}$. Finally, the actin-cluster index was computed as the product of "brightness enrichment" and "clustering." The index was quantified for ~30 cells per condition; only cells with a clear actin signal and whose midzone was free of overlapping GFP-actin from neighboring cells were included. Actin clearance timings on approximately one hundred cells per condition were scored visually.

### Correlative light electron microscopy
Adherent HeLa Kyoto stably expressing H2B-mCherry GFP-α-tubulin cells were first cultured on laser micropatterned Aclar supports (Spiegelhalter et al., 2014), synchronized with double thymidine and released for 12 h. Cells were then fixed with 1% glutaraldehyde and 4% formaldehyde in a 0.1 M phosphate buffer for 30 min. Cells in cytokinesis with or without a catenated

chromatin bridge were selected, precisely located, and imaged by an inverted Nikon Eclipse equipped with a photometric Prime 95B camera. At the end of the experiment, cells were immediately fixed with 2.5% glutaraldehyde and 4% formaldehyde in 0.1 M phosphate buffer for 1 h (or longer) at 4°C, rinsed in buffer, and followed by 1-h postfixation in 1% osmium tetroxide $[OsO_4]$ reduced by 0.4% potassium hexacyanoferrate (III) $[K_3Fe(CN)_6]$ in $H_2O$ at 4°C. Samples were rinsed in distilled water and stained with 1% tannic acid for 30 min on ice and, after extensive rinses, with 2% uranyl acetate for 1 h at room temperature, rinsed in water. Samples were dehydrated with increasing concentrations of ethanol (25%, 50%, 70%, 90%, and 3 × 100%) and embedded with a graded series of epoxy resin. Samples were finally polymerized at 60°C for 48 h. Ultrathin serial sections (70 nm) were picked up on 1% Pioloform-coated copper slot grids and observed with a Philips CM12 operated at 80 kV equipped with an Orius 1000 CCD camera (Gatan).

### Statistical methods

GraphPad Prism software was used to generate graphs and perform statistical tests. The Mann–Whitney test was used on datasets that did not follow a normal distribution. For experiments with at least three independent biological replicates, Student's paired $t$ test was used. Fisher's exact test was performed on data that were pooled from at least two biological replicates to compare fractions.

### Online supplemental material

Fig. S1 shows all frames of time-lapse microscopy of chromosome segregation and RPA focus formation in yeast strains shown in Fig. 1 A. Fig. S2 presents cumulative frequency plots and quantifications comparing the timing of chromatin bridge resolution, membrane ingression, and abscission relative to anaphase onset in WT and *srs2Δ* cells, under normal and HU-treated conditions; and the fraction of anaphase cells with RPA foci in *srs2* and *top2* mutants at 37°C. Fig. S3 details the synchronization of HeLa cells transfected with siCtrl or siPARI siRNAs, analysis of DNA content by flow cytometry, and RT-qPCR quantification of PARI mRNA levels. Fig. S4 displays CLEM images of cells expressing H2B-mCherry and GFP-alpha-tubulin transfected with control or PARI-1 siRNA. Fig. S5 includes design and characterization of eGFP-PARI$^R$ cell line. Supplementary Table S1 lists all *Saccharomyces cerevisiae* strains used in this study. Table S2 provides siRNA sequences employed for gene knockdown in human cells. Table S3 lists primer sequences used for qPCR analysis. Table S4 describes the primary and secondary antibodies used for western blotting and immunofluorescence. Videos 1, 2, 3, and 4 show time-lapse imaging of HeLa cells expressing H2B-mCherry and GFP-α-tubulin, providing examples of how chromatin bridges induced by ICRF-193 and PARI depletion affect midbody severing. Videos 5, 6, and 7 show HeLa cells expressing eGFP-BAF and stained with SiR-tubulin, giving examples of divisions without bridges, with transient bridges, and with persistent bridges followed by binucleation. Videos 8, 9, 10, and 11 show actin dynamics in HeLa cells expressing H2B-mCherry and GFP-actin, providing examples of normal actin clearance, long-lived ICRF-193–induced actin clusters, and their shortened persistence upon PARI depletion.

### Data availability

Data are available in the article itself and its supplemental material.

## Acknowledgments

We thank all members of the Mendoza laboratory for insightful discussions, and Arnaud Echard, Andrés Clemente, and Felix Prado for their critical reading of the manuscript. We are grateful to the CRG Advanced Light Microscopy Unit and the IGBMC Imaging Centre—part of the national infrastructure France-BioImaging and supported by the French National Research Agency (ANR-10-INBS-04)—as well as the IGBMC Cell Culture and Flow Cytometry Facilities, for their technical support.

This work was funded by a European Research Council (ERC) Starting Grant (2010-St-20091118) and by grants from the Ligue Nationale Contre le Cancer (3FI13531TJQN and 3FI14004UUGH) awarded to Manuel Mendoza. Additional support came from the Spanish Ministry of Economy and Competitiveness through the "Centro de Excelencia Severo Ochoa 2013–2017" program (SEV-2012-0208) to the CRG. Monica Dam was supported by a LabEx PhD fellowship from IGBMC and a fellowship from the Ligue Nationale Contre le Cancer. Nicola Brownlow received a Marie Skłodowska-Curie postdoctoral fellowship (ID 705602). David F. Moreno was supported by Fondation ARC (POSTDOC 2023080006949). Finally, this work, part of the Interdisciplinary Thematic Institute IMCBio+ 2021-2028 program of the University of Strasbourg, CNRS, and Inserm, was supported by IdEx Unistra (ANR-10-IDEX-0002), SFRI-STRAT'US project (ANR-20-SFRI-0012), and EUR IMCBio (ANR-17-EURE-0023) under the France 2030 Program framework.

Author contributions: Monica Dam: conceptualization, data curation, formal analysis, investigation, methodology, validation, visualization, and writing—original draft. David F. Moreno: conceptualization, data curation, investigation, methodology, software, validation, visualization, and writing—review and editing. Nicola Brownlow: conceptualization, data curation, formal analysis, and investigation. Audrey Furst: investigation and methodology. Coralie Spiegelhalter: formal analysis, investigation, and methodology. Manuel Mendoza: conceptualization, data curation, funding acquisition, supervision, visualization, and writing—original draft, review, and editing.

Disclosures: The authors declare no competing interests exist.

Submitted: 3 February 2025

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

# Supplemental material

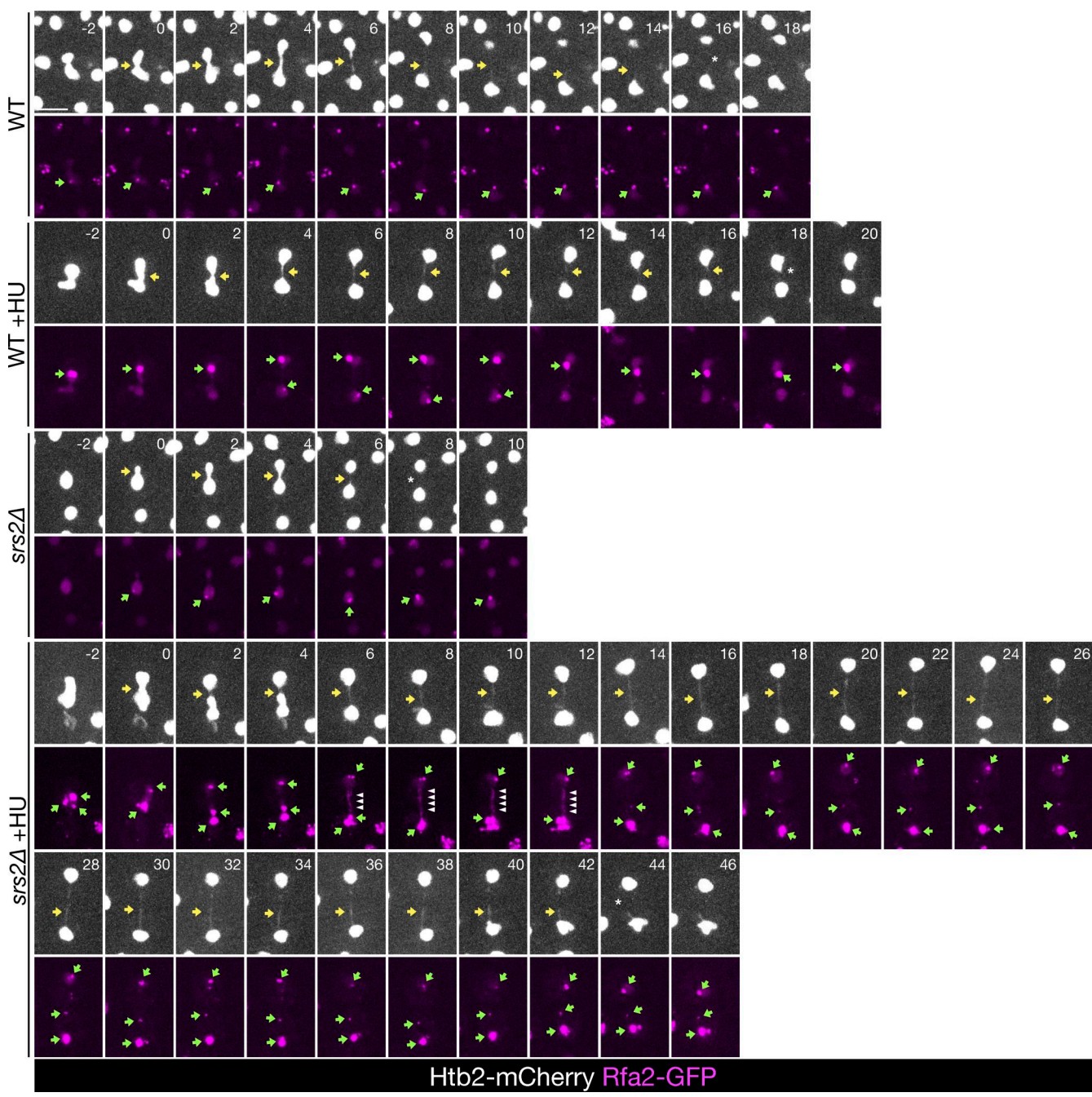

Figure S1.  **Chromosome segregation (Htb2-mCherry) and RPA focus formation (Rfa2-GFP) in cells of the indicated strains, shown in** Fig. 1 A**, including all time points, with and without previous exposure to HU.** Only cells with RPA foci are shown. Yellow arrows indicate chromatin bridges, green arrows point toward RPA foci, white arrowheads mark RPA bridges, and asterisks note bridge resolution. Scale bar: 5 µm.

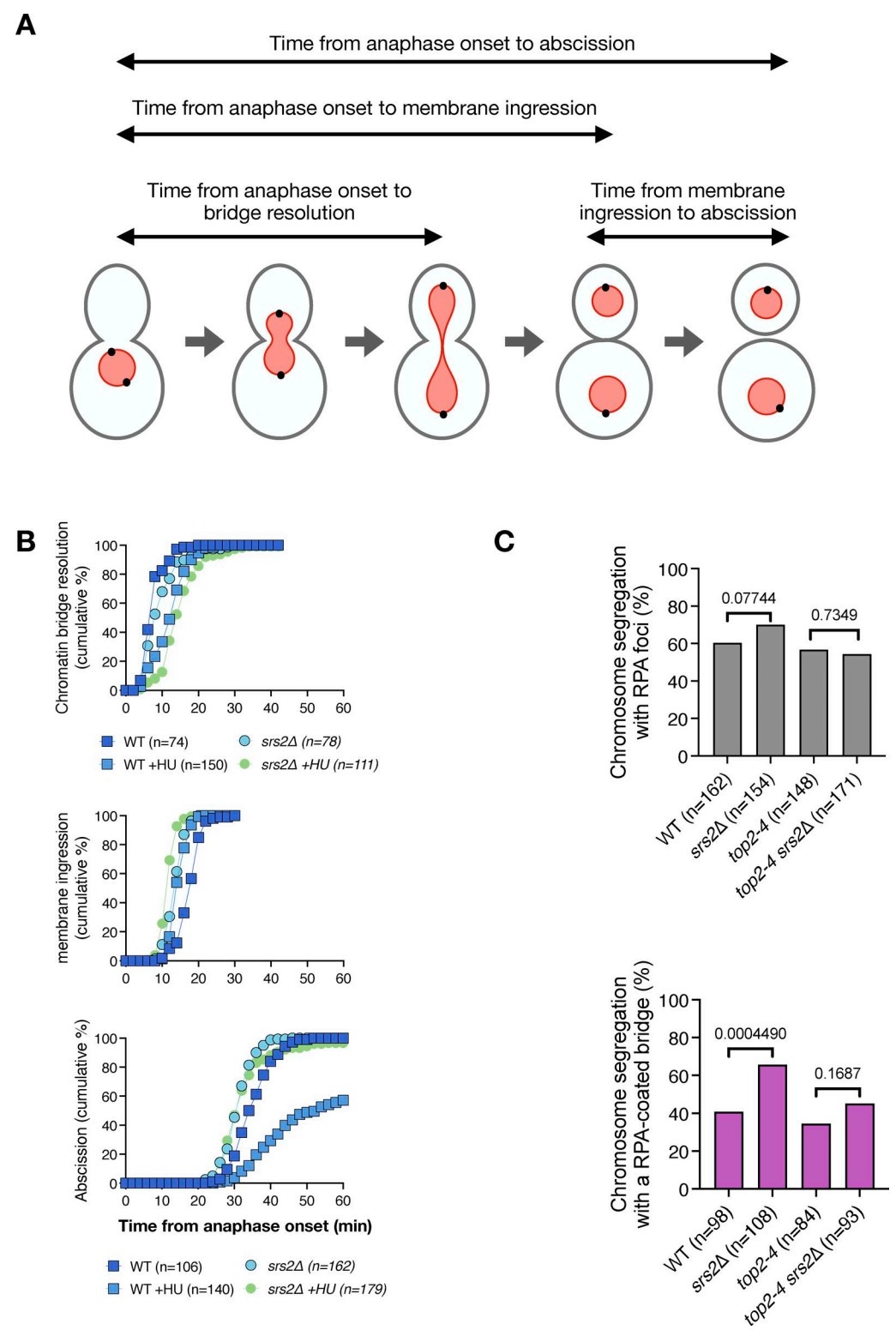

Figure S2. **Timing of chromatin bridge resolution, membrane ingression, and abscission in WT and *srs2Δ* cells. (A)** Schematic representation of key cell division events, including chromatin bridge resolution, membrane ingression, and abscission. The time intervals measured in the analysis are indicated by horizontal arrows. The nucleus is in red, the spindle pole bodies are black circles, and the plasma membrane is in gray. **(B)** Cumulative frequency plots showing the timing of chromatin bridge resolution (top), membrane ingression (middle), and abscission (bottom) in WT and *srs2Δ* cells, with or without HU treatment. The number of cells analyzed for each condition is indicated in the legend. Time from anaphase onset was defined by nuclear elongation (Htb2-mCherry) in the top panel, and by spindle elongation (Spc42-GFP) in the middle and bottom panels. Data in the top panel are replotted from Fig. 1, A and B; data in the middle and bottom panels are from Fig. 2, B and C. **(C)** Fraction of cells of the indicated strains undergoing chromosome segregation with RPA foci at 37°C (top) and fraction of cells that segregated with RPA foci and had RPA-coated chromatin bridge (bottom). Cells are pooled from two independent experiments; P-values correspond to Fisher's exact test.

**A**

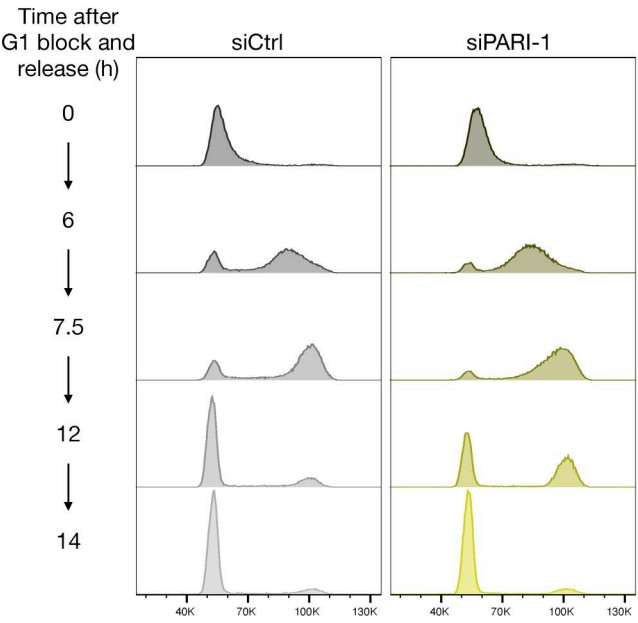

**B**

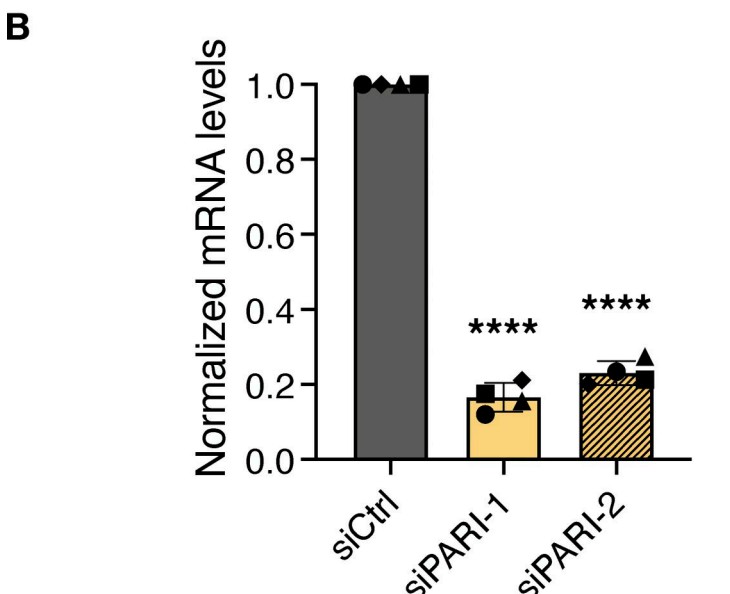

Figure S3. **Workflow for cell cycle synchronization and validation of PARI depletion efficiency by siRNA. (A)** Top: Experimental workflow to synchronize siRNA-transfected cells and inducing catenated chromatin bridges during cytokinesis by the addition of ICRF-193 during G2. Bottom: Flow cytometry analysis of DNA content in siCtrl and siPARI-1 cells at the indicated times after G1 release. **(B)** RT-qPCR measures the mRNA levels of PARI in HeLa cells after transfecting twice with 25 nM siCtrl, siPARI-1, and siPARI-2. Relative mRNA levels have been normalized to siCtrl. One-sample t and Wilcoxon test (mean ± SD, ****$P < 0.0001$, $N = 4$).

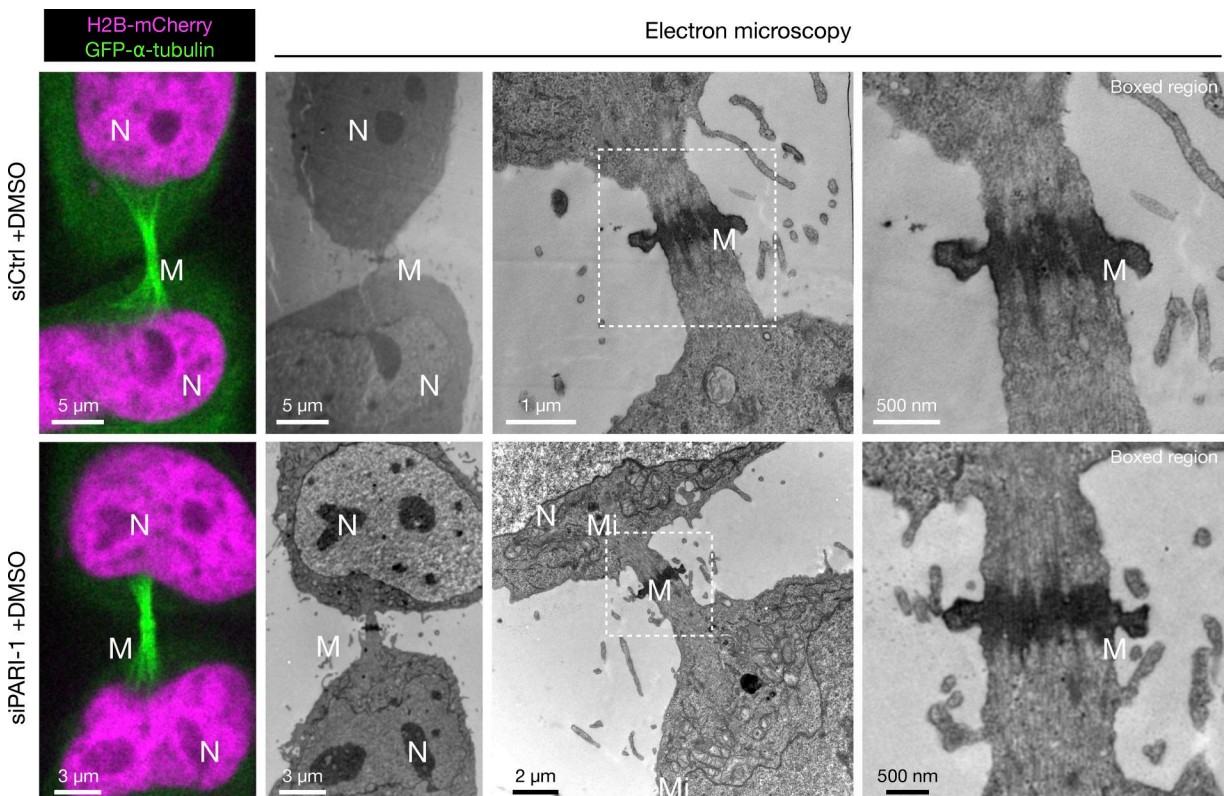

Figure S4. **Cells stably expressing H2B-mCherry and GFP-alpha-Tubulin, transfected with control or PARI-1–specific siRNAs.** Boxed regions zoom in on the Flemming body. N = nucleus; M = midbody; Mi = mitochondria.

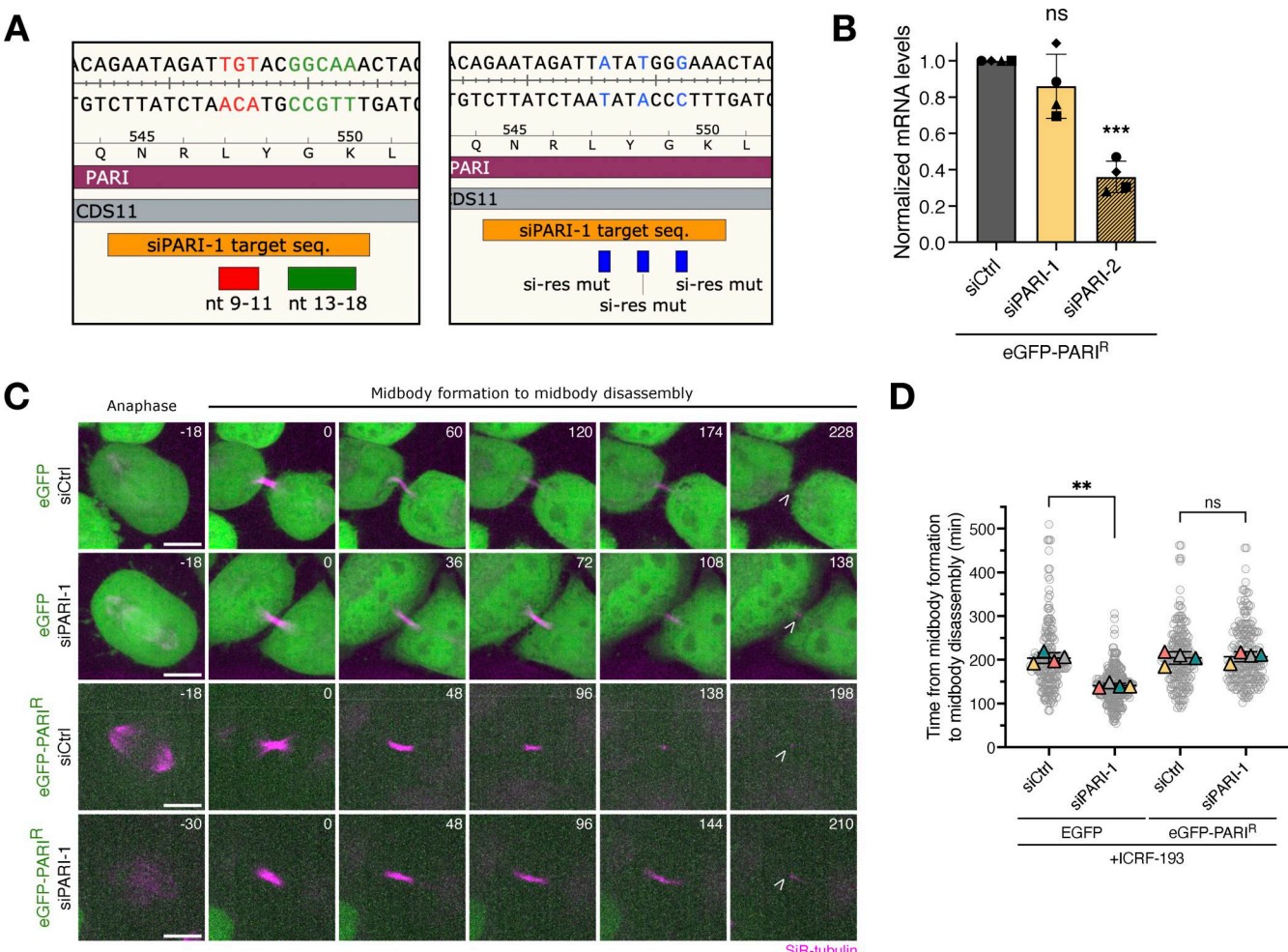

**Figure S5.  Expression of siPARI-1-resistant eGFP-PARI rescues the ICRF-193-induced midbody lifetime increase. (A)** Left panel: WT siPARI-1 target sequence, features highlighting seed sequences. Right panel: siPARI-1–resistant PARI sequence; features show where three silent point mutations were introduced by cloning. **(B)** RT-qPCR measures the mRNA levels of PARI in HeLa cells after transfecting twice with 25 nM siCtrl, siPARI-1, and siPARI-2. Relative mRNA levels have been normalized to siCtrl. One-sample t and Wilcoxon test (mean ± SD, *P < 0.05, **P < 0.01, ***P < 0.001, ****P < 0.0001, N = 4). **(C)** Cells treated with ICRF-193 progressing from anaphase to midbody formation (SiR-tubulin) and its disassembly. Cells have been synchronized and treated as illustrated in Fig. 5. Open arrowhead specifies midbody disassembly. Scale bar: 5 μm. **(D)** SuperPlot showing the quantification of the midbody lifetime (time from midbody formation to disassembly). Student's paired t test (mean ± SD, *P < 0.05, **P < 0.01, ***P < 0.001, ****P < 0.0001, n = 165, N = 4).

Video 1.  **Associated with** Fig. 6 A**.** Time-lapse imaging of a HeLa cell expressing H2B-mCherry and GFP-α-tubulin, treated with DMSO and transfected with control siRNA. Shows normal progression from metaphase to midbody severing.

Video 2.  **Associated with** Fig. 6 A**.** Time-lapse imaging of a HeLa cell expressing H2B-mCherry and GFP-α-tubulin, treated with ICRF-193 and transfected with control siRNA. Midbody severing is delayed.

Video 3.  **Associated with** Fig. 6 A**.** Time-lapse imaging of a HeLa cell expressing H2B-mCherry and GFP-α-tubulin, treated with DMSO and PARI siRNA. Shows normal progression from metaphase to midbody severing.

Video 4.  **Associated with** Fig. 6 A**.** Time-lapse imaging of a HeLa cell expressing H2B-mCherry and GFP-α-tubulin, treated with ICRF-193 and depleted of PARI. Despite Topo II inactivation, midbody severing occurs prematurely.

Video 5.  **Associated with** Fig. 7 A**.** Time-lapse imaging of a HeLa cell expressing eGFP-BAF and stained with SiR-tubulin, control siRNA + DMSO. Cell divides without a chromatin bridge.

Video 6.  **Associated with** Fig. 7 A**.** Time-lapse imaging of a HeLa cell expressing eGFP-BAF and stained with SiR-tubulin, control siRNA + DMSO. A spontaneous chromatin bridge is present, persists during cytokinesis, and is eventually retracted.

Video 7.  **Associated with** Fig. 7 A**.** Time-lapse imaging of a HeLa cell expressing eGFP-BAF and stained with SiR-tubulin, control siRNA + ICRF-193. A persistent chromatin bridge is accompanied by cytokinesis failure and binucleation.

Video 8.   **Associated with** Fig. 9 A**.** Time-lapse imaging of a HeLa cell expressing H2B-mCherry and GFP-actin, control siRNA + DMSO. Actin accumulates at the cleavage furrow and clears after ∼60 min.

Video 9.   **Associated with** Fig. 9 A**.** Time-lapse imaging of a HeLa cell expressing H2B-mCherry and GFP-actin, PARI siRNA + DMSO. Actin accumulates and clears normally, similar to control.

Video 10.   **Associated with** Fig. 9 A**.** Time-lapse imaging of a HeLa cell expressing H2B-mCherry and GFP-actin, control siRNA + ICRF-193. Dynamic actin clusters form at the ICB and persist for hours in association with chromatin bridges.

Video 11.   **Associated with** Fig. 9 A**.** Time-lapse imaging of a HeLa cell expressing H2B-mCherry and GFP-actin, PARI siRNA + ICRF-193. Actin clusters appear but dissipate prematurely compared with controls.

**Provided online are Table S1, Table S2, Table S3, and Table S4. Table S1 shows *Saccharomyces cerevisiae* strains. Table S2 shows siRNA sequences for gene knockdown. Table S3 shows primer sequences for qPCR. Table S4 shows antibodies used for western blot and immunofluorescence.**

