## [Peer Review File · The Journal of Cell Biology]

Roles of Srs2/PARI-family DNA helicases in NoCut checkpoint signaling and abscission regulation

Monica Dam, David Moreno, Nicola Brownlow, Audrey Furst, Coralie Spiegelhalter, and Manuel Mendoza

Corresponding Author(s): Manuel Mendoza, Institute of Genetics and Molecular and Cellular Biology

Review Timeline:

Submission Date:	2025-02-03
Editorial Decision:	2025-03-05
Revision Received:	2025-05-09
Editorial Decision:	2025-06-23
Revision Received:	2025-09-04
Editorial Decision:	2025-09-10
Revision Received:	2025-09-11

Monitoring Editor: Karen Oegema

Scientific Editor: Dan Simon

Transaction Report:

DOI: <https://doi.org/10.1083/jcb.202502014>

Revision 0

Review #1

1. Evidence, reproducibility and clarity:

Evidence, reproducibility and clarity (Required)

The abscission checkpoint, also known as NoCut, is a genome protection mechanism that remains poorly understood. This pathway is conserved from yeast to humans and protects the genome against chromosome bridges, a dangerous missegregation event that can have catastrophic consequences on genome stability. Dam et al now report the role of Srs2, a DNA helicase, as a key factor in the abscission checkpoint. The authors establish Srs2 as bona fide factor in this pathway by showing its involvement in abscission delays when chromatin bridges are induced. Importantly, yeast defective for Srs2 show increased levels of DNA damage when the frequency of chromatin bridges is increased. The authors also provide genetic evidence supporting a model whereby the interaction of Srs2 with PCNA is required for abscission regulation. In the second part of the manuscript, the authors study the human homologue of SRS2, PARI, in abscission regulation. The manuscript provides convincing evidence that PARI is also required for abscission delays in the presence of chromatin bridges. Critically, this role is specific for chromosome missegregation as abscission delays in response to nucleoporin depletion remain intact in PARI-depleted cells. Thus there is a conserved requirement for these DNA helicases in the abscission checkpoint.

Overall, these are important advances in our understanding of the abscission checkpoint. The data is high quality and convincing in general. However, the impact of PARI depletion on genome stability needs to be further demonstrated to support key claims in the manuscript. Specifically:

Disruptions of the abscission checkpoint in human cells result in bi-nucleation or increased levels of DNA damage. In this context, the authors need to show that PARI-depleted cells with increased frequency of chromatin bridges exhibit increased levels of bi-nucleation, DNA damage or both.

2. Significance:

Significance (Required)

The abscission checkpoint, remains poorly understood. There is evidence in the literature that disruptions in this pathway increase susceptibility to cancer. The identification of the Srs2/PARI helicases as key components in this pathway is a considerable step forward in this field.

3. How much time do you estimate the authors will need to complete the suggested revisions:

Estimated time to Complete Revisions (Required)

(Decision Recommendation)

Between 1 and 3 months

4. Review Commons values the work of reviewers and encourages them to get credit for their work. Select 'Yes' below to register your reviewing activity at Web of Science Reviewer Recognition Service (formerly Publons); note that the content of your review will not be visible on Web of Science.

No

Review #2

1. Evidence, reproducibility and clarity:

Evidence, reproducibility and clarity (Required)

The Aurora B-mediated abscission checkpoint ("NoCut" in yeast) prevents tetraploidization or chromatin breakage in the presence of chromatin bridges in cytokinesis and the mechanisms of its activation are a matter of active investigation. In the present study, Dam et al propose that the conserved Srs2/PARI DNA helicase is required for the activation of the abscission checkpoint in response to chromatin bridges generated by DNA replication stress or topoisomerase inhibition. This is a timely and very interesting topic and the potential identification of a novel regulatory protein that activates the abscission checkpoint would be important. However, in my opinion, some Figures are of relatively low quality and need improving, there are apparent discrepancies between data and important control experiments are missing, which preclude the reader from fully evaluating the conclusions of this study. Some direct evidence of the role of Srs2/PARI on DNA bridges is also required. Also, it would be nice to investigate mechanistic details of the potential Srs2/PARI functions in the abscission checkpoint, and how it fits with other recently published signaling pathways that activate the abscission checkpoint in cytokinesis.

****Specific comments:****

1. The DNA channel (Ht2B-mCherry) in Figure 1A is of very low quality to be able to verify

the authors' interpretations of when the individual chromatin bridges are resolved (probably broken). For example, in the WT movie, they claim that the bridge is intact in frames 10 min and 14 min (yellow arrow) and that the bridge is resolved at 16 min (asterisk); however, I'm not convinced this is the case, because I can only see a very small portion of the bridge already at the 10 min and 14 min time-points. In my opinion, this bridge could have been broken much earlier, probably at 10 min. Also, WT +HU, is this bridge really intact at 10 min and at 14 min? In Srs2 Δ + HU, the bridge appears broken to me much earlier, perhaps at 30 min. There is a distinct possibility that the authors could not calculate the resolution times accurately from these movies (please also see my next comment, #2). The authors could perhaps use a more sensitive bridge marker such as GFP-BAF.

2. In Figure 1B, they conclude that Srs2 Δ cells treated with HU exhibit increased time from anaphase onset to bridge resolution compared with WT or Srs2 Δ cells. This result appears at odds with data from Fig. 2C showing that Srs2 Δ +HU finish abscission at similar times to WT or Srs2 Δ cells as judged by plasma membrane morphology. (final cut). Given that the final cut of the plasma membrane should cause chromatin bridges to break, if Srs2 is required for an abscission delay in response to HU-induced chromatin bridges, I would expect Srs2 Δ + HU cells to exhibit accelerated plasma membrane cut and also faster chromatin bridge resolution compared with controls. This discrepancy could at least in part be caused by the relatively low quality of movies used for the calculations in Fig. 1.

3. Fig. 2 shows faster abscission times (membrane cut) in Srs2 Δ +HU cells compared with WT+HU. The authors interpret this data as evidence for a role of Srs2 in abscission delay in response to HU-induced chromatin bridges (page 7 and elsewhere). However, there is no direct evidence that the cells analyzed in Fig.2 exhibited DNA bridges in cytokinesis. One could argue that HU-induced DNA replication stress caused DNA lesions at the nuclear chromatin, which affected completion of cytokinesis in the absence or presence of Srs2. What proportion of HU-treated cells in cytokinesis exhibit DNA bridges? Judging from Fig. 1D this could be as low as 0-20%. The authors should analyze HU-treated cells that clearly exhibit DNA bridges, either by live-cell imaging or in fixed cells experiments. As it stands and together with my previous comments #1 and 2, I'm not convinced this data fully supports a role for Srs2 in the abscission delay in response to HU-induced DNA bridges.

4. In Fig. 2D, there is no evidence to support that Mre11 foci are caused by bridge breakage, and not by replication-stress induced DNA lesions at the main nucleus (no DNA bridge is evident, also see comment #3).

5. Figure 3: the authors use a top2-4 mutant strain to generate DNA bridges from catenated DNA and investigate the potential role of Srs2 in the abscission delay. However, no DNA bridges are obvious in the cells shown in Fig. 3. What proportion of top2-4 mutant cells in cytokinesis exhibit DNA bridges? Does this explain the striking difference in the percentage of cells that haven't completed abscission after 30-60 min in WT+HU vs Top2-4 cells?

Please also see my previous comments above.

6. The authors propose that association of Srs2 with PCNA is required for complete inhibition of abscission in top2-4 mutant cells with chromatin bridges. Assuming a role for Srs2 in abscission timing in cytokinesis with chromatin bridges is fully proven, it is essential that the authors also investigate the localization of Srs2 and PCNA on chromatin bridges, using GFP-tagged proteins or appropriate antibodies in fixed and/or living cells. This would suggest a direct role of these proteins on chromatin bridges and considerably strengthen the authors hypothesis. Alternatively, Srs2 and PCNA may indirectly affect abscission timing through their well-established roles at nuclear chromatin.

7. In Fig. 4D, the authors show an abscission delay in *elg1Δ* mutant cells in the presence of dicentric bridges compared with cytokinesis without bridges and interpret this as evidence that artificially retaining PCNA on dicentric chromatin bridges is sufficient to inhibit abscission. It is important that the authors demonstrate that PCNA localizes to dicentric bridges in *elg1Δ* mutant, but not in ELG1 control, cells, e.g., by immunofluorescence, to support their claim and their proposed model.

8. In Fig. 5, the authors claim that HeLa cells treated with the Top2 inhibitor ICRF193 exhibit delayed midbody resolution compared with controls and that depletion of PARI by siRNA accelerates abscission in ICRF-treated cells. They interpret this as evidence for a role of PARI in the abscission delay in response to ICRF-induced chromatin bridges. However, no bridges are visible at any time-frame in cells in Fig. 5B raising the possibility that the observed time-differences are due to some effect of ICRF in cytokinesis without bridges. I'm also not convinced that in Fig. 5B the midbodies in NT/ICRF/230 min, siPARI/DMSO/110 min and siPARI/ICRF/150 min were resolved as indicated by the authors, as I can definitely see both midbody arms very clearly in these photos. The p-values are also just below the $p < 0.05$ threshold, which could in part be due to the quality of the movies quantified. Also, in Fig. 5C, the authors show evidence of DNA at the midbody in ICRF-treated cells by CLEM; however, this DNA appears broken before abscission in both cases and could not have been derived from premature abscission.

9. In Fig. 6, the authors examine actin patches in PARI-depleted and control cells as a marker of abscission. Although a role for PARI in actin patch formation would be very interesting, I'm not sure how it fits with the present story. The actin inside the intercellular canal described by Bai et al (removal of which correlates with abscission) appears very different to the accumulations of actin at the base of the intercellular canal described by Steigemann et al and by Dandoulaki et al. I can definitely see actin patches (similar to the ones in Steigemann et al) in Fig. 6 NT/ICRF, but I can't see any at the other treatments (I disagree with the arrows). Incidentally, I can see a DNA bridge only in NT/ICRF, but not in the other treatments.

10. Midbody resolutions are clearer in Fig. 7, perhaps with the exception of siPARI/DMSO.

However, no DNA bridges are visible, raising again the possibility that the authors investigate effects in cytokinesis without DNA bridges.

11. Can the authors investigate whether the helicase activity of PARI is required for the abscission checkpoint, by depletion-reconstitution experiments with a helicase-mutant protein?

12. The authors should investigate localization of PARI to the midbody/ DNA bridge in cytokinesis with chromatin bridges. Recent reports have proposed that a Top2-MRN-ATM-Chk2 pathway activates the Aurora B-dependent abscission checkpoint in human cells (PMIDs: 37638884, 33355621). The authors should examine localization of Aurora B and some of the above proteins in control and PARI-deficient cells to establish if/how PARI fits in the above pathway.

13. The authors use ICRF to generate chromatin bridges. If ICRF is continuously present in their assays, one would expect it to inhibit Top2 and impair the abscission checkpoint (PMIDs: 37638884, 33355621). How do the authors reconcile this with their proposed model?

****Additional comments:****

14. Page 8: "Although SIM-defective Srs2 has a lower affinity to SUMOylated PCNA, it can still interact with PCNA". The authors should test this experimentally or provide appropriate references supporting this claim.

15. Page 6: "Deletion of SRS2 further increased the fraction of anaphase cells with RPA foci, rising to approximately 30% in the absence of HU..."; however, this rise was not statistically significant as indicated in Fig. 1C.

16. Fig. 1C, D: SDs are missing. Fig. 1E: please show the p-values.

17. Fig. 2D: please show SDs and individual values.

18. Why do the authors show the spindle pole body in their movies?

19. Fig. 4A: WT and top2-4 cells have the same symbol in the graph.

2. Significance:

Significance (Required)

***Strengths:** potentially novel regulator of the abscission checkpoint. Timely and interesting topic of broad scientific interest.

***Limitations:** problems with quality of some data and with the interpretation. Also, more mechanistic evidence is required to significantly advance our knowledge in the field.

3. How much time do you estimate the authors will need to complete the suggested revisions:

Estimated time to Complete Revisions (Required)

(Decision Recommendation)

More than 6 months

4. Review Commons values the work of reviewers and encourages them to get credit for their work. Select 'Yes' below to register your reviewing activity at Web of Science Reviewer Recognition Service (formerly Publons); note that the content of your review will not be visible on Web of Science.

No

Review #3

1. Evidence, reproducibility and clarity:

Evidence, reproducibility and clarity (Required)

****Summary:**** Building on the specific connection between DNA bridges that bear marks of replication stress and the NoCut checkpoint (Amaral 2016, 2017), which prevents completion of cytokinesis, Dam et al. test the helicase Srs2/PARI for a role in this checkpoint pathway. The authors have produced a thorough study investigating the role of this helicase in both yeast and mammalian cells in the presence of DNA bridges. The manuscript includes clear evidence that Srs2 is important to resolve chromatin bridges, remove replication protein A (RPA) from chromatin, and delay cytokinesis under replication stress. Further, the authors show that loss of Srs2 under replication stress increases DNA damage, marked by elevated MRE11 foci in a manner dependent on cytokinesis (i.e., dependent on Cyk3). Srs2 deletion also partially abrogates the abscission delay seen upon topo-II inactivation. They further report that Srs2 must interact with PCNA to delay abscission in *S. cerevisiae*. While chromatin bridges formed when a dicentric chromosome is present escape detection by the NoCut checkpoint, inactivation of Elg1, which unloads PCNA and associated factors following DNA replication, results in delayed abscission. In HeLa cells, the Srs2 ortholog PARI is shown to similarly help promote abscission delay in the presence of DNA bridges following topoisomerase inhibition, as loss of PARI through siRNA knockdown prevents this abscission delay. Mechanistically, when PARI levels are reduced in HeLa cells, actin patches that function to stabilize the midbody and protect DNA bridges do not form/persist robustly as in cells with intact PARI. Consistent with a specific role in sensing the presence of a DNA bridge, depletion of PARI did not impact

abscission checkpoint activity in response to depletion of the NPC component, Nup153. Finally, the authors show that PARI depletion reduced time to abscission to the same extent as treatment with an Aurora B inhibitor, and PARI depletion in conjunction with Aurora B inhibition did not reduce abscission timing further than singular treatments, suggesting that PARI works within the Aurora B-mediated NoCut signaling cascade.

****Major comments:**** The manuscript is well written and, in general, the conclusions are thoroughly supported, but there are a few recommendations for addition or revision. The first of these is for a more thorough introduction of helicases potentially involved in cytokinesis and more clear rationale for why the focus is on Srs2.

Figure 1 E lacks statistical analysis. In addition, the text referring to 1E leads to confusion because the distinction between "RPA foci during anaphase" and "RPA coated chromatin bridges" is not made clear. The authors should clarify that the data presented in 1E shows quantification of cells with RPA foci during anaphase, not RPA coated chromatin bridges, and use consistent wording between the text and figure/figure legend. Further, how cells with RPA foci were identified, and what is classified as an RPA focus from images should be described in the methods.

In some cases, it is unclear whether DNA bridge formation is prevented vs aberrantly broken. For example, under Top2 inactivation, does the absence of Srs2 prevent bridge formation or promote their breakage along with premature midbody abscission? Confirming the frequency of chromatin bridge formation would address this and, further, monitoring RPA persistence would validate whether RPA clearance from bridges is consistently correlated with Srs2 activity (an interesting observation from Figure 1 that is not followed up on). Similarly, other conditions that appear to interfere with abscission delay (e.g., disrupting Srs2-PCNA interaction) should be monitored for whether the formation of DNA bridges has been altered.

In Figure 4A, the data show that the PIP-box is required for timely abscission. Imaging data from yeast strains with the PIP-box deletion alone should be included, rather than only showing the deletion in combination with the SIM deletion.

While the authors state that PARI and PCNA were not detectable at bridges in mammalian cells, it would be worth examining whether RPA is persistent on DNA bridges in mammalian cells depleted of PARI to understand how closely this pathway resembles the features found in yeast.

In Figure 6, the authors should describe in the methods how cells with actin patches were identified and quantified and explain what criteria must be met to be identified as an actin patch. Actin patches were described as "disassembling more quickly" in PARI-depleted cells, but the images look as if actin patches are not forming properly in these cells. The images are crisp and clear, but a change in wording may be necessary to accurately describe the data.

Minor suggestions to improve the manuscript are:

Include a diagram that shows hallmarks of cell division and what is being tracked in particular assays (e.g., DNA bridge duration vs time to abscission).

In the elegant CLEM experiments presented in Figure 5, organelle labels could be added to orient the readers.

The data in supplemental Figure 2 should be moved to Figure 5. The fact that there are similar levels of chromatin bridges is vital information and stresses that the defect lies in detection and response to the bridge as opposed to formation of bridges when PARI is depleted.

2. Significance:

Significance (Required)

The link between DNA bridges and NoCut/abscission checkpoint signaling is a fundamental aspect of cell cycle regulation. This manuscript makes a significant contribution to our understanding of this pathway by introducing a novel role for the helicase Srs2/PARI in execution of an abscission delay in the presence of DNA bridges. This is an important contribution as there is sparse information about cellular factors that mediate detection and response to DNA bridges, which is vital to protecting genome integrity. Although, as the authors themselves state, "the molecular mechanisms by which Srs2 and PARI function in NoCut remain unclear," this study, with some revisions, merits publication as it reveals a conserved role for a factor in this important response pathway and provides new insights into why certain DNA bridges (i.e., bridges formed by dicentric chromosomes) are not recognized by the NoCut pathway.

3. How much time do you estimate the authors will need to complete the suggested revisions:

Estimated time to Complete Revisions (Required)

(Decision Recommendation)

Between 1 and 3 months

No

We thank the three reviewers for their insightful and constructive comments, which have helped improve the manuscript. Our replies to each comment are provided below in blue.

Reviewer #1

Evidence, reproducibility and clarity

The abscission checkpoint, also known as NoCut, is a genome protection mechanism that remains poorly understood. This pathway is conserved from yeast to humans and protects the genome against chromosome bridges, a dangerous missegregation event that can have catastrophic consequences on genome stability. Dam et al now report the role of Srs2, a DNA helicase, as a key factor in the abscission checkpoint. The authors establish Srs2 as bona fide factor in this pathway by showing its involvement in abscission delays when chromatin bridges are induced. Importantly, yeast defective for Srs2 show increased levels of DNA damage when the frequency of chromatin bridges is increased. The authors also provide genetic evidence supporting a model whereby the interaction of Srs2 with PCNA is required for abscission regulation. In the second part of the manuscript, the authors study the human homologue of SRS2, PARI, in abscission regulation. The manuscript provides convincing evidence that PARI is also required for abscission delays in the presence of chromatin bridges. Critically, this role is specific for chromosome missegregation as abscission delays in response to nucleoporin depletion remain intact in PARI-depleted cells. Thus there is a conserved requirement for these DNA helicases in the abscission checkpoint.

Overall, these are important advances in our understanding of the abscission checkpoint. The data is high quality and convincing in general. However, the impact of PARI depletion on genome stability needs to be further demonstrated to support key claims in the manuscript. Specifically:

Disruptions of the abscission checkpoint in human cells result in bi-nucleation or increased levels of DNA damage. In this context, the authors need to show that PARI-depleted cells with increased frequency of chromatin bridges exhibit increased levels of bi-nucleation, DNA damage or both.

We thank the reviewer for its positive assessment of our work. While our data establish that Srs2 inhibits abscission to prevent DNA damage in yeast, we agree with the reviewer that we have not tested the consequences of PARI loss on DNA damage or cytokinesis failure in HeLa cells. We will address this in the revised version of our study.

Significance

The abscission checkpoint, remains poorly understood. There is evidence in the literature that disruptions in this pathway increase susceptibility to cancer. The identification of the Srs2/PARI helicases as key components in this pathway is a considerable step forward in this field.

Reviewer #2

Evidence, reproducibility and clarity

The Aurora B-mediated abscission checkpoint ("NoCut" in yeast) prevents tetraploidization or chromatin breakage in the presence of chromatin bridges in cytokinesis and the mechanisms of its activation are a matter of active investigation. In the present study, Dam et al propose that the conserved Srs2/PARI DNA helicase is required for the activation of the abscission checkpoint in response to chromatin bridges generated by DNA replication stress or topoisomerase inhibition. This is a timely and very interesting topic and the potential identification of a novel regulatory protein that activates the abscission checkpoint would be important. However, in my opinion, some Figures are of relatively low quality and need improving, there are apparent discrepancies between data and important control experiments are missing, which preclude the reader from fully evaluating the conclusions of this study. Some direct evidence of the role of Srs2/PARI on DNA bridges is also required. Also, it would be nice to investigate mechanistic details of the potential Srs2/PARI functions in the abscission checkpoint, and how it fits with other recently published signaling pathways that activate the abscission checkpoint in cytokinesis.

Specific comments:

1. The DNA channel (Ht2B-mCherry) in Figure 1A is of very low quality to be able to verify the authors interpretations of when the individual chromatin bridges are resolved (probably broken). For example, in the WT movie, they claim that the bridge is intact in frames 10 min and 14 min (yellow arrow) and that the bridge is resolved at 16 min (asterisk); however, I'm not convinced this is the case, because I can only see a very small portion of the bridge already at the 10 min and 14 min time-points. In my opinion, this bridge could have been broken much earlier, probably at 10 min. Also, WT +HU, is this bridge really intact at 10 min and at 14 min? In Srs2 Δ + HU, the bridge appears broken to me much earlier, perhaps at 30 min. There is a distinct possibility that the authors could not calculate the resolution times accurately from these movies (please also see my next comment, #2). The authors could perhaps use a more sensitive bridge marker such as GFP-BAF.

To clarify our approach, chromosome segregation was considered complete only when bridges were no longer detectable, while discontinuous or faint bridges were still classified as unresolved, as stretched DNA may result in weak nucleosome signals. This definition aligns with the bridge resolution times reported in Figure 1B-E. To improve clarity, we have revised the Results section to specify our classification criteria, and added all frames from the time-lapse movies in Figure 1A as a new figure (Supplementary Figure S1).

2. In Figure 1B, they conclude that Srs2 Δ cells treated with HU exhibit increased time from anaphase onset to bridge resolution compared with WT or Srs2 Δ cells. This result appears at odds with data from Fig. 2C showing that Srs2 Δ +HU finish abscission at similar times to WT or Srs2 Δ cells as judged by plasma membrane morphology. (final cut). Given that the final cut of the plasma membrane should cause chromatin bridges to break, if Srs2 is required for an

abscission delay in response to HU-induced chromatin bridges, I would expect *Srs2Δ* + HU cells to exhibit accelerated plasma membrane cut and also faster chromatin bridge resolution compared with controls. This discrepancy could at least in part be caused by the relatively low quality of movies used for the calculations in Fig. 1.

This is a perceptive point. To clarify, we analyzed the timing of chromosome segregation, membrane ingression at the abscission site, and abscission relative to anaphase onset, as shown in the new Supplementary Figure S2. In HU-treated cells (both WT and *srs2Δ*), bridge resolution and membrane ingression occur around the same time (~10 minutes after anaphase onset), with *srs2Δ* cells exhibiting slightly earlier membrane contraction. This suggests that bridges resolve during cytokinesis (see also our reply to the next comment) but does not distinguish whether they break prematurely or resolve normally. Our key finding is that membrane abscission is delayed in HU-treated cells in an *Srs2*-dependent manner, raising the question of whether this delay is important to prevent bridge breakage. This hypothesis is tested and supported by Figure 2D, where delaying cytokinesis (via *cyk3Δ*) reveals the protective role of *Srs2*.

3. Fig. 2 shows faster abscission times (membrane cut) in *Srs2Δ*+HU cells compared with WT+HU. The authors interpret this data as evidence for a role of *Srs2* in abscission delay in response to HU-induced chromatin bridges (page 7 and elsewhere). However, there is no direct evidence that the cells analyzed in Fig.2 exhibited DNA bridges in cytokinesis. One could argue that HU-induced DNA replication stress caused DNA lesions at the nuclear chromatin, which affected completion of cytokinesis in the absence or presence of *Srs2*. What proportion of HU-treated cells in cytokinesis exhibit DNA bridges? Judging from Fig. 1D this could be as low as 0-20%. The authors should analyze HU-treated cells that clearly exhibit DNA bridges, either by live-cell imaging or in fixed cells experiments. As it stands and together with my previous comments #1 and 2, I'm not convinced this data fully supports a role for *Srs2* in the abscission delay in response to HU-induced DNA bridges.

We appreciate the reviewer's concern. The presence of chromatin bridges in HU-treated cells during cytokinesis (membrane ingression) is documented in the new Supplementary Figure S2, as noted in our response to the previous comment. Additionally, our previous study (Amaral 2016, PMID: 27111841, Figure 1D) demonstrated that under the same HU treatment conditions used here, >90% of wild-type cells exhibit chromatin bridges during cytokinesis. This strongly supports the conclusion that the effects observed in Figure 2 are linked to the presence of DNA bridges.

4. In Fig. 2D, there is no evidence to support that *Mre11* foci are caused by bridge breakage, and not by replication-stress induced DNA lesions at the main nucleus (no DNA bridge is evident, also see comment #3).

The use of the *cyk3* mutant in Figure 2D specifically addresses this concern. If *Mre11* foci resulted from replication stress-induced lesions in the main nucleus, delaying cytokinesis should have no impact on damage levels. However, we observe that delaying cytokinesis via the *cyk3* mutation significantly reduces *Mre11* foci, strongly suggesting that these foci arise from chromatin bridge breakage rather than replication stress, and that delaying cytokinesis provides extra time to solve the chromosome segregation problem. This conclusion is further supported by previous studies showing that *cyk3Δ* delays cytokinesis

(Amaral 2016, PMID: 27111841, Figure 2C; Onishi et al. 2013, PMID: 23878277). We have clarified this point in the revised text.

5. Figure 3: the authors use a *top2-4* mutant strain to generate DNA bridges from catenated DNA and investigate the potential role of Srs2 in the abscission delay. However, no DNA bridges are obvious in the cells shown in Fig. 3. What proportion of *top2-4* mutant cells in cytokinesis exhibit DNA bridges? Does this explain the striking difference in the percentage of cells that haven't completed abscission after 30-60 min in WT+HU vs *Top2-4* cells? Please also see my previous comments above.

The *top2-4* mutant is well-characterized, and under the conditions used here, 100% of cells exhibit DNA bridges during cytokinesis (see for example Amaral et al., 2016, Figure 3A). We have clarified this point in the revised text. Notably, previous work has shown that *top2-4*-induced bridges are thicker and more persistent than those caused by HU-induced replication stress. This difference might contribute to the more severe abscission defect observed in *top2-4* cells, though we have not directly tested this.

6. The authors propose that association of Srs2 with PCNA is required for complete inhibition of abscission in *top2-4* mutant cells with chromatin bridges. Assuming a role for Srs2 in abscission timing in cytokinesis with chromatin bridges is fully proven, it is essential that the authors also investigate the localization of Srs2 and PCNA on chromatin bridges, using GFP-tagged proteins or appropriate antibodies in fixed and/or living cells. This would suggest a direct role of these proteins on chromatin bridges and considerably strengthen the authors hypothesis. Alternatively, Srs2 and PCNA may indirectly affect abscission timing through their well-established roles at nuclear chromatin.

The perturbations used in Figure 4 have been previously shown to disrupt Srs2-PCNA and PCNA-chromatin interactions (Armstrong et al., 2012; Ayyagari et al., 1995; Johnson et al., 2016; Kubota et al., 2013), as referenced in our manuscript. Given this well-established evidence, we believe additional imaging experiments would be redundant. Moreover, we do not claim that Srs2 or PCNA must specifically localize to chromatin bridges for NoCut function. Instead, our data demonstrate their genetic requirement for abscission inhibition in the presence of bridges. Whether these proteins localize exclusively on bridges or more broadly on chromatin remains unresolved, a point we explicitly discuss in the manuscript.

7. In Fig. 4D, the authors show an abscission delay in *elg1Δ* mutant cells in the presence of dicentric bridges compared with cytokinesis without bridges and interpret this as evidence that artificially retaining PCNA on dicentric chromatin bridges is sufficient to inhibit abscission. It is important that the authors demonstrate that PCNA localizes to dicentric bridges in *elg1Δ* mutant, but not in *ELG1* control, cells, e.g., by immunofluorescence, to support their claim and their proposed model.

As noted in our previous response, the association of PCNA with chromatin throughout the cell cycle and its regulation by *Elg1* have been extensively characterized in prior studies. Given this established evidence, additional imaging experiments would be redundant.

We also clarify that we do not claim that PCNA is specifically retained on chromatin bridges in *elg1Δ* mutants. Rather, our model is based on the overall retention of PCNA on chromatin in *elg1Δ* cells, as demonstrated in published studies.

Notably, *elg1Δ* mutants without dicentric bridges retain PCNA on chromatin but do not exhibit delayed abscission. However, only *elg1Δ* mutants with chromatin bridges inhibit abscission, indicating that PCNA retention alone is not sufficient—it is the presence of a bridge with retained PCNA that is critical. This distinction has been clarified in the revised manuscript.

8. In Fig. 5, the authors claim that HeLa cells treated with the Top2 inhibitor ICRF193 exhibit delayed midbody resolution compared with controls and that depletion of PARI by siRNA accelerates abscission in ICRF-treated cells. They interpret this as evidence for a role of PARI in the abscission delay in response to ICRF-induced chromatin bridges. However, no bridges are visible at any time-frame in cells in Fig. 5B raising the possibility that the observed time-differences are due to some effect of ICRF in cytokinesis without bridges. I'm also not convinced that in Fig. 5B the midbodies in NT/ICRF/230 min, siPARI/DMSO/110 min and siPARI/ICRF/150 min were resolved as indicated by the authors, as I can definitely see both midbody arms very clearly in these photos. The p-values are also just below the $p < 0.05$ threshold, which could in part be due to the quality of the movies quantified. Also, in Fig. 5C, the authors show evidence of DNA at the midbody in ICRF-treated cells by CLEM; however, this DNA appears broken before abscission in both cases and could not have been derived from premature abscission.

We acknowledge that the chromatin bridges in Figure 5B are challenging to visualize and may appear discontinuous. This is not due to poor image quality but likely reflects the low chromatin density of these structures. To clarify this, we now include magnified and contrast-enhanced images to better highlight the bridges, and quantification in Fig. 5C. Additionally, in the revised manuscript, we will provide new images using GFP-BAF, which directly binds DNA, to more clearly demonstrate the presence of chromatin bridges in ICRF-treated cells. These data will confirm that most cytokinetic cells in ICRF-treated conditions exhibit bridges.

Regarding the midbodies shown in Figure 5B, the presence of one or both arms intact does not indicate unresolved abscission but rather that the midbody has been severed, a distinction we explicitly describe in the manuscript.

Concerning the statistical analysis, we note that the p-value threshold of 0.05 is a widely accepted convention for statistical significance, and we have applied it appropriately in our analysis.

Finally, regarding the EM images in Figure 5C, these are single-section images, which do not allow us to determine definitively whether the bridges are physically broken when they appear discontinuous. It is possible that portions of the bridge extend outside the sectioned image. Regardless, we do not claim that these bridges are intact or broken. Rather, our key conclusion is that their presence at the abscission site in ICRF-treated cells is not affected by PARI knockdown, supporting our model.

9. In Fig. 6, the authors examine actin patches in PARI-depleted and control cells as a marker of abscission. Although a role for PARI in actin patch formation would be very interesting, I'm not sure how it fits with the present story. The actin inside the intercellular canal described by Bai et al (removal of which correlates with abscission) appears very different to the accumulations of actin at the base of the intercellular canal described by Sreigemann et al and by Dandoulaki et al. I can definitely see actin patches (similar to the ones in Steigemann et al) in Fig. 6 NT/ICRF, but I can't see any at the other treatments (I disagree with the arrows). Incidentally, I can see a DNA bridge only in NT/ICRF, but not in the other treatments.

We have revised our description of this figure for greater clarity. In control cells, actin accumulates at the cleavage furrow during anaphase and gradually disperses (clears) as cytokinesis progresses. We do not see patches in untreated cells, and we have updated the y-axis label in Figure 5B from “% of cells with actin patches” to “% of cells with actin clearance” to better reflect our observations.

Actin patches were observed only in ICRF-193-treated cells and were often associated with chromatin bridges. Cells that successfully disassembled these actin patches were classified as having completed actin clearance. Our data indicate that PARI depletion increases the fraction of cells that clear chromatin from the division plane, facilitating actin patch disassembly.

The actin patches observed in our study closely resemble those reported by Steigemann et al., and notably, we used the same cell line as in that study. Regarding Bai et al., they used both phalloidin and actin-GFP. For example, Figure 5C in Bai et al., shows examples of both actin patches near chromatin bridges, which resemble those in our study, and filamentous actin structures within the intercellular canal, which appear distinct.

Finally, a bridge fragment lacking actin patches is visible in PARI knockdown cells treated with ICRF, and we have now highlighted this in the revised figure.

10. Midbody resolutions are clearer in Fig. 7, perhaps with the exception of siPARI/DMSO. However, no DNA bridges are visible, raising again the possibility that the authors investigate effects in cytokinesis without DNA bridges.

See our response to point 8: while bridges are difficult to visualize, our analysis confirms that ICRF treatment induces bridges that persist during cytokinesis.

11. Can the authors investigate whether the helicase activity of PARI is required for the abscission checkpoint, by depletion-reconstitution experiments with a helicase-mutant protein?

PARI lacks detectable Walker motifs and associated ATPase activity, suggesting PARI lacks helicase activity (Moldovan et al., 2012). Therefore, we have not pursued depletion-reconstitution experiments with a helicase-mutant protein.

12. The authors should investigate localization of PARI to the midbody/ DNA bridge in cytokinesis with chromatin bridges. Recent reports have proposed that a Top2-MRN-ATM-Chk2 pathway activates the Aurora B-dependent abscission checkpoint in human cells (PMIDs:

37638884, 33355621). The authors should examine localization of Aurora B and some of the above proteins in control and PARI-deficient cells to establish if/how PARI fits in the above pathway.

As noted in our manuscript, we attempted to visualize PARI at midbodies and DNA bridges but were unable to detect any signal. This could be due to either its absence in these regions or its low concentration, making detection challenging.

We agree that investigating the Top2-MRN-ATM-Chk2 pathway in this context is important. We will examine the localization of key pathway components, including Aurora B, in control and PARI-deficient cells, and include the results in the revised manuscript.

13. The authors use ICRF to generate chromatin bridges. If ICRF is continuously present in their assays, one would expect it to inhibit Top2 and impair the abscission checkpoint (PMIDs: 37638884, 33355621). How do the authors reconcile this with their proposed model?

This is an important point. Studies from the Zachos lab have shown that Topoisomerase II α -DNA covalent complexes (Top2ccs) accumulate near the midbody in cells with chromatin bridges and play a key role in initiating abscission checkpoint signaling by recruiting MRN, ATM, and Aurora B. Supporting this model, ICRF-193 treatment does not alter midbody disassembly timing in HeLa cells, as shown in Petsalaki et al., 2023 (Figure S4D).

However, our results indicate that ICRF-193-treated HeLa cells exhibit delayed midbody severing, suggesting that at least some aspects of abscission checkpoint signaling remain active under these conditions. One possible explanation for this discrepancy is the difference in ICRF-193 concentration: our study uses a low dose (250 nM) versus 10 μ M in the Zachos group study. We favor the hypothesis that this lower dose preserves sufficient Top2 activity to support some level of checkpoint signaling while still effectively generating chromatin bridges.

Additional comments:

14. Page 8: "Although SIM-defective Srs2 has a lower affinity to SUMOylated PCNA, it can still interact with PCNA". The authors should test this experimentally or provide appropriate references supporting this claim.

We have clarified our statement and provided the reference: Although SIM-defective Srs2 has a lower affinity to SUMOylated PCNA, it can still interact with non-SUMOylated PCNA (Armstrong et al. 2012).

15. Page 6: "Deletion of SRS2 further increased the fraction of anaphase cells with RPA foci, rising to approximately 30% in the absence of HU..."; however, this rise was not statistically significant as indicated in Fig. 1C.

Thank you for noting this - we have removed this statement.

16. Fig. 1C, D: SDs are missing. Fig. 1E: please show the p-values.

These data in Figures 1C-D represent percentages from cells pooled from two independent experiments with similar results. P-values were calculated using Dunn's multiple comparison test. Standard deviations are not applicable in this case. We have included the p-values for Figure 1E.

17. Fig. 2D: please show SDs and individual values.

These data represent percentages from cells pooled from independent experiments with similar results. P-values were calculated using Fisher's exact test. Standard deviations and individual values are not applicable in this case.

18. Why do the authors show the spindle pole body in their movies?

We do this to infer the time of anaphase onset; see our response to points 1-3 and Fig. S2.

19. Fig. 4A: WT and top2-4 cells have the same symbol in the graph.

We have changed the symbols.

Significance

Strengths: potentially novel regulator of the abscission checkpoint. Timely and interesting topic of broad scientific interest.

Limitations: problems with quality of some data and with the interpretation. Also, more mechanistic evidence is required to significantly advance our knowledge in the field.

Reviewer #3

Evidence, reproducibility and clarity:

Summary: Building on the specific connection between DNA bridges that bear marks of replication stress and the NoCut checkpoint (Amaral 2016, 2017), which prevents completion of cytokinesis, Dam et al. test the helicase Srs2/PARI for a role in this checkpoint pathway. The authors have produced a thorough study investigating the role of this helicase in both yeast and mammalian cells in the presence of DNA bridges. The manuscript includes clear evidence that Srs2 is important to resolve chromatin bridges, remove replication protein A (RPA) from chromatin, and delay cytokinesis under replication stress. Further, the authors show that loss of Srs2 under replication stress increases DNA damage, marked by elevated MRE11 foci in a manner dependent on cytokinesis (i.e., dependent on Cyk3). Srs2 deletion also partially abrogates the abscission delay seen upon topo-II inactivation. They further report that Srs2 must interact with PCNA to delay abscission in *S. cerevisiae*. While chromatin bridges formed when a dicentric chromosome is present escape detection by the NoCut checkpoint, inactivation of Elg1, which unloads PCNA and associated factors following DNA replication, results in

delayed abscission. In HeLa cells, the Srs2 ortholog PARI is shown to similarly help promote abscission delay in the presence of DNA bridges following topoisomerase inhibition, as loss of PARI through siRNA knockdown prevents this abscission delay. Mechanistically, when PARI levels are reduced in HeLa cells, actin patches that function to stabilize the midbody and protect DNA bridges do not form/persist robustly as in cells with intact PARI. Consistent with a specific role in sensing the presence of a DNA bridge, depletion of PARI did not impact abscission checkpoint activity in response to depletion of the NPC component, Nup153. Finally, the authors show that PARI depletion reduced time to abscission to the same extent as treatment with an Aurora B inhibitor, and PARI depletion in conjunction with Aurora B inhibition did not reduce abscission timing further than singular treatments, suggesting that PARI works within the Aurora B-mediated NoCut signaling cascade.

Major comments: The manuscript is well written and, in general, the conclusions are thoroughly supported, but there are a few recommendations for addition or revision.

1. The first of these is for a more thorough introduction of helicases potentially involved in cytokinesis and more clear rationale for why the focus is on Srs2.

We appreciate the reviewer's suggestion and have expanded the introduction to better contextualize helicases in cytokinesis and clarify our focus on Srs2.

2. Figure 1 E lacks statistical analysis. In addition, the text referring to 1E leads to confusion because the distinction between "RPA foci during anaphase" and "RPA coated chromatin bridges" is not made clear. The authors should clarify that the data presented in 1E shows quantification of cells with RPA foci during anaphase, not RPA coated chromatin bridges, and use consistent wording between the text and figure/figure legend. Further, how cells with RPA foci were identified, and what is classified as an RPA focus from images should be described in the methods.

We appreciate the reviewer's feedback. In the revised manuscript, we have included statistical analysis for Figure 1E and clarified the distinction between "RPA foci during anaphase" and "RPA-coated chromatin bridges" to ensure consistency. Additionally, we have updated the Methods section to specify how cells with RPA foci were identified and what criteria were used to classify RPA foci based on the imaging data.

3. In some cases, it is unclear whether DNA bridge formation is prevented vs aberrantly broken. For example, under Top2 inactivation, does the absence of Srs2 prevent bridge formation or promote their breakage along with premature midbody abscission? Confirming the frequency of chromatin bridge formation would address this and, further, monitoring RPA persistence would validate whether RPA clearance from bridges is consistently correlated with Srs2 activity (an interesting observation from Figure 1 that is not followed up on). Similarly, other conditions that appear to interfere with abscission delay (e.g., disrupting Srs2-PCNA interaction) should be monitored for whether the formation of DNA bridges has been altered.

We agree this is important and will address it in a full revision. We will quantify chromatin bridge formation under Top2 inactivation to determine whether Srs2 mutations affect bridge frequency or stability. Additionally, we will monitor RPA persistence in *top2* cells to assess whether RPA clearance correlates with Srs2 activity. While we find it unlikely that bridge formation is prevented by *srs2* mutations, as Top2 is essential for decatenation, our experiments will directly test this possibility.

4. In Figure 4A, the data show that the PIP-box is required for timely abscission. Imaging data from yeast strains with the PIP-box deletion alone should be included, rather than only showing the deletion in combination with the SIM deletion.

We agree with the reviewer's suggestion, and will include imaging data from yeast strains with the PIP-box deletion alone in the revised manuscript.

5. While the authors state that PARI and PCNA were not detectable at bridges in mammalian cells, it would be worth examining whether RPA is persistent on DNA bridges in mammalian cells depleted of PARI to understand how closely this pathway resembles the features found in yeast.

Here too, we agree with the reviewer's suggestion, and will include imaging data from HeLa cells visualizing RPA in the revised manuscript.

6. In Figure 6, the authors should describe in the methods how cells with actin patches were identified and quantified and explain what criteria must be met to be identified as an actin patch. Actin patches were described as "disassembling more quickly" in PARI-depleted cells, but the images look as if actin patches are not forming properly in these cells. The images are crisp and clear, but a change in wording may be necessary to accurately describe the data.

Thank you for pointing this out. We agree that the wording was confusing (see our reply to reviewer 2, comment 9) and have revised our description of this figure for greater clarity. In control cells, actin accumulates at the cleavage furrow during anaphase and gradually disperses (clears) as cytokinesis progresses. We do not see patches in untreated cells, and we have updated the y-axis label in Figure 5B from "% of cells with actin patches" to "% of cells with actin clearance" to better reflect our observations. Actin patches were observed only in ICRF-193-treated cells and were often associated with chromatin bridges. Cells that successfully disassembled these actin patches were classified as having completed actin clearance. Our data indicate that PARI depletion increases the fraction of cells that clear chromatin from the division plane, facilitating actin patch disassembly.

Minor suggestions to improve the manuscript are:

7. Include a diagram that shows hallmarks of cell division and what is being tracked in particular assays (e.g., DNA bridge duration vs time to abscission).

Thank you for this suggestion, which we have implemented in Figure S2A.

8. In the elegant CLEM experiments presented in Figure 5, organelle labels could be added to orient the readers.

We added organelle labels to CLEM images.

9. The data in supplemental Figure 2 should be moved to Figure 5. The fact that there are similar levels of chromatin bridges is vital information and stresses that the defect lies in detection and response to the bridge as opposed to formation of bridges when PARI is depleted.

We agree, and have moved Figure S2 to Figure 5 (now Figure 5C).

Significance

The link between DNA bridges and NoCut/abscission checkpoint signaling is a fundamental aspect of cell cycle regulation. This manuscript makes a significant contribution to our understanding of this pathway by introducing a novel role for the helicase Srs2/PARI in execution of an abscission delay in the presence of DNA bridges. This is an important contribution as there is sparse information about cellular factors that mediate detection and response to DNA bridges, which is vital to protecting genome integrity. Although, as the authors themselves state, "the molecular mechanisms by which Srs2 and PARI function in NoCut remain unclear," this study, with some revisions, merits publication as it reveals a conserved role for a factor in this important response pathway and provides new insights into why certain DNA bridges (i.e., bridges formed by dicentric chromosomes) are not recognized by the NoCut pathway.

March 5, 2025

Re: JCB manuscript #202502014T

Manuel Mendoza
Institute of Genetics and Molecular and Cellular Biology

Dear Dr. Mendoza,

Thank you for submitting your manuscript entitled "Srs2/PARI DNA helicase mediates abscission inhibition in response to chromatin bridges in yeast and human cells." We have now assessed the manuscript, reports from Review Commons, and the revision plan. We invite you to submit a revision as outlined in your revision plan.

GENERAL GUIDELINES:

Text limits: Character count for an Article is < 40,000, not including spaces. Count includes title page, abstract, introduction, results, discussion, and acknowledgments. Count does not include materials and methods, figure legends, references, tables, or supplemental legends.

Figures: Articles may have up to 10 main text figures. Figures must be prepared according to the policies outlined in our Instructions to Authors, under Data Presentation, <https://jcb.rupress.org/site/misc/ifora.xhtml>. All figures in accepted manuscripts will be screened prior to publication.

Supplemental information: An Article may generally have up to 5 supplemental figures. Up to 10 supplemental videos or flash animations are allowed. A summary of all supplemental material should appear at the end of the Materials and methods section.

Please note that JCB now requires authors to submit Source Data used to generate figures containing gels and Western blots with all revised manuscripts. This Source Data consists of fully uncropped and unprocessed images for each gel/blot displayed in the main and supplemental figures. For assays performed using capillary electrophoresis and/or immunoassay-based detection, authors should instead provide the electropherogram graph(s) for each experiment, plotting fluorescence/chemiluminescence intensity vs. molecular weight/size. Since your paper includes cropped gel and/or blot images, please be sure to provide one Source Data file for each figure gels, blots, and/or capillary electrophoresis assays along with your revised manuscript files. File names for Source Data figures should be alphanumeric without any spaces or special characters (i.e., SourceDataF#, where F# refers to the associated main figure number or SourceDataFS# for those associated with Supplementary figures). For traditional gels and blots, the lanes of the gels/blots should be labeled as they are in the associated figure, the place where cropping was applied should be marked (with a box), and molecular weight/size standards should be labeled wherever possible. For capillary electrophoresis assays, each trace in the graph should be color-coded and labeled to indicate which protein, gene, or sample is being measured (please try to avoid red/green combinations to accommodate our color-blind readers).

The typical timeframe for revisions is three to four months. If you anticipate any difficulties in meeting this aforementioned revision time limit, please contact us and we can work with you to find an appropriate time frame for resubmission. Please note that papers are generally considered through only one revision cycle, so any revised manuscript will likely be either accepted or rejected.

Thank you for this interesting contribution to Journal of Cell Biology. You can contact us at the journal office with any questions at cellbio@rockefeller.edu.

Sincerely,

Karen Oegema, PhD
Monitoring Editor
Journal of Cell Biology

Dan Simon, PhD
Scientific Editor
Journal of Cell Biology

We thank the reviewers and the editor for their constructive feedback, which significantly helped us improve the manuscript. Below, we respond point-by-point to each comment. Revisions in the manuscript are highlighted in red for clarity. Our replies to each comment are provided below in blue.

Reviewer #1

Evidence, reproducibility and clarity

The abscission checkpoint, also known as NoCut, is a genome protection mechanism that remains poorly understood. This pathway is conserved from yeast to humans and protects the genome against chromosome bridges, a dangerous missegregation event that can have catastrophic consequences on genome stability. Dam et al now report the role of Srs2, a DNA helicase, as a key factor in the abscission checkpoint. The authors establish Srs2 as bona fide factor in this pathway by showing its involvement in abscission delays when chromatin bridges are induced. Importantly, yeast defective for Srs2 show increased levels of DNA damage when the frequency of chromatin bridges is increased. The authors also provide genetic evidence supporting a model whereby the interaction of Srs2 with PCNA is required for abscission regulation. In the second part of the manuscript, the authors study the human homologue of SRS2, PARI, in abscission regulation. The manuscript provides convincing evidence that PARI is also required for abscission delays in the presence of chromatin bridges. Critically, this role is specific for chromosome missegregation as abscission delays in response to nucleoporin depletion remain intact in PARI-depleted cells. Thus there is a conserved requirement for these DNA helicases in the abscission checkpoint.

Overall, these are important advances in our understanding of the abscission checkpoint. The data is high quality and convincing in general. However, the impact of PARI depletion on genome stability needs to be further demonstrated to support key claims in the manuscript. Specifically:

Disruptions of the abscission checkpoint in human cells result in bi-nucleation or increased levels of DNA damage. In this context, the authors need to show that PARI-depleted cells with increased frequency of chromatin bridges exhibit increased levels of bi-nucleation, DNA damage or both.

We thank the reviewer for its positive assessment of our work. We agree that the consequences of PARI depletion in human cells require further clarification. In the revised manuscript (**new Fig. 7, Movies 5-7**), we now show that siRNA-mediated PARI depletion does not lead to increased DNA bridge breakage, suggesting that it does not increase DNA damage. Instead, PARI knockdown significantly increases the lifetime of chromatin bridges (labelled with GFP-BAF) indicating that, like Srs2 in yeast, PARI contributes to resolution of DNA intermediates. In addition, we find that although PARI is required for bridge-dependent midbody and actin stabilization, its depletion does not lead to binucleation. This outcome contrasts with the effects of inhibiting core abscission checkpoint components such as Aurora B or CHMP4C, whose inactivation results in furrow regression or bridge breakage, respectively. This raises the possibility that PARI plays a supportive but non-essential role in the abscission checkpoint.

Alternatively, incomplete depletion of PARI may allow residual checkpoint activity. We mention this in the discussion (p. 22).

Significance

The abscission checkpoint, remains poorly understood. There is evidence in the literature that disruptions in this pathway increase susceptibility to cancer. The identification of the Srs2/PARI helicases as key components in this pathway is a considerable step forward in this field.

Reviewer #2

Evidence, reproducibility and clarity

The Aurora B-mediated abscission checkpoint ("NoCut" in yeast) prevents tetraploidization or chromatin breakage in the presence of chromatin bridges in cytokinesis and the mechanisms of its activation are a matter of active investigation. In the present study, Dam et al propose that the conserved Srs2/PARI DNA helicase is required for the activation of the abscission checkpoint in response to chromatin bridges generated by DNA replication stress or topoisomerase inhibition. This is a timely and very interesting topic and the potential identification of a novel regulatory protein that activates the abscission checkpoint would be important. However, in my opinion, some Figures are of relatively low quality and need improving, there are apparent discrepancies between data and important control experiments are missing, which preclude the reader from fully evaluating the conclusions of this study. Some direct evidence of the role of Srs2/PARI on DNA bridges is also required. Also, it would be nice to investigate mechanistic details of the potential Srs2/PARI functions in the abscission checkpoint, and how it fits with other recently published signaling pathways that activate the abscission checkpoint in cytokinesis.

Specific comments:

1. The DNA channel (Ht2B-mCherry) in Figure 1A is of very low quality to be able to verify the authors interpretations of when the individual chromatin bridges are resolved (probably broken). For example, in the WT movie, they claim that the bridge is intact in frames 10 min and 14 min (yellow arrow) and that the bridge is resolved at 16 min (asterisk); however, I'm not convinced this is the case, because I can only see a very small portion of the bridge already at the 10 min and 14 min time-points. In my opinion, this bridge could have been broken much earlier, probably at 10 min. Also, WT +HU, is this bridge really intact at 10 min and at 14 min? In Srs2 Δ + HU, the bridge appears broken to me much earlier, perhaps at 30 min. There is a distinct possibility that the authors could not calculate the resolution times accurately from these movies (please also see my next comment, #2). The authors could perhaps use a more sensitive bridge marker such as GFP-BAF.

To clarify our approach, chromosome segregation was considered complete only when bridges were no longer detectable, while discontinuous or faint bridges were still classified as unresolved, as stretched DNA may result in weak nucleosome signals. This definition aligns with the bridge resolution times reported in Figure 1. To improve clarity, we have revised the Results section to specify our classification criteria (p. 6), and added all frames from the time-lapse movies in Figure 1A to support our

quantifications (**new Supplementary Figure S1**). Regarding the distinction between bridges that resolve with or without brakes, see our response to Comment 3.

2. In Figure 1B, they conclude that *Srs2* Δ cells treated with HU exhibit increased time from anaphase onset to bridge resolution compared with WT or *Srs2* Δ cells. This result appears at odds with data from Fig. 2C showing that *Srs2* Δ +HU finish abscission at similar times to WT or *Srs2* Δ cells as judged by plasma membrane morphology. (final cut). Given that the final cut of the plasma membrane should cause chromatin bridges to break, if *Srs2* is required for an abscission delay in response to HU-induced chromatin bridges, I would expect *Srs2* Δ + HU cells to exhibit accelerated plasma membrane cut and also faster chromatin bridge resolution compared with controls. This discrepancy could at least in part be caused by the relatively low quality of movies used for the calculations in Fig. 1.

We address this and the following comment together, see our reply to Comment 3, below.

3. Fig. 2 shows faster abscission times (membrane cut) in *Srs2* Δ +HU cells compared with WT+HU. The authors interpret this data as evidence for a role of *Srs2* in abscission delay in response to HU-induced chromatin bridges (page 7 and elsewhere). However, there is no direct evidence that the cells analyzed in Fig.2 exhibited DNA bridges in cytokinesis. One could argue that HU-induced DNA replication stress caused DNA lesions at the nuclear chromatin, which affected completion of cytokinesis in the absence or presence of *Srs2*. What proportion of HU-treated cells in cytokinesis exhibit DNA bridges? Judging from Fig. 1D this could be as low as 0-20%. The authors should analyze HU-treated cells that clearly exhibit DNA bridges, either by live-cell imaging or in fixed cells experiments. As it stands and together with my previous comments #1 and 2, I'm not convinced this data fully supports a role for *Srs2* in the abscission delay in response to HU-induced DNA bridges.

As Comments 2 and 3 are closely related, we address them together. To clarify these points, we now provide a new schematic and quantifications (**new Supplementary Figure S2A-B**) illustrating the temporal sequence of key events shown in figures 1 and 2: chromatin bridge resolution, membrane ingression (driven by actomyosin ring contraction), and abscission. These data show, in the same timescale, that: (1) HU treatment delays bridge resolution relative to anaphase onset, such that bridges resolve concurrently with or shortly after membrane ingression; (2) In *srs2* Δ cells, membrane contraction occurs slightly earlier than in WT cells; and (3) Abscission occurs only after bridge resolution is complete. Thus, HU-induced bridges resolve during the window between membrane ingression and abscission (see our new text, **p. 8**). This conclusion is consistent with our previous result from Amaral et al., 2016, reproduced in the next page:

Figure 1d from Amaral et al., 2016: Kinetics of chromosome segregation (Htb2–mCherry) relative to actomyosin ring contraction (Myo1–GFP). WT cells at 30 °C were either untreated or treated with 100mM HU for 2h, and transferred to fresh media. The arrow points to a chromatin bridge, and the asterisk marks the time of chromosome segregation (separated nuclear masses) after HU treatment. Scale bars, 2 μ m. The graph shows the time of chromosome segregation relative to the onset of ring contraction (time 0). Boxes include 50% of data points, and whiskers are 90%. Median (lines) and mean (crosses) are shown. n = 28 untreated and 43 HU- treated cells.

However, these experiments alone do not distinguish between physiological bridge resolution (e.g., via DNA replication completion or decatenation) and pathological resolution (e.g., breakage leading to DNA damage). To address this, we analyzed the formation of Mre11 foci as a readout of DNA damage (Figure 2D). We find that *srs2 Δ* cells accumulate abscission-dependent Mre11 foci following HU treatment, whereas wild-type cells do not, indicating that in the absence of Srs2, bridge resolution is frequently associated with DNA damage. These results support our model that Srs2 is required to delay abscission in the presence of HU-induced chromatin bridges, thereby allowing non-damaging bridge resolution. Additional details about this experiment are provided in our response to Comment 4, below.

4. In Fig. 2D, there is no evidence to support that Mre11 foci are caused by bridge breakage, and not by replication-stress induced DNA lesions at the main nucleus (no DNA bridge is evident, also see comment #3).

The use of the *cyk3* mutant in Figure 2D specifically addresses this concern. If Mre11 foci resulted from replication stress-induced lesions in the main nucleus, delaying cytokinesis should have no impact on damage levels. However, we observe that delaying cytokinesis via the *cyk3* mutation significantly reduces Mre11 foci, strongly suggesting that these foci arise from chromatin bridge breakage rather than replication stress, and that delaying cytokinesis provides extra time to solve the chromosome segregation problem. This conclusion is further supported by previous studies showing that *cyk3 Δ* delays cytokinesis (Amaral et al., 2016, PMID: 27111841, Figure 2C; Onishi et al. 2013, PMID: 23878277). We have clarified this point in the revised text (p. 8).

5. Figure 3: the authors use a *top2-4* mutant strain to generate DNA bridges from catenated DNA and investigate the potential role of Srs2 in the abscission delay. However, no DNA bridges are obvious in the cells shown in Fig. 3. What proportion of *top2-4* mutant cells in cytokinesis exhibit DNA bridges? Does this explain the striking difference in the percentage of cells that

haven't completed abscission after 30-60 min in WT+HU vs Top2-4 cells? Please also see my previous comments above.

The *top2-4* mutant is well-characterized, and under the conditions used here, 100% of cells exhibit DNA bridges during cytokinesis (see for example Amaral et al., 2016, Figure 3A). We now include new imaging and quantifications replicating this result in the **new Figure 3A-B**. *top2-4*-induced bridges are thicker and more persistent than those caused by HU-induced replication stress. This difference might contribute to the more severe abscission defect observed in *top2-4* cells, though we have not directly tested this.

6. The authors propose that association of Srs2 with PCNA is required for complete inhibition of abscission in *top2-4* mutant cells with chromatin bridges. Assuming a role for Srs2 in abscission timing in cytokinesis with chromatin bridges is fully proven, it is essential that the authors also investigate the localization of Srs2 and PCNA on chromatin bridges, using GFP-tagged proteins or appropriate antibodies in fixed and/or living cells. This would suggest a direct role of these proteins on chromatin bridges and considerably strengthen the authors hypothesis. Alternatively, Srs2 and PCNA may indirectly affect abscission timing through their well-established roles at nuclear chromatin.

We address this and the following comment together, see our reply to Comment 7, below.

7. In Fig. 4D, the authors show an abscission delay in *elg1Δ* mutant cells in the presence of dicentric bridges compared with cytokinesis without bridges and interpret this as evidence that artificially retaining PCNA on dicentric chromatin bridges is sufficient to inhibit abscission. It is important that the authors demonstrate that PCNA localizes to dicentric bridges in *elg1Δ* mutant, but not in *ELG1* control, cells, e.g., by immunofluorescence, to support their claim and their proposed model.

We thank the reviewer for these suggestions. As Comments 6 and 7 are closely related, we address them together. We appreciate the value of localization data; however, our conclusions are based on a combination of genetic perturbations and well-established prior work. Specifically, previous studies have shown that the mutations used in our experiments disrupt PCNA–Srs2 and PCNA–chromatin interactions (Armstrong et al., 2012; Ayyagari et al., 1995; Johnson et al., 2016; Kubota et al., 2013). These findings are referenced in our manuscript. Given this well-characterized mechanistic basis, we believe that additional imaging experiments would be redundant.

Importantly, we do not claim that PCNA or Srs2 must specifically or exclusively localize to chromatin bridges in order to inhibit abscission. Rather, our model posits that their retention on chromatin bridges is sufficient to trigger abscission inhibition. This is best illustrated by our results in Figure 5, where we manipulate PCNA retention using the *ELG1* deletion. This mutation causes global retention of PCNA (and associated factors such as Srs2) on chromatin. Despite this, only cells with chromatin bridges exhibit abscission delay. *elg1Δ* mutants that do not form bridges complete abscission with normal timing, indicating that PCNA retention on chromatin alone is not sufficient. Instead, it is the co-occurrence of PCNA retention and a chromatin bridge that triggers abscission inhibition. We have clarified this interpretation explicitly in the revised Results section (**p. 12**) to avoid any misunderstanding about our claims.

8. In Fig. 5, the authors claim that HeLa cells treated with the Top2 inhibitor ICRF193 exhibit delayed midbody resolution compared with controls and that depletion of PARI by siRNA accelerates abscission in ICRF-treated cells. They interpret this as evidence for a role of PARI in the abscission delay in response to ICRF-induced chromatin bridges. However, no bridges are visible at any time-frame in cells in Fig. 5B raising the possibility that the observed time-differences are due to some effect of ICRF in cytokinesis without bridges. I'm also not convinced that in Fig. 5B the midbodies in NT/ICRF/230 min, siPARI/DMSO/110 min and siPARI/ICRF/150 min were resolved as indicated by the authors, as I can definitely see both midbody arms very clearly in these photos. The p-values are also just below the $p < 0.05$ threshold, which could in part be due to the quality of the movies quantified. Also, in Fig. 5C, the authors show evidence of DNA at the midbody in ICRF-treated cells by CLEM; however, this DNA appears broken before abscission in both cases and could not have been derived from premature abscission.

We improved image contrast in the **revised Fig. 6A** to illustrate how we quantified bridge presence, shown in Fig. 6B (previously figure S2 in the original manuscript). We also introduced GFP-BAF-based imaging in the **new Fig. 7A** and **Movies 5-7** to validate chromatin bridge presence in ICRF-treated cells.

Regarding quantification of midbody lifetime shown in Figure 5B (now Fig 6A), we appreciate the opportunity to clarify this point. In this figure, we assess midbody severing—also referred to as microtubule “cutting” (see Advedissian et al., 2024)—rather than midbody arm disassembly. This is stated on page 13 of the manuscript. We also provide movies of these cells showing all time points acquired (**Movies 1-4**). In contrast, midbody disassembly, which occurs several minutes after severing, was scored in the **new Figure 7C–D** and in Fig. S5D using SiR-Tubulin, which provides a weaker signal. This distinction is now noted on **page 15**.

Concerning the statistical analysis, we note that the p-value threshold of 0.05 is a widely accepted convention for statistical significance, and we have applied it appropriately in our analysis.

Finally, regarding the EM images in Figure 5C (now Figure 6D), these are single-section images, which do not allow us to determine definitively whether the bridges are physically broken when they appear discontinuous. It is possible that portions of the bridge extend outside the sectioned image. Regardless, we do not claim that these bridges are intact or broken. Rather, our key conclusion is that their appearance at the abscission site in ICRF-treated cells is not affected by PARI knockdown.

9. In Fig. 6, the authors examine actin patches in PARI-depleted and control cells as a marker of abscission. Although a role for PARI in actin patch formation would be very interesting, I'm not sure how it fits with the present story. The actin inside the intercellular canal described by Bai et al (removal of which correlates with abscission) appears very different to the accumulations of actin at the base of the intercellular canal described by Sreigemann et al and by Dandoulaki et al. I can definitely see actin patches (similar to the ones in Steigemann et al) in Fig. 6 NT/ICRF, but I can't see any at the other treatments (I disagree with the arrows). Incidentally, I can see a DNA bridge only in NT/ICRF, but not in the other treatments.

We agree that the original wording in this section was unclear (see also Reviewer 3, comment 6), and we have revised the corresponding text to improve clarity. Specifically, we updated the terminology to refer

to “actin clearance” and added a new analysis focused on actin patch dynamics. Given the high degree of variability in actin patch intensity observed between individual cells, we chose to quantify a representative subset of cells to illustrate the overall trends. These new results are presented in **Figure 9A-B, Movies 8-11** and are described in **p. 18 and p. 30**.

The actin patches observed in our study closely resemble those reported by Steigemann et al., as expected since we used the same cell line as in that study. Regarding Bai et al., they used both phalloidin and actin-GFP. For example, Figure 5C in Bai et al., shows examples of both actin patches near chromatin bridges, which resemble those in our study, and filamentous actin structures within the intercellular canal, which appear distinct.

10. Midbody resolutions are clearer in Fig. 7, perhaps with the exception of siPARI/DMSO. However, no DNA bridges are visible, raising again the possibility that the authors investigate effects in cytokinesis without DNA bridges.

We agree that chromatin bridges are difficult to see in these images (now Figure 10A), which are intended to show midbody morphology. We confirmed the presence of DNA bridges experimentally under these conditions in the **new Fig. 7A-B and Movies 5-7** using eGFP-BAF, a sensitive marker for bridges.

11. Can the authors investigate whether the helicase activity of PARI is required for the abscission checkpoint, by depletion-reconstitution experiments with a helicase-mutant protein?

PARI lacks detectable Walker motifs and associated ATPase activity, suggesting PARI lacks helicase activity (Moldovan et al., 2012). Therefore, we have not pursued depletion-reconstitution experiments with a helicase-mutant protein.

12. The authors should investigate localization of PARI to the midbody/ DNA bridge in cytokinesis with chromatin bridges. Recent reports have proposed that a Top2-MRN-ATM-Chk2 pathway activates the Aurora B-dependent abscission checkpoint in human cells (PMIDs: 37638884, 33355621). The authors should examine localization of Aurora B and some of the above proteins in control and PARI-deficient cells to establish if/how PARI fits in the above pathway.

As noted in our manuscript, we attempted to visualize PARI at midbodies and DNA bridges but were unable to detect any signal (see Supplementary Figure S5C). This could be due to either its absence in these regions or its low concentration, making detection challenging.

We examined the localization of pT232-Aurora B, Topo II α , and MRE11 in control and PARI-depleted cells (**new Fig. 10C-E**). While these proteins localize appropriately in both settings, midbody lifetime is shortened upon PARI depletion or Aurora B inhibition, and no additive effect is seen. This supports the idea that PARI acts in the same pathway as Aurora B.

13. The authors use ICRF to generate chromatin bridges. If ICRF is continuously present in their assays, one would expect it to inhibit Top2 and impair the abscission checkpoint (PMIDs: 37638884, 33355621). How do the authors reconcile this with their proposed model?

Thank you for bringing up this issue, which we had not commented on in our original manuscript. Our ICRF-193 concentration (250 nM) is ~40-fold lower than in studies reporting Top2 and checkpoint inhibition. At this low dose, Top2 activity may be partially retained, allowing bridge formation and checkpoint activation. This is discussed in the revised discussion section (**p. 23**).

Additional comments:

14. Page 8: "Although SIM-defective Srs2 has a lower affinity to SUMOylated PCNA, it can still interact with PCNA". The authors should test this experimentally or provide appropriate references supporting this claim.

We have clarified our statement and provided the reference: Although SIM-defective Srs2 has a lower affinity to SUMOylated PCNA, it can still interact with non-SUMOylated PCNA (Armstrong et al. 2012) (**p. 10**).

15. Page 6: "Deletion of SRS2 further increased the fraction of anaphase cells with RPA foci, rising to approximately 30% in the absence of HU..."; however, this rise was not statistically significant as indicated in Fig. 1C.

Thank you for noting this - we have removed this statement.

16. Fig. 1C, D: SDs are missing. Fig. 1E: please show the p-values.

These data in Figures 1C-D represent percentages from cells pooled from two independent experiments with similar results. P-values were calculated using Fisher's exact test. Standard deviations are not applicable in this case. We have included the p-values for **Figure 1E**.

17. Fig. 2D: please show SDs and individual values.

These data represent percentages from cells pooled from independent experiments with similar results. P-values were calculated using Fisher's exact test. Standard deviations and individual values are not applicable in this case.

18. Why do the authors show the spindle pole body in their movies?

We do this to infer the time of anaphase onset; see **new Supplementary Fig. S2**. We have clarified this in the results section (**p. 7**).

19. Fig. 4A: WT and top2-4 cells have the same symbol in the graph.

Symbol duplication in Fig. 4A (now Figure 4B) has been corrected.

Significance

Strengths: potentially novel regulator of the abscission checkpoint. Timely and interesting topic of broad scientific interest.

Limitations: problems with quality of some data and with the interpretation. Also, more mechanistic evidence is required to significantly advance our knowledge in the field.

Reviewer #3

Evidence, reproducibility and clarity:

Summary: Building on the specific connection between DNA bridges that bear marks of replication stress and the NoCut checkpoint (Amaral 2016, 2017), which prevents completion of cytokinesis, Dam et al. test the helicase Srs2/PARI for a role in this checkpoint pathway. The authors have produced a thorough study investigating the role of this helicase in both yeast and mammalian cells in the presence of DNA bridges. The manuscript includes clear evidence that Srs2 is important to resolve chromatin bridges, remove replication protein A (RPA) from chromatin, and delay cytokinesis under replication stress. Further, the authors show that loss of Srs2 under replication stress increases DNA damage, marked by elevated MRE11 foci in a manner dependent on cytokinesis (i.e., dependent on Cyk3). Srs2 deletion also partially abrogates the abscission delay seen upon topo-II inactivation. They further report that Srs2 must interact with PCNA to delay abscission in *S. cerevisiae*. While chromatin bridges formed when a dicentric chromosome is present escape detection by the NoCut checkpoint, inactivation of Elg1, which unloads PCNA and associated factors following DNA replication, results in delayed abscission. In HeLa cells, the Srs2 ortholog PARI is shown to similarly help promote abscission delay in the presence of DNA bridges following topoisomerase inhibition, as loss of PARI through siRNA knockdown prevents this abscission delay. Mechanistically, when PARI levels are reduced in HeLa cells, actin patches that function to stabilize the midbody and protect DNA bridges do not form/persist robustly as in cells with intact PARI. Consistent with a specific role in sensing the presence of a DNA bridge, depletion of PARI did not impact abscission checkpoint activity in response to depletion of the NPC component, Nup153. Finally, the authors show that PARI depletion reduced time to abscission to the same extent as treatment with an Aurora B inhibitor, and PARI depletion in conjunction with Aurora B inhibition did not reduce abscission timing further than singular treatments, suggesting that PARI works within the Aurora B-mediated NoCut signaling cascade.

Major comments: The manuscript is well written and, in general, the conclusions are thoroughly supported, but there are a few recommendations for addition or revision.

1. The first of these is for a more thorough introduction of helicases potentially involved in cytokinesis and more clear rationale for why the focus is on Srs2.

We appreciate the reviewer's suggestion. We expanded the Introduction to better frame the rationale for studying Srs2 in the context of cytokinesis and bridge resolution (p. 5).

2. Figure 1 E lacks statistical analysis. In addition, the text referring to 1E leads to confusion because the distinction between "RPA foci during anaphase" and "RPA coated chromatin bridges" is not made clear. The authors should clarify that the data presented in 1E shows quantification of cells with RPA foci during anaphase, not RPA coated chromatin bridges, and use consistent wording between the text and figure/figure legend. Further, how cells with RPA foci were identified, and what is classified as an RPA focus from images should be described in the methods.

We appreciate the reviewer's feedback. In the revised manuscript, we have included statistical analysis for Figure 1E and clarified the distinction between "RPA foci during anaphase" and "RPA-coated chromatin bridges" to ensure consistency (p. 7). Additionally, we have updated the Methods section to specify how cells with RPA foci were identified and what criteria were used to classify RPA foci based on the imaging data (p. 26).

3. In some cases, it is unclear whether DNA bridge formation is prevented vs aberrantly broken. For example, under Top2 inactivation, does the absence of Srs2 prevent bridge formation or promote their breakage along with premature midbody abscission? Confirming the frequency of chromatin bridge formation would address this and, further, monitoring RPA persistence would validate whether RPA clearance from bridges is consistently correlated with Srs2 activity (an interesting observation from Figure 1 that is not followed up on). Similarly, other conditions that appear to interfere with abscission delay (e.g., disrupting Srs2-PCNA interaction) should be monitored for whether the formation of DNA bridges has been altered.

We now show that chromatin bridges still form in *top2-4 srs2Δ* and *srs2* PIP and SIM mutants (new figures 3A-B and 4C). We also monitor RPA persistence in *top2* mutants (new Supplementary Fig. S2C), and find that at the high temperature needed to inactivate the Top2-4 protein, RPA foci are present with increased frequency and Srs2 deletion does not increase it further.

4. In Figure 4A, the data show that the PIP-box is required for timely abscission. Imaging data from yeast strains with the PIP-box deletion alone should be included, rather than only showing the deletion in combination with the SIM deletion.

We now include quantification of PIP-only mutants (new Fig. 4B), showing they do not rescue abscission inhibition alone.

5. While the authors state that PARI and PCNA were not detectable at bridges in mammalian cells, it would be worth examining whether RPA is persistent on DNA bridges in mammalian cells depleted of PARI to understand how closely this pathway resembles the features found in yeast.

We agree with the reviewer that examining RPA persistence on chromatin bridges in PARI-depleted cells could provide useful insight into the conservation of this pathway. We attempted to address this experimentally by performing immunofluorescence imaging of RPA in HeLa cells treated with ICRF-193, with and without PARI depletion. However, we were unable to reliably detect RPA on chromatin bridges

in most cells, regardless of PARI status, with two different anti-RPA commercially available antibodies. For transparency, we included representative images below. However, due to the low signal and inconclusive nature of the data, we chose not to include these results in the main manuscript.

6. In Figure 6, the authors should describe in the methods how cells with actin patches were identified and quantified and explain what criteria must be met to be identified as an actin patch. Actin patches were described as "disassembling more quickly" in PARI-depleted cells, but the images look as if actin patches are not forming properly in these cells. The images are crisp and clear, but a change in wording may be necessary to accurately describe the data.

We agree that the original wording in this section was unclear (see our response to Reviewer 2, comment 9), and we have revised the corresponding text to improve clarity. Specifically, we updated the terminology to refer to "actin clearance" and added a new analysis focused on actin patch dynamics. Given the high degree of variability in actin patch dynamics observed between individual cells, we chose to quantify a representative subset of cells to illustrate the overall trends. These new results are presented in **Figure 9A-B, Movies 8-11** and are described in **p. 18 and p. 30**).

Minor suggestions to improve the manuscript are:

7. Include a diagram that shows hallmarks of cell division and what is being tracked in particular assays (e.g., DNA bridge duration vs time to abscission).

Thank you for this suggestion, which we have implemented in the **new Figure S2A**.

8. In the elegant CLEM experiments presented in Figure 5, organelle labels could be added to orient the readers.

We added organelle labels to CLEM images (now Figure 6).

9. The data in supplemental Figure 2 should be moved to Figure 5. The fact that there are similar levels of chromatin bridges is vital information and stresses that the defect lies in detection and response to the bridge as opposed to formation of bridges when PARI is depleted.

We agree, and have moved Figure S2 to the main Figure as suggested (now Figure 6B).

Significance

The link between DNA bridges and NoCut/abscission checkpoint signaling is a fundamental aspect of cell cycle regulation. This manuscript makes a significant contribution to our understanding of this pathway by introducing a novel role for the helicase Srs2/PARI in execution of an abscission delay in the presence of DNA bridges. This is an important contribution as there is sparse information about cellular factors that mediate detection and response to DNA bridges, which is vital to protecting genome integrity. Although, as the authors themselves state, "the molecular mechanisms by which Srs2 and PARI function in NoCut remain unclear," this study, with some revisions, merits publication as it reveals a conserved role for a factor in this important response pathway and provides new insights into why certain DNA bridges (i.e., bridges formed by dicentric chromosomes) are not recognized by the NoCut pathway.

Additional Changes: In the Western blot showing Nup153 levels (Figure S6B in the original submission, **Figure 8A** in the revised version), a lower molecular weight band was mistakenly shown. The correct band is now presented in the main figure, and the corresponding uncropped Western blot is provided as source data (SF8). In addition, FACS profiles of HeLa cells following double-thymidine synchronization and release, which were missing from Figure S3 in the original manuscript, have now been included in **Figure S3A**.

June 23, 2025

Re: JCB manuscript #202502014R

Manuel Mendoza
Institute of Genetics and Molecular and Cellular Biology

Dear Dr. Mendoza,

Thank you for submitting your revised manuscript entitled "Srs2/PARI DNA helicases mediate abscission inhibition in response to chromatin bridges."

Your revised manuscript has now been re-reviewed by two of your original reviewers. As you will see, they find your work interesting and novel. However, they also identify some important remaining issues that need to be addressed. We concur with this view. In particular, it would be important to address points #1-3 of Reviewer 2, to use the power of the yeast system to clarify the role of the yeast protein and to make clear the differences, as currently indicated by your data, between the roles of the yeast and human proteins. If you believe that the human protein does have a protective role like the yeast protein, you may at your discretion also want to try the suggestion of Reviewer 1 to determine if the reason a protective role was not observed is because it was masked because the topoisomerase inhibitor dose used was too high.

Our general policy is that papers are considered through only one revision cycle; however, in this case we are open to one additional short round of revisions. Please submit the final revision along with a cover letter that includes a point by point response to the remaining reviewer comments.

Thank you for this interesting contribution to Journal of Cell Biology. You can contact me or the scientific editor listed below at the journal office with any questions at cellbio@rockefeller.edu.

Sincerely,

Karen Oegema, PhD
Monitoring Editor
Journal of Cell Biology

Dan Simon, PhD
Scientific Editor
Journal of Cell Biology

Reviewer #1 (Comments to the Authors (Required)):

The authors submit a revised manuscript providing new data to address whether PARI protects the genome via abscission regulation. The new results do not demonstrate a protective role by PARI, at least against binucleation, as PARI depletion does not increase binucleation when DNA bridges are induced. It is unclear why the impact of PARI depletion has not been tested on markers of genetic damage such as 53BP1 foci.

However, these new experiments are not fully conclusive. The new Figure 7F shows that ICRF-193 at 250nM has a large effect on binucleation by itself, as around of 50% of cells become binucleated under these conditions. This result suggests that protection against binucleation are unlikely to be revealed as the protective mechanisms seem to be saturated with this dose of ICRF-193. Even though 250 nM of ICRF-193 is considered low-dose, 80-100 nM can be used to avoid these masking effects on bi-nucleation.

Reviewer #2 (Comments to the Authors (Required)):

The Aurora B-mediated abscission checkpoint ("NoCut" in yeast) prevents tetraploidization or chromatin breakage in the presence of chromatin bridges and the molecular mechanism involved is a matter of active investigation. Here, Dam et al propose that, in yeast cells, the conserved Srs2 DNA helicase is required for abscission delay and for preventing chromatin bridge breakage in cytokinesis with chromatin bridges, and that association of Srs2 with PCNA is required for the inhibition of abscission. They also propose that the Srs2 human homolog PARI delays midbody severing in response to chromatin bridges; however, PARI-deficient cells do not exhibit increased chromatin bridge breakage in cytokinesis or binucleation. This manuscript

addresses a timely and very interesting topic of cell biology and the data is of generally good quality. The identification of a helicase that is essential for the abscission delay in yeast is very interesting and novel and is perhaps analogous of recent discoveries published in JCB that the TopoII topoisomerase is required for the activation of the abscission checkpoint in human cells. However, there are still some important issues that need to be addressed, especially regarding mechanistic details of the proposed model and the accurate interpretation of their data.

Point 1. PARI-deficient human cells exhibit accelerated midbody severing; however, they do not exhibit increased chromatin bridge breakage or binucleation in cytokinesis with chromatin bridges. These data suggest that, although PARI is required for some processes that occur during completion of cytokinesis (e.g., midbody disassembly) it is not essential for the abscission delay (i.e., the abscission checkpoint) in human cells. In contrast, inactivation of the Srs2 yeast homolog promotes DNA damage in cells with chromatin bridges, which is reduced in mutant strains that prevent abscission. Therefore, I believe their data are more consistent with Srs2/PARI been required for inhibition of abscission in yeast, but not in human cells. The authors explain this in paragraph 4, page 21 of the Discussion, but their findings are not accurately reflected in the Title, Summary and Abstract. In addition, although actin patches (accumulations of polymerized actin at the base of the intercellular canal) disappear before abscission, there is no evidence that their disappearance is directly linked to abscission; I would therefore refrain from calling their disassembly an "abscission event".

To accurately reflect their findings, the authors should change the title to: "The yeast Srs2 DNA helicase mediates abscission inhibition in response to chromatin bridges".

In Summary, they should change "Mendoza and colleagues reveal that the DNA helicases Srs2 and PARI mediate abscission inhibition in budding yeast and human cells with chromatin bridges", to : "Mendoza and colleagues reveal that the DNA helicase Srs2 mediates abscission inhibition in budding yeast cells with chromatin bridges".

Also, they should modify their Abstract to explain that "In human cells, PARI plays a role in delaying midbody severing and actin patch disassembly during cytokinesis, in response to chromatin bridges; however, PARI-inhibition does not correlate with increased chromatin bridge breakage or binucleation. These findings reveal a role of the Srs2 DNA helicase in modulating abscission timing in response to chromatin bridge formation in yeast". Also, the conclusion in the last paragraph of the Introduction needs to be modified accordingly.

Point 2. As explained above, I believe the most important discovery of this paper is the identification of a role for Srs2 in abscission delay in response to chromatin bridges. It will therefore be important to demonstrate that Srs2 operates through the established Aurora B (Ipl1)-pathway (NoCut), by testing the effect of simultaneous Srs2/ Aurora B inhibition in abscission timing and chromatin bridge breakage in yeast cells.

Point 3. Deletion of both the PIP and SIM boxes partially abolished the abscission delay (Figure 4). Is the same observed with deletion of the UvrD-like helicase domain? If it is not, this could further strengthen the hypothesis that association of Srs2 with PCNA is required for full abscission inhibition in yeast, especially in the absence of direct evidence Srs2-PCNA interaction on chromatin bridges.

Point 4. Is PARI delaying midbody cleavage through PCNA in cytokinesis with chromatin bridges in human cells? In other words, can the authors investigate cells depleted of PCNA and double depleted PARI/PCNA cells and compare them with PARI-depleted cells for this purpose? This would demonstrate potential similarities in Srs2/PARI mechanisms during cytokinesis completion in yeast and human cells.

Point 5. Aurora B activity promotes midbody MT stability (e.g., doi.org/10.1016/j.celrep.2025.115238). Also, PARI regulates MT stability through an Aurora B-dependent pathway, but not by regulating Aurora B-T232 phosphorylation. The authors should briefly discuss how this could happen.

They should also discuss how PARI could fit into the recently published Topo2 signaling pathway that activates the abscission checkpoint in human cells (DOI: 10.1083/jcb.202303123) and DOI: 10.1083/jcb.202008029).

Point by point response to reviewers

Reviewer #1

The authors submit a revised manuscript providing new data to address whether PARI protects the genome via abscission regulation. The new results do not demonstrate a protective role by PARI, at least against binucleation, as PARI depletion does not increase binucleation when DNA bridges are induced. It is unclear why the impact of PARI depletion has not been tested on markers of genetic damage such as 53BP1 foci.

However, these new experiments are not fully conclusive. The new Figure 7F shows that ICRF-193 at 250nM has a large effect on binucleation by itself, as around of 50% of cells become binucleated under these conditions. This result suggests that protection against binucleation are unlikely to be revealed as the protective mechanisms seem to be saturated with this dose of ICRF-193. Even though 250 nM of ICRF-193 is considered low-dose, 80-100 nM can be used to avoid these masking effects on bi-nucleation.

Response:

We thank the reviewer for this thoughtful suggestion. While we agree that testing a lower ICRF-193 dose might offer insight into subtle effects, our goal was to determine whether PARI plays a measurable protective role under conditions known to robustly activate the abscission checkpoint. Under these conditions, PARI depletion fails to increase binucleation or chromatin bridge breakage, arguing against a strong protective role. We now explicitly discuss this limitation in the Discussion (page 23, paragraph 2) and cite the reviewer's suggestion as an avenue for future work.

Reviewer #2

The Aurora B-mediated abscission checkpoint ("NoCut" in yeast) prevents tetraploidization or chromatin breakage in the presence of chromatin bridges and the molecular mechanism involved is a matter of active investigation. Here, Dam et al propose that, in yeast cells, the conserved Srs2 DNA helicase is required for abscission delay and for preventing chromatin bridge breakage in cytokinesis with chromatin bridges, and that association of Srs2 with PCNA is required for the inhibition of abscission. They also propose that the Srs2 human homolog PARI delays midbody severing in response to chromatin bridges; however, PARI-deficient cells do not exhibit increased chromatin bridge breakage in cytokinesis or binucleation. This manuscript addresses a timely and very interesting topic of cell biology and the data is of generally good quality. The identification of a helicase that is essential for the abscission delay in yeast is very interesting and novel and is perhaps analogous of recent discoveries published in JCB that the Topoll topoisomerase is required for the activation of the abscission checkpoint

in human cells. However, there are still some important issues that need to be addressed, especially regarding mechanistic details of the proposed model and the accurate interpretation of their data.

Point 1. PARI-deficient human cells exhibit accelerated midbody severing; however, they do not exhibit increased chromatin bridge breakage or binucleation in cytokinesis with chromatin bridges. These data suggest that, although PARI is required for some processes that occur during completion of cytokinesis (e.g., midbody disassembly) it is not essential for the abscission delay (i.e., the abscission checkpoint) in human cells. In contrast, inactivation of the Srs2 yeast homolog promotes DNA damage in cells with chromatin bridges, which is reduced in mutant strains that prevent abscission. Therefore, I believe their data are more consistent with Srs2/PARI been required for inhibition of abscission in yeast, but not in human cells. The authors explain this in paragraph 4, page 21 of the Discussion, but their findings are not accurately reflected in the Title, Summary and Abstract. In addition, although actin patches (accumulations of polymerized actin at the base of the intercellular canal) disappear before abscission, there is no evidence that their disappearance is directly linked to abscission; I would therefore refrain from calling their disassembly an "abscission event".

To accurately reflect their findings, the authors should change the title to: "The yeast Srs2 DNA helicase mediates abscission inhibition in response to chromatin bridges".

In Summary, they should change "Mendoza and colleagues reveal that the DNA helicases Srs2 and PARI mediate abscission inhibition in budding yeast and human cells with chromatin bridges", to : "Mendoza and colleagues reveal that the DNA helicase Srs2 mediates abscission inhibition in budding yeast cells with chromatin bridges".

Also, they should modify their Abstract to explain that "In human cells, PARI plays a role in delaying midbody severing and actin patch disassembly during cytokinesis, in response to chromatin bridges; however, PARI-inhibition does not correlate with increased chromatin bridge breakage or binucleation. These findings reveal a role of the Srs2 DNA helicase in modulating abscission timing in response to chromatin bridge formation in yeast". Also, the conclusion in the last paragraph of the Introduction needs to be modified accordingly.

Response:

We agree that our data clearly demonstrate a requirement for Srs2 in inhibiting abscission in yeast, whereas the evidence for a comparable role of PARI in human cells, while suggestive, is not conclusive. To reflect this distinction more accurately, we have revised the title, summary, abstract, and final paragraph of the Introduction. At the same time, we believe it is important to acknowledge the evidence supporting PARI's involvement in regulating abscission-associated events. Therefore we have implemented the following changes:

- The title now reads: "Roles of Srs2/PARI-family DNA helicases in NoCut checkpoint signaling and abscission regulation"

- The summary and abstract now specify that only Srs2 mediates abscission inhibition, and that PARI delays abscission-associated events.
- Additionally, in line with the reviewer's suggestion, we refer to actin patch disassembly as an abscission-associated event, rather than an abscission event per se.

Point 2. As explained above, I believe the most important discovery of this paper is the identification of a role for Srs2 in abscission delay in response to chromatin bridges. It will therefore be important to demonstrate that Srs2 operates through the established Aurora B (Ipl1)-pathway (NoCut), by testing the effect of simultaneous Srs2/ Aurora B inhibition in abscission timing and chromatin bridge breakage in yeast cells.

Response:

We thank the reviewer for this important suggestion. We now include new epistasis data (Fig. 3E-F) showing that *srs2Δ ip11-321* double mutants do not delay abscission more than *ip11-321* alone, suggesting that Srs2 functions in the same pathway as Ipl1/Aurora B. This is discussed in the Results (page 10) and the Discussion (page 22).

Point 3. Deletion of both the PIP and SIM boxes partially abolished the abscission delay (Figure 4). Is the same observed with deletion of the UvrD-like helicase domain? If it is not, this could further strengthen the hypothesis that association of Srs2 with PCNA is required for full abscission inhibition in yeast, especially in the absence of direct evidence Srs2-PCNA interaction on chromatin bridges.

Response:

Yes. We now include new data using the helicase-dead *srs2-R337S* mutant, which retains PCNA-binding but lacks helicase activity. This mutant does not suppress abscission inhibition (Fig. 4D), indicating that helicase activity is not required. This supports our model that Srs2 functions via PCNA interaction. See page 11 and Discussion page 22.

Point 4. Is PARI delaying midbody cleavage through PCNA in cytokinesis with chromatin bridges in human cells? In other words, can the authors investigate cells depleted of PCNA and double depleted PARI/PCNA cells and compare them with PARI-depleted cells for this purpose? This would demonstrate potential similarities in Srs2/PARI mechanisms during cytokinesis completion in yeast and human cells.

Response:

This is an excellent suggestion; however, due to the essential function of PCNA in DNA replication, its depletion may introduce confounding effects that would complicate the interpretation of any additive interaction with PARI. For this reason, we opted not to pursue this experiment within the scope of the current study.

Point 5. Aurora B activity promotes midbody MT stability (e.g., doi.org/10.1016/j.celrep.2025.115238). Also, PARI regulates MT stability through an Aurora B-dependent pathway, but not by regulating Aurora B-T232 phosphorylation. The authors should briefly discuss how this could happen.

They should also discuss how PARI could fit into the recently published Topo2 signaling pathway that activates the abscission checkpoint in human cells (DOI: 10.1083/jcb.202303123) and DOI: 10.1083/jcb.202008029).

Response:

We thank the reviewer for these insightful comments. While our data show that PARI functions in an Aurora B-dependent pathway, we did not observe changes in Aurora B T232 phosphorylation upon PARI depletion. One possibility is that PARI regulates Aurora B activity not by affecting its activation state, but rather by modulating its substrate specificity during cytokinesis. However, this remains speculative and warrants future investigation.

Regarding the potential integration of PARI into the recently described Topo II-Aurora B signaling pathway, we addressed this point in our previous revision. Specifically, we discuss how PARI might interface with this pathway on page 26 of the revised Discussion.

September 10, 2025

RE: JCB Manuscript #202502014RR

Manuel Mendoza
Institute of Genetics and Molecular and Cellular Biology

Dear Dr. Mendoza,

Thank you for submitting your revised manuscript entitled "Roles of Srs2/PARI-family DNA helicases in NoCut checkpoint signaling and abscission regulation." We would be happy to publish your paper in JCB pending final revisions necessary to meet our formatting guidelines (see details below).

A. MANUSCRIPT ORGANIZATION AND FORMATTING:

1) Text limits: Character count for Articles is < 40,000, not including spaces. Count includes title page, abstract, introduction, results, discussion, and acknowledgments. Count does not include materials and methods, figure legends, references, tables, or supplemental legends.

2) Figure formatting: Articles may have up to 10 main text figures. Scale bars must be present on all microscopy images, including inset magnifications. Molecular weight or nucleic acid size markers must be included on all gel electrophoresis. Please add scale bars to figures 2A/D & 9A and to magnifications in 3C, 6B/D, 10C, & S4.

Also, please avoid pairing red and green for images and graphs to ensure legibility for color-blind readers. If red and green are paired for images, please ensure that the particular red and green hues used in micrographs are distinctive with any of the colorblind types. If not, please modify colors accordingly or provide separate images of the individual channels.

3) Statistical analysis: Error bars on graphic representations of numerical data must be clearly described in the figure legend. The number of independent data points (n) represented in a graph must be indicated in the legend. Please indicate whether 'n' refers to technical or biological replicates (i.e. number of analyzed cells, samples or animals, number of independent experiments). If independent experiments with multiple biological replicates have been performed, we recommend using distribution-reproducibility SuperPlots (please see Lord et al., JCB 2020) to better display the distribution of the entire dataset, and report statistics (such as means, error bars, and P values) that address the reproducibility of the findings.

Statistical methods should be explained in full in the materials and methods. For figures presenting pooled data the statistical measure should be defined in the figure legends. Please also be sure to indicate the statistical tests used in each of your experiments (both in the figure legend itself and in a separate methods section) as well as the parameters of the test (for example, if you ran a t-test, please indicate if it was one- or two-sided, etc.). Also, if you used parametric tests, please indicate if the data distribution was tested for normality (and if so, how). If not, you must state something to the effect that "Data distribution was assumed to be normal but this was not formally tested."

4) Materials and methods: Should be comprehensive and not simply reference a previous publication for details on how an experiment was performed. Please provide full descriptions (at least in brief) in the text for readers who may not have access to referenced manuscripts. The text should not refer to methods "...as previously described." Please also describe the SDS-PAGE and immunoblotting methods including the type of membrane used and the acquisition and quantification methods.

5) For all cell lines, vectors, strains, constructs/cDNAs, etc. - all genetic material: please include database / vendor ID (e.g. Addgene, ATCC, etc.) or if unavailable, please briefly describe their basic genetic features, even if described in other published work or gifted to you by other investigators (and provide references where appropriate). Please be sure to provide the sequences for all of your oligos: primers, si/shRNA, RNAi, gRNAs, etc. in the materials and methods. You must also indicate in the methods the source, species, and catalog numbers/vendor identifiers (where appropriate) for all of your antibodies, including secondary. If antibodies are not commercial, please add a reference citation if possible.

6) Microscope image acquisition: The following information must be provided about the acquisition and processing of images:
a. Make and model of microscope
b. Type, magnification, and numerical aperture of the objective lenses

- c. Temperature
- d. Imaging medium
- e. Fluorochromes
- f. Camera make and model
- g. Acquisition software
- h. Any software used for image processing subsequent to data acquisition. Please include details and types of operations involved (e.g., type of deconvolution, 3D reconstitutions, surface or volume rendering, gamma adjustments, etc.).

7) References: There is no limit to the number of references cited in a manuscript. References should be cited parenthetically in the text by author and year of publication. Abbreviate the names of journals according to PubMed.

8) Supplemental materials: Articles may have up to 5 supplemental figures and 10 videos. Please also note that tables, like figures, should be provided as individual, editable files. A summary of all supplemental material should appear at the end of the Materials and methods section. Please include one brief sentence per item.

9) Video legends: Should describe what is being shown, the cell type or tissue being viewed (including relevant cell treatments, concentration and duration, or transfection), the imaging method (e.g., time-lapse epifluorescence microscopy), what each color represents, how often frames were collected, the frames/second display rate, and the number of any figure that has related video stills or images.

10) eTOC summary: A ~40-50 word summary that describes the context and significance of the findings for a general readership should be included on the title page. The statement should be written in the present tense and refer to the work in the third person. It should begin with "First author name(s) et al..." to match our preferred style.

11) Conflict of interest statement: JCB requires inclusion of a statement in the acknowledgements regarding competing financial interests. If no competing financial interests exist, please include the following statement: "The authors declare no competing financial interests." If competing interests are declared, please follow your statement of these competing interests with the following statement: "The authors declare no further competing financial interests."

12) A separate author contribution section is required following the Acknowledgments in all research manuscripts. All authors should be mentioned and designated by their first and middle initials and full surnames. We encourage use of the CRediT nomenclature (<https://casrai.org/credit/>).

13) ORCID IDs: ORCID IDs are unique identifiers allowing researchers to create a record of their various scholarly contributions in a single place. Please note that ORCID IDs are required for all authors. At resubmission of your final files, please be sure to provide your ORCID ID and those of all co-authors.

14) JCB requires authors to submit Source Data used to generate figures containing gels and Western blots with all revised manuscripts. This Source Data consists of fully uncropped and unprocessed images for each gel/blot displayed in the main and supplemental figures. For assays performed using capillary electrophoresis and/or immunoassay-based detection, authors should instead provide the electropherogram graph(s) for each experiment, plotting fluorescence/chemiluminescence intensity vs. molecular weight/size. Since your paper includes cropped gel and/or blot images, please be sure to provide one Source Data file for each figure gels, blots, and/or capillary electrophoresis assays along with your revised manuscript files. File names for Source Data figures should be alphanumeric without any spaces or special characters (i.e., SourceDataF#, where F# refers to the associated main figure number or SourceDataFS# for those associated with Supplementary figures). For traditional gels and blots, the lanes of the gels/blots should be labeled as they are in the associated figure, the place where cropping was applied should be marked (with a box), and molecular weight/size standards should be labeled wherever possible. For capillary electrophoresis assays, each trace in the graph should be color-coded and labeled to indicate which protein, gene, or sample is being measured (please try to avoid red/green combinations to accommodate our color-blind readers).

Source Data files will be directly linked to specific figures in the published article. Source Data Figures should be provided as individual PDF files (one file per figure). Authors should endeavor to retain a minimum resolution of 300 dpi or pixels per inch. Please review our instructions for export from Photoshop, Illustrator, and PowerPoint here: <https://rupress.org/jcb/pages/submission-guidelines#revised>

15) Journal of Cell Biology now requires a data availability statement for all research article submissions. These statements will be published in the article directly above the Acknowledgments. The statement should address all data underlying the research presented in the manuscript. Please visit the JCB instructions for authors for guidelines and examples of statements at (<https://rupress.org/jcb/pages/editorial-policies#data-availability-statement>).

B. FINAL FILES:

****It is JCB policy that if requested, original data images must be made available to the editors. Failure to provide original images upon request will result in unavoidable delays in publication. Please ensure that you have access to all original data images prior to final submission.****

****The license to publish form must be signed before your manuscript can be sent to production. A link to the electronic license to publish form will be sent to the corresponding author only. Please take a moment to check your funder requirements before choosing the appropriate license.****

Thank you for your attention to these final processing requirements. Please revise and format the manuscript and upload materials within 7 days. If you need an extension for whatever reason, please let us know and we can work with you to determine a suitable revision period.

Thank you for this interesting contribution, we look forward to publishing your paper in Journal of Cell Biology.

Sincerely,

Karen Oegema, PhD
Monitoring Editor
Journal of Cell Biology

Dan Simon, PhD
Scientific Editor
Journal of Cell Biology